# UQ: Assessing Language Models on Unsolved Questions

## Abstract

Benchmarks shape progress in AI research. A useful benchmark should be both *difficult* and *realistic*: questions should challenge frontier models while also reflecting real-world usage. Yet, current paradigms face a difficulty–realism tension: exam-style benchmarks are often made artificially difficult with limited real-world value, while benchmarks based on real user interaction often skew toward easy, high-frequency problems. In this work, we explore a radically different paradigm: assessing models on *unsolved* questions. Rather than a static benchmark scored once, we curate unsolved questions and evaluate models asynchronously over time with validator-assisted screening and community verification. We introduce UQ, a testbed of 500 challenging, diverse questions sourced from Stack Exchange, spanning topics from CS theory and math to sci-fi and history, probing capabilities including reasoning, factuality, and browsing. UQ is difficult and realistic by construction: unsolved questions are often hard and naturally arise when humans seek answers, thus solving them yields direct real-world value. Our contributions are threefold: (1) UQ-Dataset and its collection pipeline combining rule-based filters, LLM judges, and human review to ensure question quality (e.g., well-defined and difficult); (2) UQ-Validators, compound validation strategies that leverage the generator-validator gap to provide evaluation signals and pre-screen candidate solutions for human review; and (3) UQ-Platform, an open platform where experts collectively verify questions and solutions, enabling ongoing, asynchronous evaluation. The top-performing model passes UQ-validation on only 15% of questions, and preliminary human verification has already identified correct answers among those that passed. UQ charts a path for evaluating frontier models on real-world, open-ended challenges, where success pushes the frontier of human knowledge.

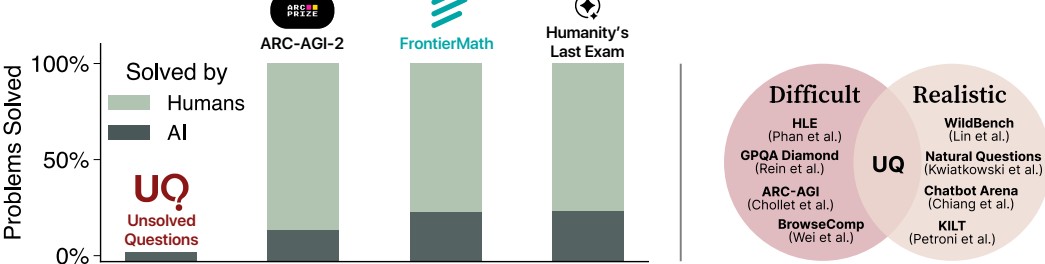

Figure 1: **Motivations of UQ.** *Left*: Many existing benchmarks consist of problems already solved by humans; in contrast, UQ focuses on the hard, open-ended problems where we most want progress. *Right*: Difficulty-realism tension in prior benchmarks motivates UQ as a new evaluation paradigm.

## 1 Introduction

Benchmarks play a pivotal role in measuring and guiding progress (Patterson, 2012). Yet, the capabilities of large language models (LLMs) continue to outpace the discriminative power of existing benchmarks. Benchmarks once considered difficult, such as MMLU (Hendrycks et al., 2020), GPQA (Rein et al., 2024), and ARC-AGI-1 (Chollet et al., 2024), have quickly become saturated by frontier models. A striking example is "Humanity's Last Exam (HLE)" (Phan et al., 2025), a benchmark explicitly designed to combat this trend by featuring the hardest evaluation problems conceived by domain experts. Despite this effort, the top performing AI system increased from 9.1% (o1, OpenAI (2024)) to 26.6% (Deep Research, OpenAI (2025)) within weeks of its initial release.

Amid the surge in model capability and the need for better model evaluation, it is worth revisiting two of the most important properties that make a benchmark meaningful:

1. **Difficult:** The benchmark should be challenging for frontier models; and
2. **Realistic:** The benchmark should reflect natural queries where answers offer real-world value.

While simple to state, these properties—and the lack thereof—largely explain the limitations of several existing benchmarks. Two prevalent paradigms in recent literature help illustrate. The first is *exam-based* benchmarking, where models are scored against evaluation questions with known, human-annotated answers (e.g., Hendrycks et al. (2020); Wang et al. (2024b); of America (2024); Rein et al. (2024); Phan et al. (2025); Wei et al. (2024); Glazer et al. (2024)). While exams can be made difficult (Phan et al., 2025; Rein et al., 2024), they are inherently unrealistic: solutions are known, and with rapid model improvement, attempts to (artificially) increase difficulty often induce a distribution shift between benchmark problems and real-world user queries. The second are benchmarks that emphasize real-world usage, where users submit authentic queries and seek answers for an actual information need (e.g., Kwiatkowski et al. (2019); Liu et al. (2023); Petroni et al. (2020); Chiang et al. (2024); Lin et al. (2024)). While realism is crucial, the reliance on user-submitted queries can introduce questions that are easy to articulate, frequently asked, and well-trodden. This leads to two limitations: many such benchmarks are now near saturation (e.g., Kwiatkowski et al. (2019); Lin et al. (2024)), and benchmarks that rely on unmoderated user interaction may be susceptible to manipulation when incentives misalign (Zhao et al., 2024; Huang et al., 2025b; Singh et al., 2025).

**These limitations motivate us to explore a radically different evaluation paradigm: assessing models on *unsolved* questions.** By construction, unsolved questions are often both *difficult*—since no known solution exists—and *realistic*—arising naturally in settings where humans seek answers. Unlike exam-based benchmarks, they are not contrived for difficulty; and unlike benchmarks designed to solicit user queries, unsolved questions emerge organically from information-seeking and carry intrinsic value that is independent of model performance and ranking. Progress on unsolved questions would also imply novel insights or solutions, making benchmark improvement inherently meaningful. In exchange for these benefits, unsolved questions introduce two primary challenges for benchmarking purposes: without ground-truth answers, we need to: (1) validate the difficulty and quality of questions; and (2) assess candidate solutions produced by different models.

We instantiate this new paradigm by introducing **UQ**, a testbed of 500 curated, *unsolved* questions sourced from Stack Exchange, a diverse network of Q&A websites. **UQ** consists of three parts:

1. **UQ-Dataset (§2):** A collection of unsolved questions curated through a three-stage pipeline: (i) rule-based filters on unanswered questions using engagement signals (e.g., views, votes, comments, age); (ii) LLM-based filtering for well-definedness, difficulty, approachability, and objectiveness; and (iii) human review by PhD-level annotators across STEM and non-STEM domains. This yields a diverse set of hard, high-quality, and open questions spanning from mathematics, physics, CS theory to bioacoustics, sci-fi, mythology, and more. See Appendix I.1 for sample questions.

2. **UQ-Validators (§3):** A set of LLM-based validation strategies designed to assess candidate LLM solutions. We leverage the observation that frontier models are better at validating solutions than generating them, and that such generator-validator gap shows transfer across datasets. We explore a hierarchical validation framework for candidate answers, combining (i) *low-level* checks, such as factual/logical correctness and question-answer cycle-consistency; (ii) *mid-level* sampling strategies, including repeated and iterated judgments; and (iii) *high-level* aggregation strategies like majority vote, unanimous vote, and sequential verification. **UQ**-Validators serve as the first stage of the evaluation cycle by attempting to rule out false answers for human verification.

3. **UQ-Platform (§4):** A live, open platform that completes the model evaluation cycle. It hosts unsolved questions with candidate model answers, **UQ**-validation results, and full provenance (prompts/metadata) for reproducibility. It also serves as the central hub for user and model developer contributions (submitting questions, answers, reviews, and ratings), enabling the crucial, continuous community-driven evaluation central to our new evaluation paradigm.

While it is possible that a question in **UQ** is posted and solved elsewhere, a correct solution remains valuable to the original asker, and **UQ** can serve as a go-to repository for questions that are challenging to LLMs. As models improve and questions get solved, **UQ** is positioned to draw from our pool of over 7,000+ candidate questions as well as from public, community-driven contributions (e.g., new unsolved questions on Stack Exchange and other sources).

**An important caveat is that unsolved questions often preclude perfect automated evaluation.**
Accordingly, **UQ** should be viewed as its distinct components: **UQ**-Dataset provides standalone
and grounded model inputs to stress test frontier models; **UQ**-Validators provide useful signals to
human expert reviewers while offering a foundation to study oracle-free validation; and **UQ**-Platform
facilitates community engagement where solution to each problem serves to advance knowledge and
guide model evaluations. We hope that **UQ** serves to accelerate future research on scaling model
capabilities in domains without ground-truth reward or verifiers.

## 2 **UQ**-DATASET: UNSOLVED QUESTIONS WITH DESIRABLE PROPERTIES

The **UQ**-Dataset consists of 500 challenging, *unsolved* questions. We carefully select them through a
three-stage filtering pipeline from over 3,000,000 unanswered questions across 80 sites on the Stack
Exchange network, as illustrated in Figure 2.

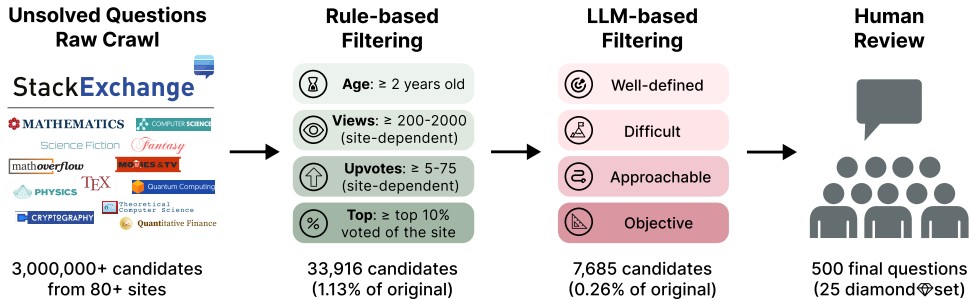

Figure 2: **UQ**-**Dataset creation pipeline**. We first crawl unsolved questions from the Stack Exchange
network and apply rule-based filters using engagement metrics. LLM judges then select for desirable
properties (e.g., difficulty and well-definedness). Finally, human reviews filter remaining questions
into the final dataset. See Appendix I.1 for sample questions.

### 2.1 DATASET CREATION

**Overview.** The dataset creation pipeline comprises of the following stages: we first crawl questions
with the Stack Exchange API (`api.stackexchange.com`), then we filter them using heuristic rules,
followed by LLM quality judgment, and finally using human reviews, as illustrated in Figure 2.

Each question in **UQ**-Dataset includes a title, a question body (detailed description of the problem in
markdown), relevant keywords for domain categorization, list of comments posted under the question,
and the originating site name for context. For filtering and validation, we only use the title, body, and
site information. See the **UQ**-Platform for a visualization (Section 4).

**Stage 1: Rule-Based Filtering.** We first apply a set of default heuristic rules, then refine them with
site-specific thresholds based on site popularity (e.g., mathematics vs. history). A subset of the rules
is shown in Figure 2 and the full list shown in Appendix D.3. These heuristic rules balance question
quality with dataset scale and trim $\approx 99\%$ of the vast pool (millions) of unanswered questions.

**Stage 2: LLM-Based Filtering.** We then use LLMs to screen each candidate question by checking
for benchmark-relevant properties. We use a dual-model approach where a general-purpose model
(e.g., GPT-4o) first attempts to answer each question, then a reasoning model (e.g., o4-mini) assesses
the question in conjunction with the generated answer based on the following five criteria:

- *Well-defined*: Whether the question is well-specified and clear (Yes/No).
- *Difficult by candidate correctness*: Likelihood that the attempted answer is correct (0-100%).
- *Difficult by solvability*: Likelihood that domain experts can solve the question (0-100%).
- *Approachable*: Whether the question is logically sound and solvable in principle (Yes/No).
- *Objective*: Whether the true answer is objective and verifiable (Yes/No).

Each criteria is evaluated independently with three repeated LLM calls. We compute an average
for the numerical criteria (answer correctness and expert solvability) and take unanimous vote for
the binary criteria (well-defined, approachable, objective). We consider questions that satisfy all
binary criteria, have an average of $\leq 40\%$ answer correctness, and have an average of $\leq 70\%$ expert
solvability to be high-quality and pass them to human review. See prompt details in Appendix I.6.

**Stage 3: Manual Filtering.** We then present each candidate question, along with its engagement signals, metadata, and three attempted answers from OpenAI o3, Gemini 2.5 Pro, and Claude 3.7 Sonnet to human reviewers. Reviewers assess the questions quality using their discretion, taking into account the question content and the plausibility of model answers (e.g., question may be hard if model answers are clearly wrong/hallucinated). For many sites, we defer to community moderation and simply select the top-$k$ unanswered questions. See Appendix D.4 for details.

**UQ Diamond Subset.** Inspired by GPQA (Rein et al., 2024), we select a high-quality subset of 25 questions as the *diamond* subset. Our selection is driven by organic engagement signals on Stack Exchange. Our intuition is that high engagement correlates with heavy moderation on Stack Exchange and is a reliable proxy for question quality and human relevance. See Appendix D.6 for details.

## 2.2 DATASET ANALYSIS

**Filtering Statistics.** Each stage of the question collection pipeline progressively prunes the question pool. Starting from roughly 3 million raw question candidates from 80 Stack Exchange sites (e.g., Math Overflow, Physics), the rule-based filtering trims the pool to 33,916 (1.13% of the original pool); the LLM-based filtering prunes to 7,685 (0.26% of the original); and human reviewing (e.g., discarding residual duplicates, near-trivial, off-topic, or policy-violating questions) yields a curated set of 500 items (0.02%). We defer additional topic-level statistics to Appendix D.

**Question Composition.** Figure 3 illustrates the composition of the UQ-Dataset across high-level domains (e.g., Science, Technology; as labeled by Stack Exchange) and across different filtering stages (Section 2.1). The majority of the dataset consists of *Science* questions (domain includes sites such as Cross Validated, MathOverflow, and Physics), followed by *Technology* (e.g., Stack Overflow) and *Life & Arts* (e.g., Puzzling). We also observe that questions from different domains probe for different model capabilities; for example, math questions often call for open-ended proofs, whereas questions on science fiction & fantasy bias towards browsing capabilities (e.g., identifying the name of a book based on partial plots); see Appendix I.1 for sample questions.

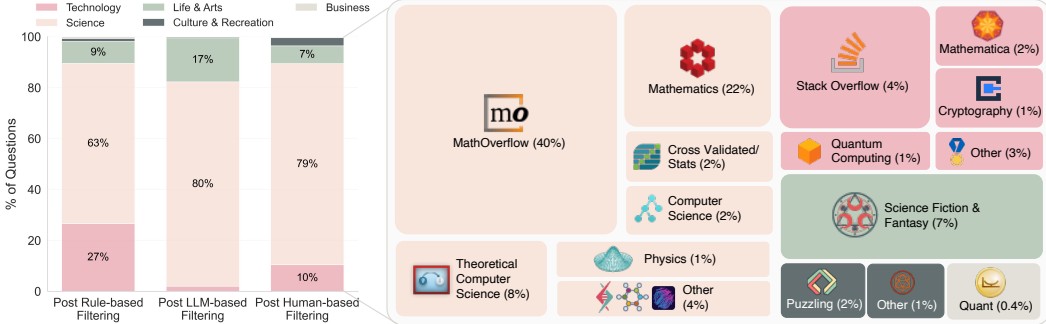

Figure 3: **Question composition of the UQ-Dataset.** Left: high-level composition across each of the three-stage filtering. We categorize the sites according to official StackExchange categories. Right: composition by Stack Exchange sites (panels not drawn to scale).

## 2.3 DATASET CURATION AND UPDATES

The UQ-Dataset can function as a semi-live dataset. Over time, we check whether any questions in the dataset have received accepted answers on Stack Exchange (where humans submit answers) or the UQ-Platform (where we accept AI answers). If a question is considered solved (e.g., a proposed answer is accepted by the original poster on Stack Exchange), we may consider removing and replacing it in future dataset versions; see Appendix D.7 for discussion on dataset updates.

If an answer is human-verified to be correct, we mark the question as resolved and credit the corresponding model in the semi-live model ranking (see Section 4). The dataset is designed to support continuous refreshes with new, verified unsolved questions, allowing UQ to evolve as a dynamic benchmark for evaluating frontier models.

## 3 UQ-VALIDATORS: REFERENCE-FREE VALIDATION OF MODEL SOLUTIONS

While the curated UQ-Dataset is a valuable artifact on its own, it needs scoring metrics to function as a benchmark of model performance. However, the absence of ground-truth answers precludes

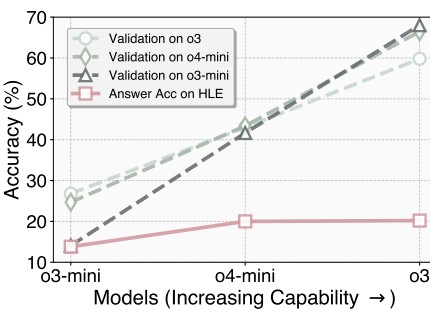
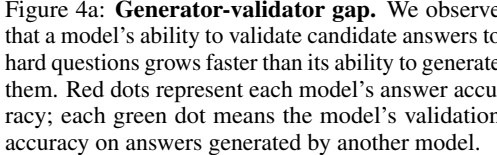
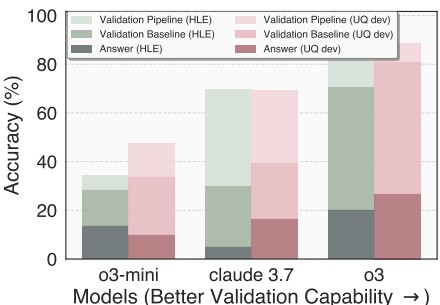

Figure 4a: **Generator-validator gap.** We observe that a model's ability to validate candidate answers to hard questions grows faster than its ability to generate them. Red dots represent each model's answer accuracy; each green dot means the model's validation accuracy on answers generated by another model.

Figure 4b: **Validator performance shows transfer.** The same models and judgment prompts tested on HLE transfer directly to the held-out development set of UQ-Dataset. Validation baseline means using only the *Correctness* strategy while validation pipeline means the 3-iter pipeline (to be discussed in §3.3).

automated verification as in exam-style benchmarks (e.g., Phan et al. (2025); Rein et al. (2024); Glazer et al. (2024)). This motivates our exploration of *oracle-free validators*—evaluation strategies that examine a question and a candidate answer and provide useful signal on answer correctness and model performance. The main goal of these validators is to *rule out false candidate answers*, rather than to prove a candidate answer's correctness; to make this distinction, we use the term "validator" as opposed to "judge" or "verifier" where appropriate. We also note that domain-specific setups may allow for more powerful, oracle-free validators (e.g., proof assistants such as Lean (Moura & Ullrich, 2021)). We aim to keep UQ-Validators to strategies that may generalize across diverse questions in UQ-Dataset, which may in turn limit validation performance; see Appendix E.1 for more discussion.

**Evaluating validators via surrogate data.** Since the evaluation of the validators itself requires ground-truth answers (e.g., how accurate are validation verdicts and how well do they match human judgment), we use Humanity's Last Exam (HLE) (Phan et al., 2025) as a challenging *surrogate* dataset. HLE's difficulty and diversity resemble that of UQ-Dataset while providing ground-truths we can use to score and compare different validation approaches. We remove multiple-choice questions (MCQs) from the text subset of HLE (2,158 questions) to better align with our target setting, and then randomly sample 500 questions. We acknowledge that alternative approaches exist (e.g, recruiting human experts to assess validation attempts), but they may be costly and difficult to scale. We motivate the use of surrogate datasets in the next section, and leave other directions to future work.

## 3.1 MOTIVATION: GENERATOR-VALIDATOR GAP WIDENS WITH MODEL CAPABILITY AND SHOWS CROSS-DATASET TRANSFER

A key motivation for developing oracle-free validators is our hypothesis that *verifying candidate answers to hard questions may be easier than generating them*. We begin by empirically testing this hypothesis in our setting. We first evaluate models of increasing capability (e.g., o3-mini → o4-mini → o3) on the HLE sample, obtaining each model's *answer accuracy*. We then ask each model to validate every other model's answers without access to the ground-truth answers and then evaluate these verdicts against the ground-truths to obtain *validation accuracy*. Figure 4a shows that as model capabilities increase, models improve more rapidly on validation accuracy than on answer accuracy. Notably, even though the strongest model has poor answer accuracy (e.g., o3 at 20%), it achieves a non-trivial validation accuracy of 65%. See more results in Appendix E.2.

Next, we examine the *transferability* of validator performance. Transfer is desirable because if a validator generalizes across datasets without modification, we gain confidence that it offers useful signal when assessing answers to unsolved questions. To test transfer, we apply the same validators evaluated on HLE directly to the held-out development set of UQ-Dataset without additional tuning. Figure 4b shows that their accuracy patterns and the generator-validator gaps closely mirror those observed on HLE, confirming meaningful transfer. The widening generator-validator gap, together with its transfer, provide empirical support for developing oracle-free validators using surrogate data.

## 3.2 VALIDATOR DESIGN GOAL AND STRATEGIES

**Design goal.** In the context of oracle-free validation, we say that *false positives* are candidate answers that are incorrect but passed a validator, and *false negatives* are candidate answers that are actually

correct but failed a validator. While achieving low false negatives (i.e., high recall) is desirable, an effective validator should prioritize low false positives (i.e., high precision); that is, it should be conservative when approving candidate answers. This is preferable for two reasons: first, unsolved questions are often hard but may appear easy, increasing the risk of models generating and approving incorrect but promising-looking answers; second, high precision minimizes the need for costly human expert verification of passed answers.

**Strategies.** With the design goal in mind, we consider a hierarchical design space of validation strategies across three levels of abstraction: low-level reasoning, mid-level judgment refinement, and high-level decision aggregation. Conceptually, a low-level strategy is an (elaborate) prompt for an LLM judge, and a mid- and high-level strategy is a prompt or scaffold that composes LLM calls into a pipeline. All prompts are provided in Appendix I.7. Specifically:

*Low-level strategies* are prompting techniques to assess basic properties of a candidate answer:

- *Correctness:* Judge whether the answer is both accurate and complete with respect to the question;
- *Fact/logic check:* Check factual, arithmetic, and logical errors within the answer;
- *Cycle consistency:* Infer the question that would have led to the given answer, then compare it to the original prompt. This probes whether the answer meaningfully engages with the question.

*Mid-level strategies* are methods to improve judgment robustness via redundancy and self-audit:

- *Repeated sampling:* Sample validators with random seeds to gather multiple validation verdicts;
- *Iterated reflection:* Prompt judge models to re-evaluate and potentially revise its initial judgment across multiple reflection iterations.

*High-level strategies* are approaches to consolidate multiple judgments into final verdicts:

- *Majority voting:* Accept the answer if a majority of validation results (e.g., across instances of low- or mid-level strategies) are positive;
- *Unanimous voting:* Similar to the above, but accept the answer only if *all* judgments are positive;
- *Pipeline verification:* Organize validator strategies into turns (or stages) where an answer proceeds to the next stage only if it passes the current stage. Pipelines use three turns unless otherwise stated.

A UQ-Validator is a composition of these strategies, whether within or across abstraction levels. For example, a simple validator may prompt a base model to check for *correctness*, repeat with three independent samples from the model, and aggregate with unanimous voting. A performant UQ-Validator, shown in Figure S2, employs pipeline verification (high-level) with iterative reflection (mid-level) of cycle consistency, fact/logic check, and correctness check (low-level) in each turn. Different strategy compositions yield validators of different properties (e.g., cost and strictness); we provide a comparison in Section 3.3. See Appendix I.7 for the prompts used for each strategy.

### 3.3 RESULTS ON UQ-VALIDATORS

We now empirically assess different validation strategies and report our findings. Unless otherwise stated, we use 500 randomly sampled HLE questions as surrogate data. For each question, we elicit answers from five models (o3, o4-mini, o3-mini, Gemini 2.5 Pro, and Claude 3.7 Sonnet), producing a total of 2,500 question-answer pairs when reporting answer and validation metrics. We defer additional findings, such as scaling behavior, to Appendix E.5.

**Finding #1: Compound Validator Strategies Outperform Simple Prompting Baselines**

At a macro level, we first find that compound validation strategies generally improve performance over one-shot prompting baselines. Table 1 compares multiple strategies across different abstraction levels and base models. Compared to the vanilla baseline (e.g., simply asking "please judge whether the given answer is correct for the question"), our validation strategies can meaningfully improve validation accuracy and precision (e.g., accuracy from 21.6% to 73.2% and precision from 13.26% to 20% for Claude 3.7 Sonnet), though often at the expense of recall (see Finding #2).

A closer look at Table 1 reveals several patterns that clarify where the gains come from. First, *unanimous voting* is systematically stricter than *majority voting* and yields better performance (accuracy and precision) on difficult questions. Second, *iterated reflection* as a mid-level strategy can outperform simple *repeated sampling*, but its benefit is model-dependent (e.g., Claude benefits from *iterative reflection* while o3-mini doesn't). Third, multi-model ensembles are not automatically superior: adding weaker validators can dilute the signal of stronger ones and reduce precision (compare *Correctness*

| Model | Strategy | Accuracy (%) | Precision (%) | Recall (%) |
|---|---|---|---|---|
| Claude Sonnet 3.7 | Vanilla Prompt (Baseline) | 21.60 | 13.26 | 90.77 |
| | Correctness | 30.20 | 14.85 | 92.31 |
| | Correctness × 5 \| Majority | 29.40 | 14.53 | 90.77 |
| | Correctness × 5 \| Unanimous | 41.20 | 15.82 | 81.52 |
| | Correctness ↻ 5 \| Unanimous | 54.32 | 23.08 | 56.25 |
| | 3-Iter Pipeline | 73.20 | 20.00 | 16.00 |
| o3-mini | Vanilla Prompt (Baseline) | 24.00 | 14.29 | 96.92 |
| | Correctness | 28.60 | 15.24 | 98.46 |
| | Correctness × 5 \| Majority | 29.20 | 15.18 | 96.92 |
| | Correctness × 5 \| Unanimous | 33.00 | 15.56 | 93.85 |
| | Correctness ↻ 5 \| Unanimous | 30.00 | 15.16 | 95.38 |
| | 3-Iter Pipeline | 34.40 | 15.84 | 93.85 |
| o3 | Vanilla Prompt (Baseline) | 58.12 | 20.73 | 78.46 |
| | Correctness | 70.60 | 22.00 | 50.00 |
| | Correctness × 5 \| Majority | 73.15 | 25.87 | 56.92 |
| | Correctness × 5 \| Unanimous | 83.77 | 26.47 | 13.85 |
| | Correctness ↻ 5 \| Unanimous | 78.60 | 28.57 | 43.08 |
| | 1-Iter Pipeline | 75.40 | 24.00 | 42.00 |
| | **3-Iter Pipeline** | **81.65** | **30.99** | **34.38** |
| | 5-Iter Pipeline | 81.50 | 26.23 | 25.40 |
| Multi-model ensemble | Correctness (5 Models) \| Majority | 45.00 | 17.99 | 90.77 |
| | Correctness (5 Models) \| Unanimous | 78.60 | 25.00 | 32.31 |
| | **3-Iter Pipeline (2 Models) \| Unanimous** | **85.40** | **40.00** | **24.62** |

Table 1: **UQ-Validators metrics.** Scores are computed on 500 subsampled HLE question-answer pairs, where ground-truth is withheld during validator judgment. × and ↻ denote *repeated sampling* and *iterated reflection*, e.g. "Correctness ×3 | Majority" repeats the correctness check thrice and takes majority vote. Pipelines are the following strategies: 1-Iter = [CC⇒FLC⇒C]; 3-Iter = [(CC×3 | U)⇒(FLC×3 | U)⇒(C×3 | U)], with C = correctness, CC = cycle consistency, FLC = fact/logic check, U = unanimous vote. Multi-model ensemble uses Gemini 2.5 Pro, o3, o3-mini, o4-mini, Claude Sonnet 3.7, with pipeline ensembling using Gemini and o3. Bold marks the best UQ-Validators by precision. Owing to API-budget constraints, we use five models to produce the 500 candidate answers (a random non-overlapping subset of 100 each). See Appendix E.4 for more results.

ensemble vs. *Correctness* by o3); using cross-model *unanimous voting* restores strictness but further reduces recall and increases cost. Finally, prompt quality matters as much as scale: replacing the vanilla baseline with a structured *Correctness* prompt yields sizeable improvements across models.

Another observation is that validation strategies are (somewhat) amenable to test-time scaling (see results in Appendix E.5 and also Kim et al. (2025b); Kalra & Tang (2025)): strategies that spend more LLM calls and tokens, use more base models, and involve more sequential steps tend to perform better. The trend, however, isn't sufficiently consistent and predictable.

**Finding #2: Attaining High Precision is Difficult**

On the flip side, Table 1 also shows that the best performing UQ-Validator still has limited precision at 40% (high false positives), and there is a sharp tradeoff between precision and recall across validators of different complexity. Attaining high precision is difficult for two reasons:

1. First, to minimize distribution shift to real-world unsolved questions, we run evaluations on extremely difficult questions, and in doing so, very few questions can be correctly answered by current frontier models, thus limiting the number of true positives, and in turn, precision.

2. Second, unlike probabilistic classifiers whose precision–recall tradeoff can be smoothly adjusted via confidence thresholds, UQ-Validators operate more akin to black boxes without tunable thresholds. Making them stricter does not reliably boost precision. As shown in Table 1, the 5-iter o3 validator lowers *both* precision and recall relative to the 3-iter version as the impact on true positives is larger than false positives. This suggests that validator strictness is not analogous to confidence thresholding and that fine-grained control remains an open research challenge.

To sanity-check that UQ-Validators are useful for human reviewers, we provide discussions in Appendix E.3 on human/UQ-Validator agreement and visualize sample judgment traces in Appendix I.2.

**Finding #3: Simple Validators Show Over-Optimism and Self-Bias**

Another challenge with using LLMs for answer validation is that they often exhibit considerable self-evaluation bias, as documented in prior work (Panickssery et al., 2024; Wataoka et al., 2024; Ye et al., 2024; Xu et al., 2024; Goel et al., 2025). When naively applying LLMs in our setting, we observe similar bias by all frontier models in the form of *over-optimism* for evaluating self and

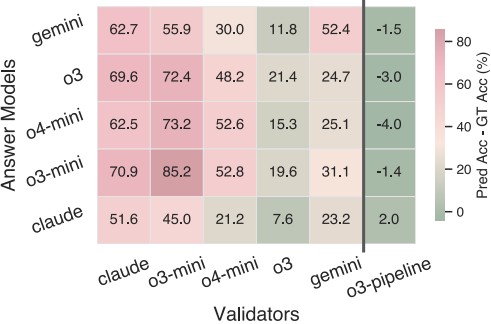 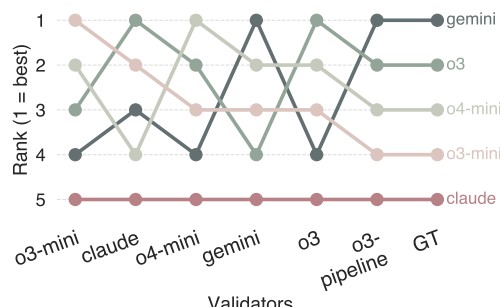

Figure 5: **LLM validators overrate self and sibling answers.** Heatmap shows evaluation bias, measured in (predicted − ground-truth (GT)) answer accuracy, for each validator (columns) and each answer model (rows); red means larger overestimation. Our o3 pipeline validator (rightmost column) drastically reduces this bias.

Figure 6: **Model ranking is unstable across validator performance.** Each line traces the rank (1 = best) that six validators of varying strength assign to an answer model. Frequent crossings show that the relative ordering of models changes unpredictably, though the strongest validator (o3 pipeline) agrees with ground-truth (GT).

sibling models (those from the same model developer), where the predicted model performance is drastically higher than actual model performance, as shown in Figure 5. Gemini significantly favors itself compared to other models; Claude exhibits over-optimism across *all* answer models (not just itself); and OpenAI o-series models overrate all other o-series models. Increasing model capability (o3-mini → o3) reduces but does not eliminate this bias.

**Finding #4: Compound Validator Strategies Mitigate Over-Optimism and Self-Bias**

We next observe that a compound validator can significantly reduce self-bias and over-optimism in answer validation. Figure 5 shows that the 3-iter o3 pipeline (Figure S2) largely removes over-optimism across all models, and in particular, removes the preferential treatment toward models from the same family (no significant bias on o-series over other models). This suggests that scaling validation strategies improves not only performance (finding #1) but also fairness across models.

**Finding #5: Model Rankings Are Unstable Across Validator Performance**

While weak validators may be unreliable, one may assume that they should still infer the correct *ranking* of answer model performance even if they misjudge the absolute answer model accuracy. We test this assumption by ranking five answer models with six validators of varying strength (from a weak validator model like o3-mini to a strong 3-iter o3 validator pipeline).

As shown in Figure 6, the model rankings shift erratically: every answer model (Gemini, o3, o4-mini, o3-mini) occupies first place under at least one validator, yet may drop multiple positions under others. These swings show no systematic relation to validator performance—although the ranking converges to the ground-truth at the strongest o3 pipeline validator. Because validators have no ground-truths at test time when applied to unsolved questions (as opposed to the experiments where we use HLE as the surrogate dataset with ground-truths), this ranking instability cautions against the reliance on such oracle-free validators to build model leaderboards. This is also an important motivation behind UQ-Platform (§4): UQ-Validators alone cannot produce automated model rankings and community-driven human verification is necessary. We defer addition findings to Appendix E.5.

## 4 UQ-PLATFORM: CONTINUOUS, COMMUNITY-BASED MODEL EVALUATION

Unsolved questions need human-in-the-loop model evaluation. To this end, we develop UQ-Platform to continue where UQ-Validators leave off: domain experts can rate and verify model responses (that passed UQ validation), comment on question quality, and contribute to the resolution of unsolved questions. UQ-Platform is central to our new evaluation paradigm: model evaluation on unsolved questions is no longer static but a continuous, community-based effort, necessitating an open platform.

UQ-Platform is partly a simplified AI-native mirror of Stack Exchange. It is a central hub to view AI answers to open questions with expert assessments and transparency (e.g., prompts for reproducibility) and track model performance. Another property of UQ-Platform is its compounding evaluation quality: UQ-validation lowers marginal efforts of human verification, and as models improve and we collect

| Answer Model | UQ-Validator Pass Rate | | Human Pass Rate (2025-09-24)* |
|---|---|---|---|
| | # Passed | % | |
| O3-PRO | 75 / 500 | 15.0% | 4 / 46 |
| ↪ on UQ diamond subset | 3 / 25 | 4.0% | 0 / 2 |
| GEMINI 2.5 PRO | 25 / 500 | 5.0% | 3 / 10 |
| O4-MINI (HIGH) | 25 / 500 | 5.0% | 2 / 14 |
| O3 | 44 / 500 | 8.8% | 1 / 25 |
| ↪ on UQ diamond subset | 1 / 25 | 2.0% | 0 / 1 |
| DEEPSEEK-R1-0528 | 11 / 500 | 2.2% | 1 / 5 |
| CLAUDE OPUS 4 | 7 / 500 | 1.4% | 0 / 3 |
| CLAUDE SONNET 3.7 (16K) | 6 / 500 | 1.2% | 0 / 3 |
| GPT-4O | 0 / 500 | 0.0% | 0 / 0 |
| Total unique questions | 144 / 500 | 28.8% | 10 / 91 |

Table 2: **Assessing various models on the full UQ-Dataset.** We report pass rates on the 3-iter pipeline UQ-Validator (Table 1) and the number of answers that cleared initial human verification. Without ground-truth answers, UQ-Validator pass rates are indicative, but not conclusive, of actual performance. (*): Human pass rates have smaller denominators due to limited expert availability (only 91/144 questions passing the UQ-Validator are verified). The selection of human-rated answers is biased toward wrong answers, as it is easier to prove an answer wrong than correct.

human feedback, UQ-Validators can improve continuously, in turn increasing the share of solvable questions. This makes UQ-Platform more useful to reviewers and answer-seekers over time. Due to space limitations, see Appendix F for a summary of key features, discussions on user incentives, and screenshots/visualizations of the platform.

## 5 PARTIAL MODEL EVALUATION

We test frontier models on the UQ-Dataset. We first report model pass rates on our 3-iter pipeline UQ-Validator (Figure S2) and then solicit human experts to verify passed candidate answers.

**UQ-Validator pass rates.** Table 2 shows the difficulty of UQ-Dataset: all models have a low UQ-Validator pass rate. The pass rate model ranking mirrors recent benchmarks, with reasoning models like o3 and Gemini 2.5 Pro outperforming Claude 3.7 Sonnet and non-reasoning models like GPT-4o.

**Human verification.** We pool questions that passed our UQ-Validator to solicit human verdict (domain experts and/or original question posters) on the candidate answers. These questions are very challenging and span diverse subjects. It is beyond our expertise to accurately verify all candidate solutions; we instead report partial verification results as noted with asterisk (*) and date in Table 2.

Within the human-verified subset (91 questions out of the 144 that passed UQ-validation), most models produce wrong solutions. A common failure mode is the model citing references that do not exist, which our UQ-Validator failed to catch (discussed in Appendix C). A total of 10 questions passed our secondary human validation: 6 from math, 1 from physics, 1 from stackoverflow, 1 from stats, and 1 from retrocomputing. O3-PRO stands out with meaningful answers to at least four questions that were accepted by human reviewers—breaking the initial streak of zero verified solutions during the early stages of this project. On the UQ diamond subset, we observe 4 answers approved by UQ-Validator, though none of the 3 answers that were human-verified were correct.

We visualize some human-verified answers in Appendix I.3 (answers verified as *incorrect*) and Appendix I.4 (answers verified as *correct*). All model candidate answers are on UQ-Platform for community-based verification, which will inform updates to human verification results.

## 6 CONCLUDING REMARKS

UQ explores a radically new paradigm for AI evaluations: instead of building increasingly harder exams that are decreasingly realistic, we shift the focus to evaluating unsolved problems that are often hard and realistic by design. UQ has three standalone components: UQ-Dataset (§2) provides model inputs, UQ-Validators (§3) assess model outputs, and UQ-Platform (§4) facilitates community-based evaluation. As models improve and questions get solved, we seek to update UQ-Dataset with new, harder questions. We also hope to explore generator-validator interaction for UQ-Validators in future work. See Appendices B and C for related work and limitations. In sum, UQ serves as a foundation for future work on scaling model capabilities in oracle-free, hard-to-verify domains.

ETHICS STATEMENT

This work introduces a new paradigm for evaluating language models on open, unsolved questions. All questions in the current version of the dataset are sourced from publicly available Stack Exchange content, which is licensed under CC-BY-SA 4.0 (see Appendix H for more discussions). We respect this license and preserve full attribution in the dataset release.

To verify candidate answers from top-performing models, we occasionally contacted original question askers on Stack Exchange via the public contact information they themselves provided on Stack Exchange. These messages simply asked whether a proposed model answer correctly addressed their question, and responses were treated only as correctness judgments of the technical content, not as data about the individuals themselves. The activity does not involve sensitive information or draw conclusions about the question askers, and thus does not constitute human subjects research. Nevertheless, we consulted our institution's IRB office, who confirmed that such activity would not require IRB review because the research is about language models rather than human participants.

Our project raises minimal risks. The dataset contains no private or identifiable information beyond what users have already made public under an open license. The **UQ**-Platform is open and designed to improve transparency, reproducibility, and community engagement in model evaluation. We do not anticipate negative societal or environmental consequences.

REPRODUCIBILITY STATEMENT

The full codebase, covering **UQ**-Dataset crawling and filtering pipelines and **UQ**-Validators scaffolds, is included in the supplementary materials. The current **UQ**-Dataset is also provided alongside details of the collection pipeline (Section 2 and Appendix D). Prompts for filtering and validation are listed in Appendix I, and model versions and hyperparameters are detailed in Appendix G. **UQ**-Validators (Section 3) and their experimental comparisons (Appendix E) are described in full, with additional reasoning traces and human-verification examples in Appendix I. The **UQ**-Platform (Section 4) provides a live mirror of our evaluation setting, hosting questions, model answers, validation outcomes, and full provenance for reproducibility. Together, these resources allow researchers to replicate and extend our results.

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

APPENDIX

## A    LLM USAGE FOR PAPER WRITING

We used LLMs as general-purpose writing assistants for limited purposes. Specifically, we used LLMs to help draft a small set of sentences and non-critical paragraphs, and to polish language in sections initially written by human authors. In all cases, human authors reviewed, edited, and, when necessary, rewrote the LLM drafts. LLM usage was overall minimal, and the core research design, experiments, analysis, and writing remain entirely the responsibility of the human authors.

## B    RELATED WORK

**Exam-based Benchmarks.** Early benchmarks on language models tested narrow skills with questions using human-annotated answers—reading comprehension (e.g., SQuAD (Rajpurkar et al., 2016)), natural language inference (e.g., GLUE (Wang et al., 2018), SuperGLUE (Wang et al., 2019)), and commonsense (e.g., HellaSwag (Zellers et al., 2019), PIQA (Bisk et al., 2020)). Subsequent exams kept the format but broadened scope or difficulty: MMLU (Hendrycks et al., 2020) and its variants (Wang et al., 2024b; Gema et al., 2024) for general knowledge; MATH (Hendrycks et al., 2021b) and its variants (Huang et al., 2025a; Cobbe et al., 2021; Zhang et al., 2024) for math; HumanEval (Chen et al., 2021), APPS (Hendrycks et al., 2021a), BigCodeBench (Zhuo et al., 2025) for code; LiveBench (White et al., 2024), LiveCodeBench (Jain et al., 2024) for contamination-controlled tests; AGIEval (Zhong et al., 2023), HELM (Liang et al., 2022) for broad coverage. As frontier models nearly saturate these benchmarks, new suites such as FrontierMath (Glazer et al., 2024), Humanity's Last Exam (Phan et al., 2025), ARC-AGI (Chollet et al., 2024), GPQA (Rein et al., 2024; Team et al., 2025), BrowseComp (Wei et al., 2025), and contest problems such as AIME (of America, 2024) pivot to *expert-crafted, artificially difficult* questions. These questions expose edge-case failures but diverge away from how real-world problems arise—they are not posed by a human with an information need, and the test maker already knows the answers.

**Realistic Benchmarks.** In contrast, *realistic benchmarks* begin with real user interactions and derive an evaluation protocol. Natural Questions (Kwiatkowski et al., 2019) uses Google queries; WildBench (Lin et al., 2024) samples prompts from public chatbot logs. Preference-based evaluation (e.g., Chatbot Arena (Chiang et al., 2024)) relies on crowd votes to score open-ended responses. SWE-bench (Jimenez et al., 2024; Yang et al., 2024) scores GitHub patch generation, $\tau$-bench (Yao et al., 2024) tests tool-using agents, and the recent terminal-bench (Stanford & Laude Collaborators, 2025) measures problem solving in terminal settings. Although these settings mirror everyday use, they tend to saturate quickly: retrieval-augmented models solve most search queries, preference-based evaluations based on crowd-sourced prompts skew toward simple inputs, and terminal-bench pass rates already reaches 50% within months of its release. Real-world interaction with these benchmarks also mean they are vulnerable to adversarial manipulation; see, for example, Huang et al. (2025b) and Singh et al. (2025).

**LLM-as-a-Judge.** Recent work also explores using capable models to grade other models' outputs when exact-match metrics (multiple-choice, BLEU, ROUGE) fall short. MT-bench and Chatbot Arena showed that GPT-4 can reach roughly 80 % human agreement, but the judge may exhibit position/verbosity biases (Zheng et al., 2023). Follow-ups extend the idea: AlpacaFarm (Dubois et al., 2023) uses LLM judges to simulate feedback for RLHF, LIMA (Zhou et al., 2023) explores mixed LLM and human ratings, Prometheus (Kim et al., 2024a;b) adds rubric structure, FLASK (Lee et al., 2023) ensembles judges for robustness, and PandaLM (Wang et al., 2023) offers an open-source preference-tuned judge. New directions include chain-of-verification (Dhuliawala et al., 2023), multi-turn judging (Kim et al., 2025a), and domain-specific judges for code (Jain et al., 2023) and math (Lightman et al., 2023; Wang et al., 2024a). Most studies score tasks with *known* answers; we instead deploy LLM *validators* (§3) to triage responses to unsolved questions, where ground truth is absent but quality can still be judged against clear criteria.

## C    DISCUSSIONS & LIMITATIONS

In this section, we provide discussion on each **UQ** component in terms of their design choices, potential limitations, and potential future work.

## C.1 UQ-DATASET

**Apparent rather than intrinsic unsolvedness.** Certain questions may be unsolved due to lack of attention rather than their inherent difficulty, and it is possible that frontier systems optimized for web browsing could quickly resolve a subset of the UQ-Dataset. To mitigate this, we try to identify such questions during answer validation with UQ-Validators and manual inspection, as well as filtering for high engagement questions which receives high moderation effort on Stack Exchange and are, in turn, more likely to be truly unsolved.

**Limited annotation budget.** Hard problems often need multiple rounds of human reviews, but our human-review budget is modest. Additional review may reduce reliance on engagement signals.

**Source bias and STEM skew.** The current version of the UQ-Dataset is sourced entirely from Stack Exchange, which favors certain formats and domains (e.g., mathematics over astronomy). While we source questions from 80+ sites on Stack Exchange, the final questions surviving the filters (Section 2.1) may concentrate in STEM topics, reflecting both Stack Exchange usage and our filters for question quality. Some questions may also be difficult only because they once required extensive web search—an obstacle frontier models can now overcome. We do not claim that the UQ-Dataset is broadly representative of unsolved questions in the wild, particularly at research level (e.g., open theoretical computer science problems such as List of Open Problems in Sublinear Algorithms (2025)).

**Should questions for humans be used to measure progress for AI?** Recent papers such as Sühr et al. (2025); Salaudeen et al. (2025) caution the use of human-centered problems for model evaluation on the grounds of measurement validity. While the UQ-Dataset consists of hard questions posed by and for humans, they arise organically and solving them yields direct real-world value; we therefore view progress on the UQ-Dataset as a distinct, complementary objective to benchmarking model performance.

## C.2 UQ-VALIDATORS

**Reliance on surrogate data.** Budget constraints on expert grading necessitate our use of smaller dev sets and external datasets such as Humanity's Last Exam (Phan et al., 2025) for evaluating the UQ-Validators. While surrogate data do provide useful signal (Section 3.1), they may not perfectly match the distribution of the UQ-Dataset.

**Open-ended nature of (oracle-free) validator design.** Designing and evaluating answer verifier, especially in the absence of ground-truth signal, is an active research topic (e.g., Zhou et al. (2025)). While we extensively experimented with various validation strategies (Section 3), the broader design space remains underexplored which we may pursue in future work.

**Cost and latency constraints.** Experiments show that higher-capacity models and ensembles boost validator accuracy (Table 1), yet the required inference volume increases API costs. We have not benchmarked some systems such as Grok 4 and o3-deep-research due to their substantially longer inference times and higher cost.

**Limited reference verification.** For topics where the credibility of an answer depends on accurate citations (e.g., history), UQ-Validator may fail to discern hallucinated citations since we leverage reasoning models as opposed to models that specialize in web browsing (e.g., deep research agents (OpenAI, 2025)).

## C.3 UQ-PLATFORM

**Community-engagement bias.** Early participants are more likely to be LLM hobbyists and researchers than a wider pool of domain experts that the UQ-Platform ultimately seeks. An important benefit of the UQ-Platform is that it serves as an "AI-native" mirror of Stack Exchange, where generative AI answers are currently heavily censored (see Appendix H.2). The UQ-Platform offers a convenient venue for accessing (and verifying) AI-generated solutions.

**Sparse evaluation signal.** At launch, most models solve few if any questions, so UQ offers little ranking power until solutions accumulate.

**Moderation and abuse prevention.** Open contribution to UQ-Platform also means susceptibility to adversarial engagement (e.g., Huang et al. (2025b); Singh et al. (2025)); we thus need continuous moderation.

# D ADDITIONAL DETAILS ON UQ-DATASET

This section provides additional details on the UQ-Dataset. For question samples, see Appendix I.

## D.1 LIST OF SOURCE STACK EXCHANGE SITES

Recall from Section 2.1 that the dataset creation first involves a raw crawl from Stack Exchange. We initially crawled unanswered questions from **80** distinct Stack Exchange sites. After the entire filtering pipeline, **35 / 80** sites (43.75%) remained in the final UQ-Dataset (counting Stack Overflow and its multi-lingual sites such as ja.stackoverflow and ru.stackoverflow altogether as one site).

Table S1 lists every site; check-mark (✓) indicates that at least one question from that site survives the entire filtering pipeline.

Table S1: All 80 Stack Exchange communities in the crawl (✓ = retained, – = fully filtered, ).

| Community | Community | Community |
|---|---|---|
| 3D Printing – | Economics ✓ | Poker – |
| Academia – | Electrical Engineering – | Proof Assistants ✓ |
| Anime & Manga – | Engineering – | Psychology & Neuroscience – |
| Artificial Intelligence ✓ | es.stackoverflow – | pt.stackoverflow – |
| Ask Patents – | Ethereum – | Puzzling ✓ |
| Astronomy – | Expatriates – | Quantitative Finance ✓ |
| Aviation – | Genealogy & Family History – | Quantum Computing ✓ |
| Bioacoustics ✓ | History ✓ | Retrocomputing ✓ |
| Bioinformatics – | History of Science & Mathematics ✓ | Reverse Engineering – |
| Biology ✓ | Information Security ✓ | Robotics – |
| Bitcoin – | ja.stackoverflow ✓ | Role-playing Games ✓ |
| Board & Card Games – | Law – | ru.stackoverflow ✓ |
| Cardano – | Linguistics ✓ | Science Fiction & Fantasy ✓ |
| Chemistry ✓ | Mathematica ✓ | Signal Processing ✓ |
| Chess – | Mathematics ✓ | Software Engineering – |
| Code Golf ✓ | MathOverflow ✓ | Software Quality Assurance & Testing – |
| Code Review – | Matter Modeling ✓ | Sound Design – |
| Computational Science ✓ | Medical Sciences ✓ | Space Exploration ✓ |
| Computer Graphics – | Monero – | Sports – |
| Computer Science ✓ | Motor Vehicle Maintenance & Repair – | Stack Overflow ✓ |
| Cross Validated ✓ | Movies & TV – | Substrate and Polkadot – |
| Cryptography ✓ | Music: Practice & Theory – | TeX – LaTeX ✓ |
| Data Science – | Mythology & Folklore ✓ | Tezos – |
| DevOps – | Network Engineering – | Theoretical Computer Science ✓ |
| Drones and Model Aircraft – | Open Source – | Unix & Linux ✓ |
| Earth Science – | Operations Research ✓ | Vi and Vim – |
| Ebooks – | Physics ✓ | |
| | | **Retained after LLM-based filters: 35 / 80** |

## D.2 NATURE OF QUESTIONS

As questions progress through the filtering pipeline, their difficulty and quality gradually increase. In particular, LLM-based filtering substantially increases question difficulty while tightening the quality metrics (approachability, well-definedness, and objectivity). Figure S1 shows that, as judged by o4-mini, the averaged expert solvability dropped from 77.8% to 32.2% (i.e., the question appears harder), and answer correctness by GPT-4o as the answer model drops from 51.2% to 14.1% (i.e., the questions actually became harder by considering the answers). On the other hand, the fractions of questions meeting the binary quality criteria rise to 100%, because any question failing these criteria is discarded by the LLM-based filter.

## D.3 ADDITIONAL DETAILS ON RULE-BASED FILTERING

In Section 2.1, we described the dataset creation pipeline, and the first stage is the rule-based filtering of the questions crawled directly from Stack Exchange. Below we provide the full list of rules:

- *Age:* Questions must be $\geq 2$ years old. This excludes fresh questions that may be answered soon and allows sufficient time to attract attention.
- *Views:* Questions must have $\geq$ 200-2000 views (site-dependent). This filters low-interest questions.

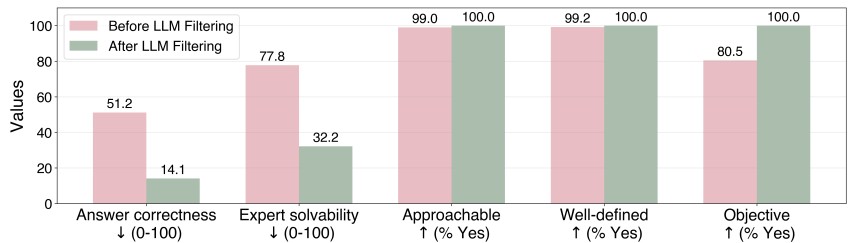

Figure S1: **Effects of LLM-based questions filters.** We compare question difficulty metrics (i.e., attempted answer correctness and expert solvability) and quality metrics (i.e., approachability, well-definedness, and objectivity) before and after applying the LLM-based filter. Arrows ($\uparrow$, $\downarrow$) indicate desired direction of improvement. These LLM-based filters reduce 33,916 candidate questions to 7,685 (or 22.7%). Quality metrics saturate at 100% as we discard questions failing these metrics.

- *Votes:* Questions must have $\geq$ 5-75 net upvotes (site-dependent) to exclude low-engagement ones.
- *Views-to-Votes Ratio:* The views-to-votes must be $\leq$ 5000 to exclude questions that attract views but not engagement. Such questions tend to be generic or poorly-specified.
- *Top-ranking:* Questions must be in the top 10% of unanswered questions by votes per site. This rule primarily triggers on high-volume sites like Mathematics with many eligible questions to additionally filter for quality.
- *No Answers:* Questions must have zero answers (as opposed to having candidate answers not accepted by the original poster). Questions with high engagement but no answers after a timespan are strong candidates for being truly unsolved. This increases the likelihood that the questions are unsolved.
- *No "Why":* We also remove questions with "why" in the title, as they can be open-ended or subjective, complicating downstream answer validation.
- *No Images:* The question body must not contain images as we focus on language models.
- *No unrelated tags:* We also exclude questions tagged with off-topic keywords like "homework", "advice", "policy", or "recommendation".

Note that these rules are not exhaustive; they aim to heuristically trim the vast pool (millions) of unanswered questions. We then pass filtered questions to an LLM judge and expert review.

### D.4 Additional Details on Human Filtering

Recall from Section 2.1 that the final stage of dataset creation involves human review.

For several high-volume sites, we simply select the top-$k$ unanswered questions based on net upvotes. The rationale is that these high-volume sites are already significantly moderated, and the top unanswered questions are very likely to possess the desirable properties we want for an unsolved question (the same set we used to define the LLM-based filter).

- **MathOverflow**: top 200
- **Mathematics**: top 90, plus 18 manually selected questions (by manual review), for a total of 108
- **Theoretical Computer Science**: top 40
- **Science Fiction & Fantasy**: top 35
- **Cryptography**: top 5
- **Mathematica**: votes $\geq$ 10 (8 questions)
- **Physics**: votes $\geq$ 10 (6 questions)
- **Stack Overflow**: votes $\geq$ 10 (18 questions)
- **Computer Science**: votes $\geq$ 10 (12 questions)

For smaller or domain-specific communities, we manually select by jointly considering content and engagement signals such as vote counts:

- **History**: 5 manually selected
- **Linguistics**: 2 manually selected

- **Retrocomputing**: top 4 by votes and review
- **Quantum Computing**: top 4 by votes and review

For the remaining sites, questions are manually selected. These include sites such as: Matter Modeling, Biology, Role-playing Games, 3D Printing, Bioacoustics, Code Golf, TeX – LaTeX, Artificial Intelligence, Economics, Signal Processing, Puzzling, Information Security, Computational Science, Medical Sciences, Mythology & Folklore, Quantitative Finance, Space Exploration, Operations Research, History of Science & Mathematics, and Chemistry.

This final round ensures the inclusion of diverse and high-quality questions that might not be captured solely by automated filtering, especially in lower-volume or specialized domains.

### D.5 ADDITIONAL DETAILS ON HELD-OUT DEVELOPMENT SET

We source 30 questions from Stack Exchange *with* ground-truth answers (e.g., accepted on Stack Exchange) with the same set of criteria as the rest of the UQ-Dataset. This dev set helps inform the design of automated answer validation strategies (discussed in Section 3). Although these questions have ground-truth answers, we still let human experts to check the correctness of the models' answers.

### D.6 ADDITIONAL DATASET STATISTICS

This section augments the filtering statistics provided in Section 2.2:

- Table S2 shows high-level question filtering statistics.
- Table S3 augments Table S2 and Section 2.2 by showing the per-stage filtering statistics for each of five high-level domains categorized by Stack Exchange (Science, Technology, Life & Arts, Culture & Recreation, and Business).
- Table S4 breaks down the diamond subset of the UQ-Dataset to site-level statistics. We select the questions based on engagement signals from Stack Exchange. For example, questions must have $\geq 2,000$ views and $\geq 75$ net upvotes for Mathematics, or $\geq 50$ for MathOverflow. We also include additional human reviews for the diamond set to catch exceptional cases not captured by filters. We kept the subset small given the scarcity of such high-engagement questions and the cost of human review.
- Table S5 breaks down the full UQ-Dataset to site-level statistics.

| Stage | # Questions | Retained (%) of Original | Retained (%) of Previous |
|---|---|---|---|
| Raw question pool | 3,000,000 | 100% | - |
| Rule-based filtering | 33,916 | 1.13% | 1.13% |
| LLM-based filtering | 7,685 | 0.26% | 22.66% |
| Manual filtering | 500 | 0.02% | 6.51% |

Table S2: **Question pool size per filtering stage.** See also Figure 2 and Figure 3.

| Stage | Category | # Questions | Percentage(%) |
|---|---|---|---|
| Rule-based filtering | Technology | 8,994 | 26.5 |
| | Science | 21,344 | 63.0 |
| | Culture & Recreation | 394 | 1.2 |
| | Life & Arts | 2,922 | 8.6 |
| | Business | 245 | 0.7 |
| LLM-based filtering | Technology | 152 | 2.0 |
| | Science | 6,167 | 80.3 |
| | Culture & Recreation | 27 | 0.4 |
| | Life & Arts | 1,330 | 17.3 |
| | Business | 8 | 0.1 |
| Human-reviewed final | Technology | 52 | 10.4 |
| | Science | 395 | 78.8 |
| | Culture & Recreation | 16 | 3.2 |
| | Life & Arts | 35 | 7.0 |
| | Business | 2 | 0.4 |

Table S3: **Category pool size per filtering stage.** This table augments Table S2 with the category specific question counts for each of the five high-level domains categorized by Stack Exchange.

| Category | Site | # Questions |
|---|---|---|
| Science | Math Overflow | 6 |
| | Mathematics | 9 |
| | Theoretical Computer Science | 7 |
| | Physics | 1 |
| | **Subtotal** | **23** |
| Culture & Recreation | Puzzling | 1 |
| Life & Arts | Science Fiction & Fantasy | 1 |
| **Total** | - | **25** |

Table S4: **UQ-Dataset Diamond Subset Composition.** Breakdown of the 25-question diamond subset by Stack Exchange site and high-level category.

| Category | Site | # Questions |
|---|---|---|
| Technology | Stack Overflow | 21 |
| | Mathematica | 8 |
| | Cryptography | 5 |
| | Retrocomputing | 4 |
| | Quantum Computing | 4 |
| | Space Exploration | 3 |
| | Unix & Linux | 2 |
| | TeX - LaTeX | 2 |
| | Code Golf | 1 |
| | Signal Processing | 1 |
| | Information Security | 1 |
| | **Subtotal** | **52** |
| Science | Math Overflow | 200 |
| | Mathematics | 108 |
| | Theoretical Computer Science | 41 |
| | Computer Science | 12 |
| | Cross Validated | 9 |
| | Physics | 6 |
| | Chemistry | 4 |
| | History of Science and Mathematics | 3 |
| | Linguistics | 2 |
| | Proof Assistants | 2 |
| | Artificial Intelligence | 1 |
| | Economics | 1 |
| | Bioacoustics | 1 |
| | Biology | 1 |
| | Medical Sciences | 1 |
| | Matter Modeling | 1 |
| | Operations Research | 1 |
| | Computational Science | 1 |
| | **Subtotal** | **395** |
| Culture & Recreation | Puzzling | 8 |
| | History | 5 |
| | Mythology & Folklore | 2 |
| | Role-playing Games | 1 |
| | **Subtotal** | **16** |
| Life & Arts | Science Fiction & Fantasy | 35 |
| Business | Quantitative Finance | 2 |
| **Total** | - | **500** |

Table S5: **Full UQ-Dataset Composition.** Breakdown of question counts by Stack Exchange site, grouped by high-level category, in the final UQ-Dataset.

### D.7 DATASET UPDATES AND VERSIONING

To ensure clarity for future work, we will assign the **UQ**-Dataset an explicit *version identifier*. Versioning provides several benefits:

- It ensures that the results can be unambiguously tied to a specific dataset snapshot, avoiding inconsistencies across experiments.
- It facilitates tracking of changes over time, including additions and removals of questions.

A potential criterion for issuing a new dataset version is when at least 20% of the **UQ**-Dataset is considered solved, as manually verified by qualified domain experts. If a version update occurs, it will be reflected consistently across all public release channels, including the **UQ**-Platform, Hugging Face, GitHub, as well as this paper. At the time of this writing, we have not planned an updated version of the **UQ**-Dataset.

# E ADDITIONAL DETAILS ON UQ-VALIDATORS

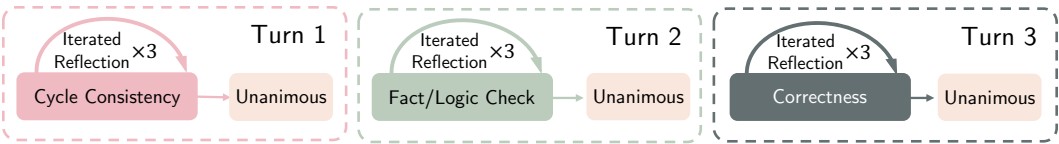

Figure S2: Illustration of the default, performant UQ-Validator pipeline used in experiments.

## E.1 ADDITIONAL DISCUSSIONS ON DOMAIN-SPECIFIC UQ-VALIDATORS

In domains where the solution space is formally structured, one can leverage domain-specific invariants or heuristics to build much stronger oracle-free validators than the general-purpose strategies designed for the UQ-Dataset. For instance, in competition mathematics (e.g., IMO problems), a candidate proof can be type-checked in Lean/Coq and then subjected to tactic-level consistency checks; in programming challenges, one can execute candidate code against adversarial test suites that test edge cases; and in chemistry or physics, validators can automatically enforce conservation laws or dimensional consistency.

By hard-coding such domain rules, validators shift from heuristic plausibility tests toward near-deterministic correctness filters, substantially boosting precision at the cost of narrow applicability. Designing these (oracle-free) validation strategies therefore often reduces to identifying the domain's formal specification and translating it into machine-checkable assertions.

When designing UQ-Validators, we intentionally limit the use of domain-specific rules and instead favor broadly applicable checks that apply to the diverse domains that the UQ-Dataset spans. Specialized validators remain complementary and we leave the exploration of richer domain-tailored strategies to future work.

## E.2 ADDITIONAL RESULTS ON GENERATOR-VALIDATOR GAP

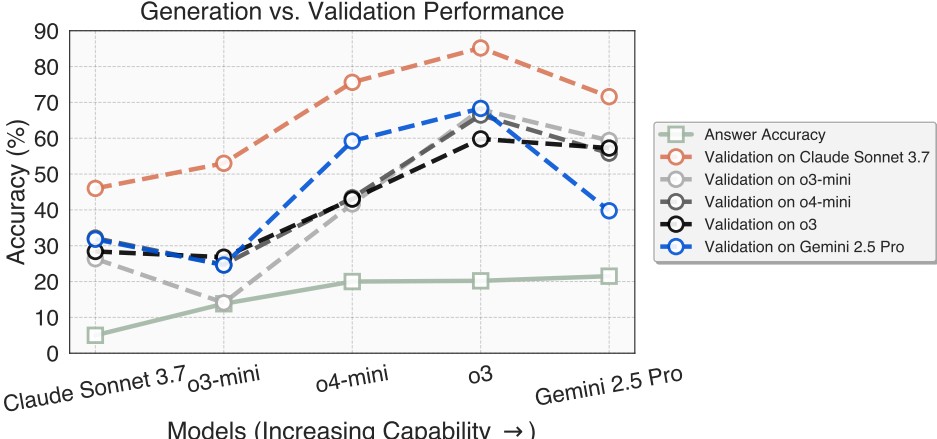

Figure S3: **Generator-validator gap** (extended version of Figure 4a). We observe that a model's ability to validate candidate answers to hard questions grows faster than its ability to generate them. Red plot means each model's answer accuracy; each green plot means the model's validation accuracy on answers generated by another model.

Figure S3 augments Figure 4a in Section 3.1 (motivations of UQ-Validators) by including two additional models. While the generator–validator gap still widens with model capability, the trend is noisier here. Interpret this pattern alongside the findings in Appendix E.5, which shows that a stronger answer generator is not necessarily a stronger validator across model families (in particular, o3 is a stronger validator than Gemini 2.5 Pro).

### E.3 ADDITIONAL RESULTS ON HUMAN/UQ-VALIDATOR AGREEMENT

In Section 3.3 (Finding #2), we reported that it is hard to attain high precision with UQ-Validators. To confirm that the best resulting UQ-Validator remains useful for human reviewers—in particular, for ruling out incorrect answers—we ask human reviewers to rate whether its *judgment reasoning traces* make logically valid arguments over 25 validation questions (20 math and 5 non-STEM) which (mostly) received incorrect answers. Table S6 shows high human-validator agreement and judging trace accuracy, suggesting its utility for human reviewers.

To account for label bias (most candidate answers being wrong), Table S6 also shows Cohen's kappa coefficient (a statistic that measures inter-rater reliability). In cases where all answers are wrong (as marked by humans), the Cohen's $\kappa$ becomes undefined (denoted as - in Table S6). For Gemini 2.5 Pro as the answer model which produced (only) one correct answer, the coefficient is 0.468, which is considered "moderate" agreement (Landis & Koch, 1977). In future work, and as models improve to produce more correct answers, we expect to obtain more meaningful measurements of human/UQ-Validator agreement.

| Metric | Answer Models | | | |
|---|---|---|---|---|
| | o3 | Claude Sonnet 3.7 | Gemini 2.5 Pro | GPT-4o |
| % answers passed UQ-Validator | 0% | 0% | 12% | 0% |
| % answers passed human reviewers (i.e., GT accuracy) | 0% | 0% | 4% | 0% |
| Human/UQ-Validator judgment agreement | 100% | 100% | 92% | 100% |
| Human-rated accuracy of UQ-Validator reasoning trace | 96% | 96% | 76% | 100% |
| Human/UQ-Validator Cohen's $\kappa$ | - | - | 0.468 | - |

Table S6: Cohen's $\kappa$ of Human/UQ-Validator agreement.

### E.4 ADDITIONAL RESULTS ON UQ-VALIDATORS PERFORMANCE

Table S7 augments Table 1 in the Section 3 by listing all UQ-Validator strategies we have explored.

Observe that:

- **Validator strength scales with model quality.** The accuracy of the baseline "Correctness" strategy climbs from $\approx 30\%$ on Claude Sonnet 3.7 to $\approx 71\%$ on o3, confirming that stronger models tend to be stronger one-shot validators (with the caveat mentioned in Figure S4).
- **Stricter voting rules (majority $\rightarrow$ unanimous) trade recall for precision.** Switching from majority to unanimous voting raises precision by $\approx 2$–6 pp across models, but recall can fall by 20–40 pp.
- **Sequential pipelines boost precision but slash recall.**
  - Claude Sonnet 3.7: 3-Iter pipeline raises accuracy from $30.2\% \rightarrow 73.2\%$ and precision from $14.9\% \rightarrow 20.0\%$, yet recall drops to $16\%$.
  - o3: 3-Iter pipeline achieves the best single-model trade-off ($81.7\%$ accuracy, $31.0\%$ precision, $34.4\%$ recall).
- **Model ensembling is most effective but expensive overall.** A two-model, 3-Iter unanimous pipeline reaches the highest accuracy ($85.4\%$) and precision ($40.0\%$), albeit with lower recall ($24.6\%$). Majority voting over 3–5 models maintains high recall ($\sim 80$–91%) but at the expense of precision.

The main takeaway from the table is that tighter consensus mechanisms and multi-turn pipelines make the validation stricter and convert recall into precision. However, the tradeoff is hard to control, and the optimal point depends on downstream tolerance for false positives versus false negatives as well as costs for model inference. Unless otherwise stated, we use the o3 3-Iter pipeline as our main UQ-Validator in our experiments.

| Model | Strategy | Accuracy (%) | Precision (%) | Recall (%) |
|---|---|---|---|---|
| Claude Sonnet 3.7 | Vanilla Prompt (Baseline) | 21.60 | 13.26 | 90.77 |
| | Correctness | 30.20 | 14.85 | 92.31 |
| | Correctness × 3 \| Majority | 26.80 | 13.73 | 87.69 |
| | Correctness × 3 \| Unanimous | 35.20 | 14.52 | 81.54 |
| | Correctness ↻ 3 \| Majority | 34.20 | 14.71 | 84.86 |
| | Correctness ↻ 3 \| Unanimous | 49.60 | 16.00 | 68.00 |
| | Correctness × 5 \| Majority | 29.40 | 14.53 | 90.77 |
| | Correctness × 5 \| Unanimous | 41.20 | 15.82 | 81.52 |
| | Correctness ↻ 5 \| Majority | 44.44 | 22.64 | 75.00 |
| | Correctness ↻ 5 \| Unanimous | 54.32 | 23.08 | 56.25 |
| | 1-Iter Pipeline | 33.60 | 14.78 | 86.15 |
| | 3-Iter Pipeline | 73.20 | 20.00 | 16.00 |
| o3-mini | Vanilla Prompt (Baseline) | 24.00 | 14.29 | 96.92 |
| | Correctness | 28.60 | 15.24 | 98.46 |
| | Correctness × 3 \| Majority | 28.60 | 15.07 | 96.92 |
| | Correctness × 3 \| Unanimous | 32.20 | 15.58 | 95.38 |
| | Correctness ↻ 3 \| Majority | 28.86 | 15.14 | 96.92 |
| | Correctness ↻ 3 \| Unanimous | 29.26 | 15.22 | 96.92 |
| | Correctness × 5 \| Majority | 29.20 | 15.18 | 96.92 |
| | Correctness × 5 \| Unanimous | 33.00 | 15.56 | 93.85 |
| | Correctness ↻ 5 \| Majority | 29.40 | 15.05 | 95.38 |
| | Correctness ↻ 5 \| Unanimous | 30.00 | 15.16 | 95.38 |
| | 1-Iter Pipeline | 35.34 | 16.09 | 93.85 |
| | 3-Iter Pipeline | 34.40 | 15.84 | 93.85 |
| o3 | Vanilla Prompt (Baseline) | 58.12 | 20.73 | 78.46 |
| | Correctness | 70.60 | 22.00 | 50.00 |
| | Correctness × 3 \| Majority | 72.60 | 26.92 | 64.62 |
| | Correctness × 3 \| Unanimous | 82.40 | 29.09 | 24.62 |
| | Correctness ↻ 3 \| Majority | 68.81 | 23.21 | 60.00 |
| | Correctness ↻ 3 \| Unanimous | 76.60 | 28.21 | 50.77 |
| | Correctness × 5 \| Majority | 73.15 | 25.87 | 56.92 |
| | Correctness × 5 \| Unanimous | 83.77 | 26.47 | 13.85 |
| | Correctness ↻ 5 \| Majority | 69.80 | 23.13 | 56.92 |
| | Correctness ↻ 5 \| Unanimous | 78.60 | 28.57 | 43.08 |
| | 1-Iter Pipeline | 75.40 | 24.00 | 42.00 |
| | **3-Iter Pipeline** | **81.65** | **30.99** | **34.38** |
| | 5-Iter Pipeline | 81.50 | 26.23 | 25.40 |
| Multi-model | Correctness (3 Model) \| Majority | 56.20 | 20.16 | 80.00 |
| | Correctness (3 Model) \| Unanimous | 77.40 | 23.33 | 32.31 |
| | Correctness (5 Model) \| Majority | 45.00 | 17.99 | 90.77 |
| | Correctness (5 Model) \| Unanimous | 78.60 | 25.00 | 32.31 |
| | **3-Iter Pipeline (2 Model) \| Unanimous** | **85.40** | **40.00** | **24.62** |
| | Debate (3 Model) | 77.60 | 24.73 | 35.38 |

Table S7: **UQ-Validators metrics** (augmenting Table 1). Scores are computed on 500 subsampled HLE question-answer pairs, where ground-truth is withheld during validator judgment. × and ↻ denote *repeated* and *iterated* sampling, e.g. "Correctness ×3 | Majority" repeats the correctness check thrice and takes majority vote. Pipelines are the following *sequential verification* strategies: 1-Iter = [CC⇒FLC⇒C]; 3-Iter = [(CC×3 | U)⇒(FLC×3 | U)⇒(C×3 | U)], with C = correctness, FLC = fact/logic check, U = unanimous vote. Boldface marks the best UQ-Validators.

### E.5 Additional Findings

This section augments Section 3.3 and provides additional findings regarding the UQ-Validators.

**Finding #6: Better Answer Generators May Not Be Better Answer Validators**

We also find that a better answer generator may not, in general, be a better answer validator. Figure S4 shows the validation accuracy of a model via baseline prompting and a 3-iter validation pipeline (recall Figure S2) against its answer accuracy over 500 HLE questions. While better answer performance is broadly indicative of better validation performance (general upright trend), it is not always the case. For example, without any pipeline validation, o3 is a weaker answer model yet a stronger validator compared to Gemini 2.5 Pro. With pipeline validation, we observe the same reversal trend between o3-mini and Claude 3.7 Sonnet. Also, while Claude Sonnet 3.7 substantially underperforms Gemini 2.5 Pro in answer accuracy, its pipeline-based validation performance is higher than the baseline validation performance of Gemini 2.5 Pro.

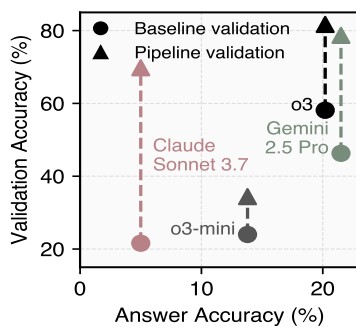

Figure S4: Generation vs. validation accuracies across four models.

**Finding #7: Validation Strategies are (Somewhat) Amenable to Test-Time Scaling**

We additionally explore whether answer validation is amenable to scaling in the sense that spending more test-time inference calls and tokens would yield better performance. Figure S5 shows a scaling trend: validation accuracy generally increases as we allocate more API calls for the validator. Sequential pipelines and unanimity voting consistently outperform single-prompt baselines, with deeper pipelines achieving the highest accuracy at greater cost. We also observe diminishing marginal gains as the call budget grows, reflecting a natural cost-accuracy trade-off. Multi-model unanimous voting (o3 + Gemini 2.5 Pro) attains the best accuracy among the tested strategies, indicating that model diversity further reduces judgment variance beyond additional turns with a single model.

Importantly, and as discussed in Section 3.3, prompt design matters even at a fixed, small budget. Among single-call strategies, a structured "Correctness" prompt substantially outperforms the generic vanilla baseline prompt.

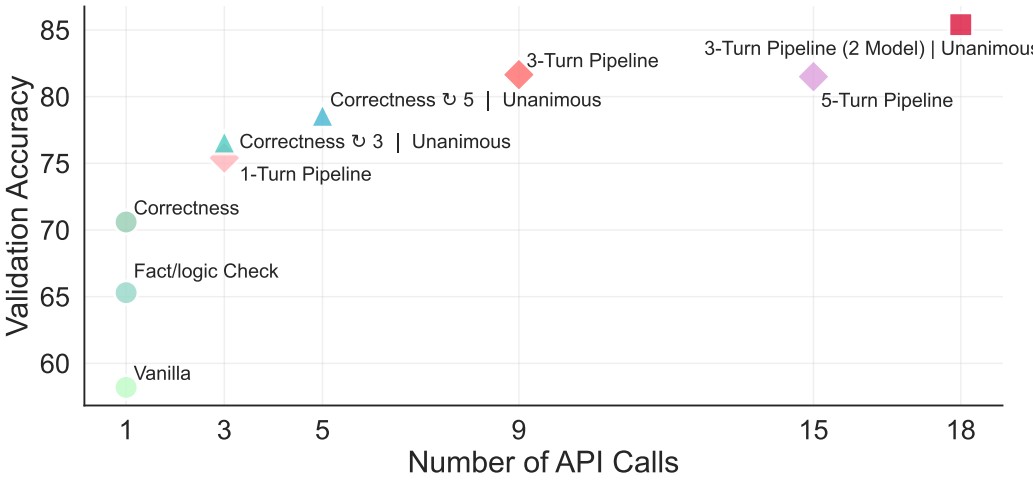

Figure S5: **Scaling behaviors of validation strategies.** Validation accuracy vs. per-answer API calls on 500 HLE questions, comparing single-prompt baselines (Vanilla Baseline, Fact/Logic Check, Correctness) with sequential pipelines and unanimity voting, including a 3-Iter, 2-model unanimous pipeline. We use o3 as the judge model except the "2 Model" strategy where we both use o3 and Gemini 2.5 Pro. "Vanilla Baseline" means we directly ask the model to give a judgment without detailed prompts (see Appendix I.7). Accuracy generally improves as we spend more calls and/or ensemble models, with deeper pipelines yielding the highest accuracy at greater cost.

**Finding #8: Weaker Models Fail Earlier in UQ-Validator Pipeline**

We additionally perform a more granular analysis on the UQ-Validators pass rates across different answer models. Figure S6 shows, for different answer models of increasing strength, where the answer model fails in the 3-stage UQ-Validator pipeline (recall Figure S2). Observe that:

- **Stronger models fail less often in early stages.** Models like o3-pro and Gemini 2.5 Pro have very few answers failing Stage 1, while weaker models (e.g., GPT-4o, Claude Sonnet 3.7) fail early more frequently.
- **Fully validated answers correlate with model strength.** Stronger models generate more answers that pass all three validation stages (as opposed to just pass more but not all stages), with o3-pro achieving the highest pass rate.
- **Some models often fail at factual checks.** Models such as Claude Opus 4 and DeepSeek-R1 frequently fail at Stage 2, suggesting their answers are fluent but factually unreliable.
- **Pipeline stages are calibrated.** Failures are distributed across stages as opposed to concentrating at a particular stage. This indicates that each stage of the three-stage UQ-Validator adds meaningful filtering, and the pipeline is not overly strict at the end.

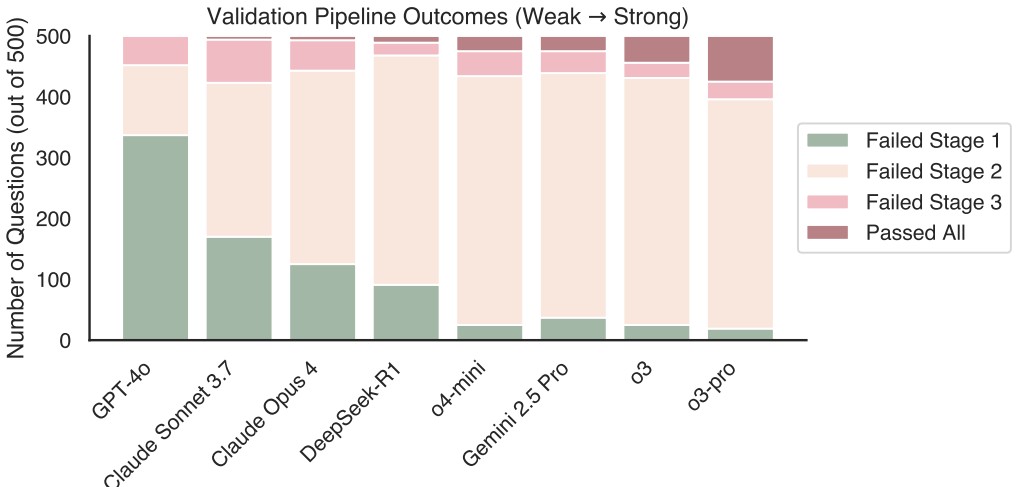

Figure S6: **Validation outcomes across models.** We visualize the outcomes of different answer models across 500 questions when applied a 3-stage answer validation pipeline (UQ-Validator). Each stacked bar represents the number of answers that failed at each validation stage (Stage 1, 2, or 3), or passed all stages. Stronger models (right) tend to fail less frequently in early stages and provide more answers that pass all validation stages, while weaker models (left) generate answers that are more likely to be filtered out early. This highlights the correlation between model strength and robustness to multi-stage answer validation.

# F  ADDITIONAL DETAILS ON UQ-PLATFORM

## F.1  SUMMARY OF KEY FEATURES

Here we provide the key features of our proposed UQ-Platform:

- **Question browsing and sorting.** Users can sort questions by votes, resolution status, categories, and Stack Exchange sites. Each question has a dedicated page displaying candidate answers from frontier models (e.g., o3-pro, Gemini 2.5 Pro) alongside human reviews and comments.
- **Answer submissions.** Model developers can submit answers to open questions either for new models/systems or their updated versions. Submissions must include an organization name, system name, base model (if applicable), candidate answer, and full prompt for reproducibility.
- **Human reviews.** Users can submit reviews for candidate model answers under each question. Reviews consist of a *correctness* and *confidence* ratings similar to academic peer reviews, and are shown along the model answer for public review. Users can also comment on the question quality.
- **UQ-validation and additional AI reviews.** UQ-Validator results are displayed along candidate answers, and developers can submit additional answer reviews by their models/systems to augment the UQ-validation. This facilitates future work on better oracle-free validation models or strategies.
- **Resolution statistics.** The platform provides an overview of the dataset's resolution status, UQ-Validator pass rates, number of resolved questions, number of unique models evaluated, etc.
- **Model ranking.** Models are ranked based on their number of verified resolved questions. Note that initial rankings may have limited informative value as during the current release: (1) models solve very few questions, and (2) we are unable to verify all candidate model answers.

## F.2  USER INCENTIVES

As evaluation critically hinges on user contributions on UQ-Platform, we envision the following incentivizing factors apart the properties mentioned earlier:

1. *Public attribution.* UQ-Platform may offer lightweight reputation signals (e.g., verifier badges) to active users. Original question posters on Stack Exchange are also explicitly invited to verify solutions and receive public attribution.
2. *Educational use.* Educators or learners may find reading and critiquing model candidate answers on UQ-Platform (e.g., spotting logical errors and hallucinated citations) to be educationally valuable and they may produce high-quality reviews as a by-product.

In the same way that users are incentivized to engage on Stack Exchange, we hope that the platform's convenience, attribution, and educational value will similarly sustain expert participation and improve evaluation quality.

## F.3  SCREENSHOTS OF UQ-PLATFORM

Due to the double-blind reviewing process, we attach screenshots of the UQ-Platform for visualization purposes in lieu of a link to a live deployment:

- Figure S7 shows the landing page. We show, for each model, a live (but slowly updating) counter of how many questions for which it produces answers that pass our best UQ-Validator and that are *human verified* to be correct (main ranking metric).
- Figure S8 shows the top questions in the UQ-Dataset as ranked by engagement metrics. These are typically the hardest, longest standing questions that have accrued significant community attention and interest.
- Figure S9 shows an example of a specific question page. This page shows the question and relevant information such as its source, its published date, tags, and community engagement scores. It then shows model attempted answers on this question, and each answer's status (e.g. UQ-validated or not). This page is in essence an AI-mirror of the corresponding Stack Exchange page, where users can seek AI answers that the original Stack Exchange site may forbid.
- Figure S10 shows an example of a model answer page. It provides the full answer by the model, its status, as well as the raw markdown for downstream processing. It also shows any AI-generated or human-generated reviews for this answer, allowing a "peer review" of such AI answers.

- Figure S11 shows the answer submission form under each question. Anyone can submit an answer from their model, along with necessary info such as prompt for reproducibility and URL for more details; the entire submission is publicly viewable.

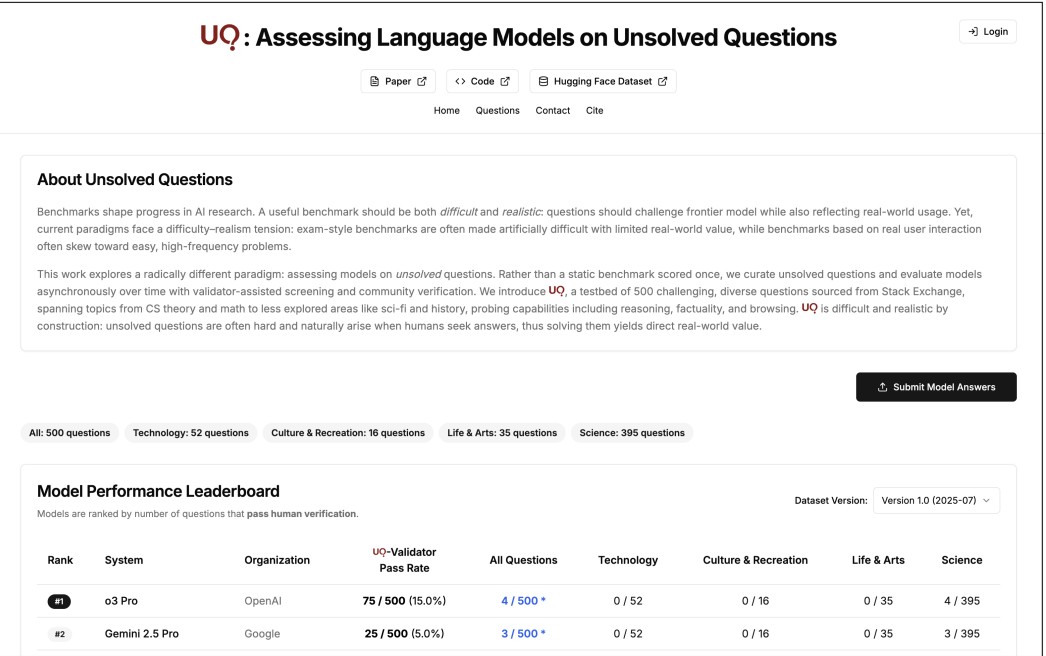

Figure S7: Screenshot of UQ-Platform landing page.

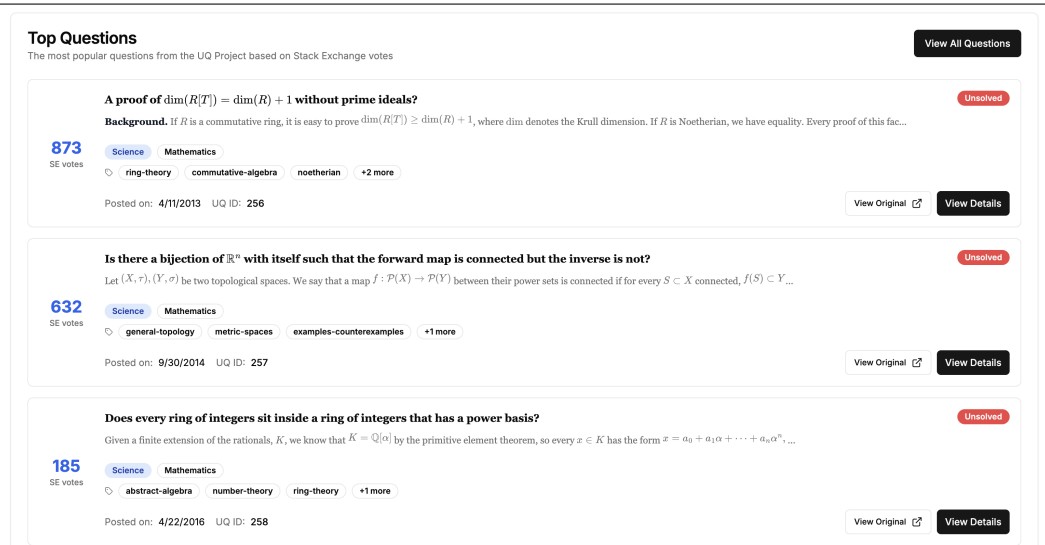

Figure S8: Screenshot of UQ-Platform top questions page.

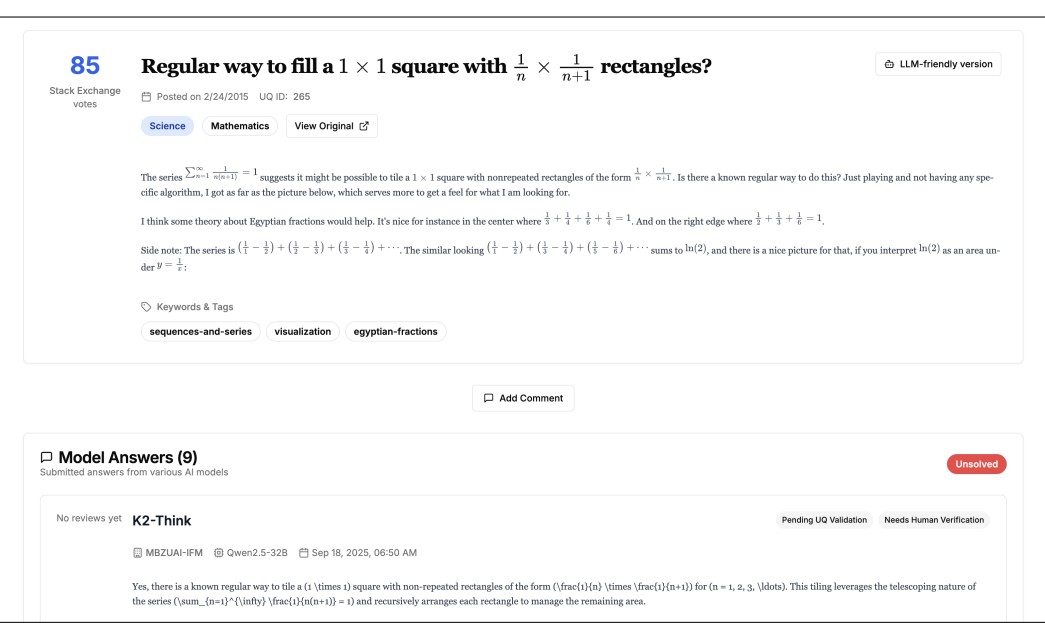

Figure S9: Screenshot of a specific question page on UQ-Platform.

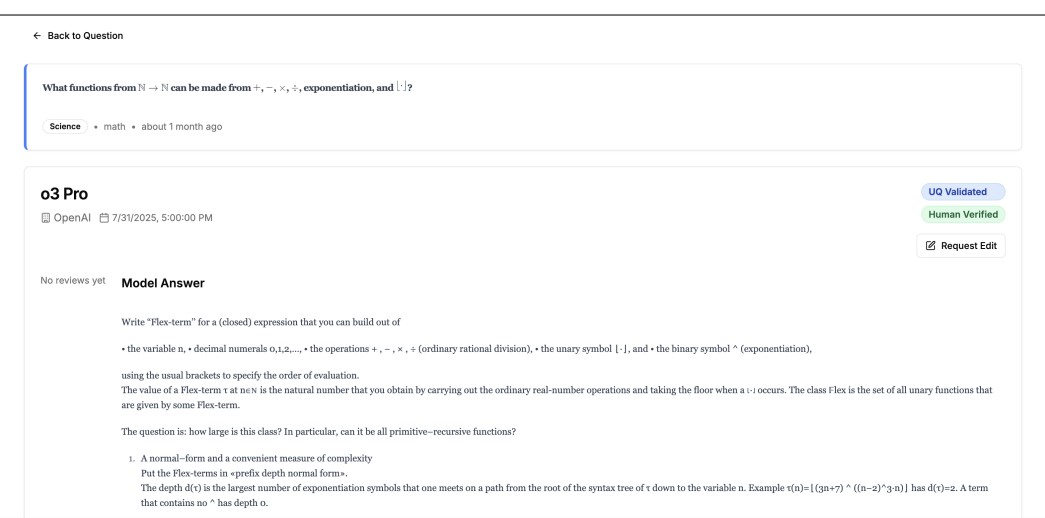

Figure S10: Screenshot of a specific answer page on UQ-Platform.

**Submit Your Model's Answer**

Your submission will be validated by our LLM validator and then by human experts

System Name *

e.g., o3, Claude-3.5-Sonnet

If using a single model, enter the model name (e.g., o3)

Organization *

Enter your organization

Base Model

e.g., GPT-4, Claude-3, Llama-2 (optional)

Prompt

Enter the prompt you used to generate the answer (optional)...

Answer *

Enter your model's answer to this question...

Anything Else

Any additional information or context you'd like to provide (optional)...

URL

https://example.com (optional reference URL)

⚠ Please review your submission carefully. Once submitted, any edits or deletions must be requested.

⬆ Submit Answer

Figure S11: Screenshot of answer submission page on UQ-Platform.

# G ADDITIONAL EXPERIMENTAL DETAILS

## G.1 MODEL VERSIONS

Table S8 lists the specific model versions used throughout the UQ project. Unless otherwise stated, all models use temperature 0.0 when configurable for deterministic sampling.

| Model | Version |
|---|---|
| O3-PRO | o3-pro-2025-06-10 |
| O3 | o3-2025-04-16 |
| O4-MINI (HIGH) | o4-mini-2025-04-16 |
| GEMINI 2.5 PRO | June 2025 release |
| CLAUDE OPUS4 | claude-opus-4-20250514 |
| CLAUDE 3.7 SONNET | claude-3-7-sonnet-20250219 |
| O3-MINI (HIGH) | o3-mini-2025-01-31 |
| DEEPSEEK-R1 | DeepSeek-R1-0528 |
| GPT-4O | gpt-4o-2024-08-06 |

Table S8: Model versions used throughout the project.

## G.2 ADDITIONAL HYPERPARAMETERS

Apart from hyperparameters described along each experimental setting in the main text, we summarize additional hyperparameters that may be helpful in this section.

- UQ-Dataset LLM-based filtering (Section 2.1): Candidate answers are generated using gpt-4o with a temperature of 0.3 and a single inference call. The LLM-based question quality judge employs o4-mini with three inference calls; this model does not permit user-specified decoding temperature.
- Answer generation (Section 3 and Section 5): For o3-mini and o4-mini, we set the reasoning effort to high. For Claude 3.7 Sonnet, we allocate a thinking budget of 16,000. For all models that allow temperature setting, the temperature is set to 0.3.
- UQ-Validators for model evaluation (Section 5): UQ-Validator pass rate is calculated by using o3 with our 3-iter pipeline (Figure S2) as the UQ-Validator.

## G.3 ANECDOTAL HUMAN PERFORMANCE

During our extensive analysis of questions and answers, we solved one of the 500 questions in the current version of UQ-Dataset, and a Stack Exchange user solved another (see Appendix I.5). Thus, we may seed our leaderboard on the UQ-Platform with a human performance of 2 / 500. Further questions in UQ-Dataset resolved by humans, such as via natural interactions on Stack Exchange, can be added to this potential human entry in our leaderboard.

For dataset version consistency, we decide to keep these questions in the UQ-Dataset for assessing models and may indicate this discussion on the UQ-Platform.

## H    INTERACTIONS WITH STACK EXCHANGE

### H.1    CONTENT PERMISSIONS AND LICENSING

**Stack Exchange Licensing.**    UQ uses user-contributed Q&A from Stack Exchange network sites (e.g., Stack Overflow, Cross Validated). These contributions ("Subscriber Content") are copyrighted and licensed under *Creative Commons Attribution–ShareAlike (CC BY-SA).*[1] Per the Help Center, license versions depend on post date: CC BY-SA 2.5 (before 2011-04-08 UTC), 3.0 (2011-04-08 to 2018-05-02), and 4.0 (on/after 2018-05-02).

Stack Exchange's Public Network Terms of Service[2] state that "Subscriber Content" is "licensed ... pursuant to Creative Commons . . .  CC BY-SA 4.0," and that "all such Public Content must have appropriate attribution." These terms apply network-wide (e.g., stats.stackexchange.com carries the same Public Network Terms). UQ also uses the Stack Exchange API, whose terms of use lay out guidelines to follow.[3] On the use of Stack Exchange site logos, guidance is provided separately.[4]

**What this means for UQ.**    We do *not* need special permission to include Stack Exchange excerpts, provided we follow CC BY-SA: attribute and share-alike. In this paper, on the UQ-Platform, and in any dataset releases (e.g., Hugging Face), we: (i) name the source site, (ii) link to the original post(s), (iii) indicate any edits/truncation, and (iv) note that we follow the CC BY-SA.

Our use of Stack Exchange site logos is limited to identifying source sites in an academic, non-commercial context. Stack Exchange's trademark guidance notes that reproducing a logo in editorial coverage (e.g., a news story or blog post) is generally permissible and not "in trade"; a scholarly article is analogous. We do not alter the marks, imply sponsorship or endorsement, or use them in any promotional materials.

### H.2    UPLOADING CANDIDATE ANSWERS TO STACK EXCHANGE

Recall in Section 2.3 that when a candidate model answer passes UQ-Validator and human verification, we may consider posting the answer to the source question. For such posting, we pay attention to the individual AI policies mantained by Stack Exchange.

At the time of this writing, sites on Stack Exchange maintain individual AI policies, and there is no network-wide ban.[5] For example:

- **Cross Validated**: "Generative artificial intelligence (a.k.a. GPT, LLM, generative AI, genAI) tools can be used to generate content for Cross Validated, but this content must be properly referenced as per our guidance."[6]
- **Cryptography**: "Generative artificial intelligence (a.k.a. GPT, LLM, generative AI, genAI) tools can be used to generate content for Cryptography Stack Exchange, but this content must be properly referenced as per our guidance."[7]
- **History**: "Generative artificial intelligence (a.k.a. GPT, LLM, generative AI, genAI) tools can be used to generate content for History Stack Exchange, but this content must be properly referenced as per our guidance."[8]
- **Mathematics**: "Generative artificial intelligence (a.k.a. GPT, LLM, generative AI, genAI) tools may not be used to generate content for Mathematics Stack Exchange."[9]
- **MathOverflow**: "If the mathematical component of your content is deemed to be generated by AI, it will likely be deleted, along with any reputation earned from it. Repeatedly posting AI-generated

---

[1]https://creativecommons.org/licenses/by-sa/4.0/deed.en
[2]https://stackoverflow.com/legal/terms-of-service/public
[3]https://stackoverflow.com/legal/api-terms-of-use
[4]https://policies.stackoverflow.co/company/trademark-guidance
[5]https://meta.stackexchange.com/questions/384396/ban-chatgpt-network-wide/385002#385002
[6]https://stats.stackexchange.com/help/gen-ai-policy
[7]https://crypto.stackexchange.com/help/gen-ai-policy
[8]https://history.stackexchange.com/help/gen-ai-policy
[9]https://math.stackexchange.com/help/gen-ai-policy

mathematical content may lead to a warning from moderators, or possibly a suspension for repeated infractions."[10]

- **Physics**: "Generative artificial intelligence (a.k.a. GPT, LLM, generative AI, genAI) tools can be used to generate content for Physics Stack Exchange, but this content must be properly referenced as per our guidance."[11]

- **Puzzling**: "Generative artificial intelligence (a.k.a. GPT, LLM, generative AI, genAI) tools can be used to generate content for Puzzling Stack Exchange, but this content must be properly referenced as per our guidance."[12]

- **Science Fiction & Fantasy**: "Generative artificial intelligence (a.k.a. GPT, LLM, generative AI, genAI) tools may not be used to generate content for Science Fiction & Fantasy Stack Exchange."[13]

- **Stack Overflow**: "Generative artificial intelligence (a.k.a. GPT, LLM, generative AI, genAI) tools may not be used to generate content for Stack Overflow."[14]

- **Theoretical Computer Science**: "Generative artificial intelligence (a.k.a. GPT, LLM, generative AI, genAI) tools can be used to generate content for Theoretical Computer Science Stack Exchange, but this content must be properly referenced as per our guidance."[15]

---

[10] https://mathoverflow.net/help/gen-ai-policy
[11] https://physics.stackexchange.com/help/gen-ai-policy
[12] https://puzzling.stackexchange.com/help/gen-ai-policy
[13] https://scifi.stackexchange.com/help/gen-ai-policy
[14] https://stackoverflow.com/help/gen-ai-policy
[15] https://cstheory.stackexchange.com/help/gen-ai-policy

# I VISUALIZATIONS

## I.1 SAMPLE QUESTIONS FROM UQ-DATASET

We visualize some sample questions from the UQ-Dataset in this section.

---

**Sample Question from Mathematics**

- **Title:** Does every ring of integers sit inside a ring of integers that has a power basis?
- **Keywords:** abstract-algebra, number-theory, ring-theory, algebraic-number-theory
- **Site:** math
- **Link:** https://math.stackexchange.com/questions/1754860

Given a finite extension of the rationals, $K$, we know that $K = \mathbb{Q}[\alpha]$ by the primitive element theorem, so every $x \in K$ has the form

$$x = a_0 + a_1\alpha + \cdots + a_n\alpha^n,$$

with $a_i \in \mathbb{Q}$.

However, the ring of integers, $\mathcal{O}_K$, of $K$ need not have a basis over $\mathbb{Z}$ which consists of 1 and powers of a single element (a power basis). In fact, there exist number fields which require an arbitrarily large number of elements to form such a basis.

**Question**: Can every ring of integers $\mathcal{O}_K$ that does not have a power basis be extended to a ring of integers $\mathcal{O}_L$ which does have a power basis, for some finite $L/K$?

---

**Sample Question from Math Overflow**

- **Title:** A kaleidoscopic coloring of the plane
- **Keywords:** real-analysis, mg.metric-geometry, measure-theory, harmonic-analysis, geometric-measure-theory
- **Site:** mathoverflow
- **Link:** https://mathoverflow.net/questions/219860

**Problem.** Is there a partition $\mathbb{R}^2 = A \sqcup B$ of the Euclidean plane into two Lebesgue measurable sets such that for any disk $D$ of the unit radius we get $\lambda(A \cap D) = \lambda(B \cap D) = \frac{1}{2}\lambda(D)$?

(I.V.Protasov called such partitions *kaleidoscopic*).

Observe that for the $\ell_1$- or $\ell_\infty$-norms on the plane such partitions exist: just take a suitable chessboard coloring.

The problem can be reformulated in terms of convolutions: *Is there a measurable function $f : \mathbb{R}^2 \to \{1, -1\}$ such that its convolution with the characteristic function $\chi_D$ of the unit disk $D$ is identically zero?*

(The problem was posed 08.11.2015 by T.Banakh and I.Protasov on [page 19][1] of [Volume 0][2] of the [Lviv Scottish Book][3]).

[1]: http://www.math.lviv.ua/szkocka/viewpage.php?vol=0&page=19

[2]: http://www.math.lviv.ua/szkocka/viewbook.php?vol=0

[3]: http://www.math.lviv.ua/szkocka

---

**Sample Question from Science Fiction & Fantasy**

- **Title:** Looking for science fiction assassination story with mysterious girl
- **Keywords:** story-identification, short-stories
- **Site:** scifi

• **Link:** https://scifi.stackexchange.com/questions/27694

For years I've been looking for a short science fiction story about a man who was released from a prison planet in order to assassinate a candidate for galactic president. He does this on a space station, and escapes with the help of a mysterious and very attractive girl with silvery hair and multi-colored skin.

<added after a few days> I read this short story some 5-6 years ago online.

The assassin was released from the prison planet by some influential people so that he may assassinate the galactic president candidate, whose rule would, according to said people, be bad for the galaxy.

I also remember that the girl, who is actually a young woman is also a prostitute. The first time he sees her she's walking into the bar accompanied by another man. I don't remember why she helps him. She was likely his contact, given to him by those who sent him on the assassination.

They escape together after creating a diversion with an explosion, running off into some corridor that probably leads to a way off the space station.

The story story ends there but in my opinion leaves room for a sequel, even if only to explore the protagonists' developing relationship.

## Sample Question from Theoretical Computer Science

• **Title:** Problem unsolvable in $2^{o(n)}$ on inputs with $n$ bits, assuming ETH?
• **Keywords:** cc.complexity-theory, sat, planar-graphs, succinct
• **Site:** cstheory
• **Link:** https://cstheory.stackexchange.com/questions/16148

If we assume the Exponential-Time Hypothesis, then there is no $2^{o(n)}$ algorithm for $n$-variable 3-SAT, and many other natural problems, such as 3-COLORING on graphs with $n$ vertices. Notice though that, in general, encoding the input for $n$-variable 3-SAT or $n$-vertex 3-COLORING takes something like $O(n \log n)$ bits. For example, to describe a sparse graph as input to 3-COLORING, for each edge we would have to list its endpoints. So the lower bound is not exponential in the length of the input. Therefore, my question is the following:

Is there a problem for which no $2^{o(n)}$ algorithm exists for inputs of length $n$ bits (assuming ETH)?

Ideally, the problem would be in NP (no cheating with succinct NEXP-hard problems!) and be reasonably natural, but I won't be picky.

Let me also note that after digging around I found that there are efficient ways to encode planar graphs with $O(n)$ bits. So, if one could find a problem that takes time exponential in the number of vertices even for planar graphs, the question would be settled. However, because planar graphs have treewidth $O(\sqrt{n})$, most natural problems have sub-exponential algorithms in this case.

## Sample Question from Physics

• **Title:** Quantum statistics of branes
• **Keywords:** statistical-mechanics, string-theory, topological-field-theory, branes, quantum-statistics
• **Site:** physics
• **Link:** https://physics.stackexchange.com/questions/26826

Quantum statistics of particles (bosons, fermions, anyons) arise due to the possible topologies of curves in $D$-dimensional spacetime winding around each other

What happens if we replace particles with branes? It seems like their quantum statistics should be described by something like a generalization of TQFT in which the "spacetime" (world brane) is equipped with an embedding into an "ambient" manifold (actual spacetime). The inclusion of non-trivial topology for the "ambient" manifold introduces additional effects, to 1st approximation describable by the inclusion

of k-form fluxes coupling to the brane. To 2nd approximation, however, there is probably non-trivial coupling between these fluxes and the "generalized quantum statistics"

A simple example of non-trivial "brane quantum statistics" is the multiplication of quantum amplitudes of strings by the exponential of the Euler characteristic times a constant. In string theory, this corresponds to changing the string coupling constant/dilaton background.

> Were such generalized TQFTs studied? Which non-trivial examples are there for branes in string theory?

---

### Sample Question from History

- **Title:** What was the first overland road from Sweden to Finland?
- **Keywords:** transportation, sweden, finland
- **Site:** history
- **Link:** https://history.stackexchange.com/questions/62286

The Swedish post road [1] from Norway, through Sweden, used the Åland archipelago to pass into Sweden, and this is easily found (evidence of) in the south of Finland to the present day. **When (and where) was the first overland route constructed overland from Sweden into (Swedish) Finland?**

The only (poor) evidence I have for roads existing in the north is by the War of 1808–9 [2] where Russian forces were planning to advance overland into Sweden (along with an army group advancing across the Gulf of Bothnia). One of the WP article's references does say "In addition, several new good roads had been built into Finland greatly reducing the earlier dependency on naval support for any large operation in Finland." but it doesn't specify where these roads were.

I looked through all articles on Swedish [3] and Finnish road networks on the English Wikipedia, and the most I found was a reference to a 'Finnmark path' [4] which was meant to have gone from Finnish Lapland to Finnmark in the 16th century. The Finnish WP article for the same page *does not* mention the Finnmark path at all, and I couldn't find anything else on a road of that name.

I understand—from the comments—that the term "road" can be meaningless without further definition for a period much longer than a few centuries ago. For clarity, I'm defining road as purpose-built (or purpose-developed) and used regionally for that purpose, such as the post road mentioned above. This would mean hunting tracks that slowly developed don't count, while a merchant-led endeavour to expand (and maintain) the tracks between two townships would.

[1]: https://en.wikipedia.org/wiki/King's_Road_(Finland)

[2]: https://en.wikipedia.org/wiki/Finnish_War

[3]: https://en.wikipedia.org/wiki/Category:Roads_in_Sweden

[4]: https://en.wikipedia.org/wiki/Arctic_Ocean_Highway

## I.2 Sample Judgment Reasoning Traces by UQ-Validator

We visualize some sample judgment reasoning traces of UQ-Validators that have also been verified by human reviewers. The $\boxed{\text{✓Human-reviewed}}$ box means that the judgment reasoning trace has been approved by a human reviewer.

### I.2.1 Sample #1: Mathematics

---

**Sample #1 Question**

- **Title:** A question connected with the decomposition of a functional on $C(X)$ on Riesz and Banach functionals
- **Keywords:** real-analysis, probability-theory, measure-theory
- **Site:** math
- **Link:** https://math.stackexchange.com/questions/54377

Let $X$ be a metric space and let $C(X)$ be a family of all bounded and continuous functions from $X$ in $\mathbb{R}$.

We call a positive linear functional $\varphi : C(X) \to \mathbb{R}$ the functional of Riesz if there is a borel measure $\mu$ on $X$, such that $\varphi(f) = \int_X f \, d\mu$, for $f \in C(X)$.

We call a positive linear functional $\varphi : C(X) \to \mathbb{R}$ the functional of Banach if for each borel measure $\nu$ on $X$ the condition: $\int_X f d\nu \leq \varphi(f)$, for $f \in C(X)$ - implies that $\nu$ is trivial.

There is a well known theorem :

Let $X$ be a polish space. Then, for each positive linear functional $\varphi : C(X) \to \mathbb{R}$ there is a unique couple $(\varphi_0, \varphi_*)$ of positive linear functionals defined on $C(X)$, such that $\varphi_0$ is the functional of Riesz, $\varphi_*$ is the functional of Banach and $\varphi = \varphi_* + \varphi_0$. Moreover, the measure $\mu$ related to $\varphi_0$ is defined by:

$$\mu(K) = \inf\{\varphi(f) : f \in C(X), 1_X \geq f \geq 1_K\},$$

for each compact set $K \subset X$.

More pecisely, for the proof, we define:

$$\varphi_\delta(f) = \sup\{\varphi(h) : \ \operatorname{supp} h \in N(\delta), 0 \leq h \leq f\},$$

for $\delta > 0$,

$$\varphi_0(f) = \lim_{\delta \to 0^+} \varphi_\delta(f),$$

for $f \in C(X), f \geq 0$, and

$$\varphi_0(f) = \varphi_0(f^+) - \varphi_0(f^-),$$

for $f \in C(X)$, where $N(\delta)$ is a family of sets that possess a covering composed of finite number of open balls with a radius equal to $\delta$.

My question concerns the truth of the following sentence: Let $X$ be a $\sigma$-compact and polish space. Assume that $\varphi^x : C(X) \to \mathbb{R}$ is a positive linear functional, for all $x \in X$ and let $((\varphi^x)_0, (\varphi^x)_*)$ be a couple of Banach-Riesz functionals, for $x \in X$. If the mapping $X \ni x \mapsto \varphi^x(f)$ is continuous for all $f \in C(X)$ and $\varphi^x(1_X) = 1$, for $x \in X$, then mapping $X \ni x \mapsto (\varphi^x)_0(f)$ is continuous for all $f \in C(X)$ (or may be for only $f \in C_c(X)$).

I was able to proof only that the mapping $X \ni x \mapsto (\varphi^x)_0(f)$ is upper semi-continuous, for $f \in C_c(X)$.

---

**Sample #1 Candidate Answer by o3**

Let $X$ be $\sigma$–compact and Polish and assume that for every $x \in X$ we have a positive linear functional $\varphi^x : C(X) \to \mathbb{R}$ with $\varphi^x(1) = 1$, and that the map $x \mapsto \varphi^x(f)$ is continuous for all $f \in C(X)$. For every $x$, let $\varphi^x = (\varphi^x)_0 + (\varphi^x)^*$ be the Riesz–Banach (Yosida–Hewitt) decomposition. We prove that for every $f \in C(X)$ (already for every $f \in C_c(X)$) the map

$$x \mapsto (\varphi^x)_0(f) = \int f \, d\mu^x$$

(with $\mu^x$ the Radon measure attached to $(\varphi^x)_0$) is continuous. (The argument below shows slightly more: $x \mapsto \mu^x$ is continuous for the weak topology on the space of Radon probability measures.)

**1. A basic property of the Banach part.** Let $\varphi$ be a positive functional on $C(X)$ and write $\varphi = \varphi_0 + \varphi^*$ for its Riesz–Banach decomposition. Recall

$$\varphi_\delta(f) = \sup\{\ \varphi(h) : 0 \le h \le f,\ \operatorname{supp} h \in N(\delta)\ \}, \qquad \varphi_0(f) = \lim_{\delta \to 0} \varphi_\delta(f).$$

If $0 \le h \in C_c(X)$ and $\operatorname{diam}(\operatorname{supp} h) < \delta$, then $h$ itself belongs to the admissible family that defines $\varphi_\delta(h)$, so $\varphi_\delta(h) \ge \varphi(h)$. Because always $\varphi_\delta(h) \le \varphi(h)$, we get equality:

$$\varphi_\delta(h) = \varphi(h) \qquad \text{whenever } \delta > \operatorname{diam}(\operatorname{supp} h).$$

Consequently $\varphi_0(h) = \varphi(h)$ and therefore

$$\varphi^*(h) = \varphi(h) - \varphi_0(h) = 0. \tag{1}$$

Hence, if $\operatorname{diam}(\operatorname{supp} h)$ is small enough, then $(\varphi^x)^*(h) = 0$ for every $x \in X$.

**2. "Small–support" functions give continuity immediately.** Because of (1), for such an $h$ we have

$$(\varphi^x)_0(h) = \varphi^x(h) \qquad (\forall x).$$

But $x \mapsto \varphi^x(h)$ is continuous by hypothesis; hence

$$x \mapsto (\varphi^x)_0(h) \ \text{ is continuous}$$

for every (not necessarily positive) $h$ whose support is sufficiently small.

**3. Continuity for every $f \in C_c(X)$.** Let $f \in C_c(X)$ and fix $x_0 \in X$, $\varepsilon > 0$. Since $f$ is uniformly continuous (its support $K$ is compact), choose $\eta > 0$ such that

$$|f(y) - f(z)| < \varepsilon \qquad \text{whenever } d(y,z) < \eta.$$

Cover $K$ by finitely many open balls $B_1, \dots, B_m$ of radius $\eta$ and pick a partition of unity $\psi_1, \dots, \psi_m$ with $0 \le \psi_i \le 1$ and $\operatorname{supp} \psi_i \subset B_i$. Choose $y_i \in B_i$ and put

$$g := \sum_{i=1}^m f(y_i)\, \psi_i.$$

Then $|f - g| \le \varepsilon$ on $X$. Because each $\operatorname{supp} \psi_i \subset B_i$, we have $\operatorname{diam}(\operatorname{supp} \psi_i) < 2\eta$; taking $\eta$ small so that (1) applies yields $(\varphi^x)_0(\psi_i) = \varphi^x(\psi_i)$ and thus $x \mapsto (\varphi^x)_0(\psi_i)$ is continuous.

Write

$$A_x := (\varphi^x)_0(f), \qquad C_x := \sum_{i=1}^m f(y_i)\, (\varphi^x)_0(\psi_i).$$

For every $x$,

$$|A_x - C_x| \le (\varphi^x)_0(|f - g|) \le \|f - g\|_\infty \le \varepsilon.$$

Because each $x \mapsto (\varphi^x)_0(\psi_i)$ is continuous, so is $x \mapsto C_x$. Hence, for $x \to x_0$,

$$|A_x - A_{x_0}| \le |A_x - C_x| + |C_x - C_{x_0}| + |C_{x_0} - A_{x_0}| \le 2\varepsilon + |C_x - C_{x_0}|.$$

The middle term tends to 0, so

$$\limsup_{x \to x_0} \big| (\varphi^x)_0(f) - (\varphi^{x_0})_0(f) \big| \le 2\varepsilon.$$

Since $\varepsilon > 0$ is arbitrary, $x \mapsto (\varphi^x)_0(f)$ is continuous for every $f \in C_c(X)$.

**4. From compact support to all bounded continuous functions.** Let $f \in C_b(X)$. Because $X$ is $\sigma$–compact, choose an increasing sequence of compact sets $K_n \uparrow X$ and $\chi_n \in C_c(X)$ with $0 \le \chi_n \le 1$, $\chi_n = 1$ on $K_n$, and $\chi_n = 0$ outside $K_{n+1}$. Put $f_n := \chi_n f$. Then $f_n \to f$ uniformly, and each $f_n$ has compact support.

For every $n$, $x \mapsto (\varphi^x)_0(f_n)$ is continuous by Step 3. Moreover,

$$\big| (\varphi^x)_0(f) - (\varphi^x)_0(f_n) \big| \le \|f - f_n\|_\infty\, (\varphi^x)_0(1) \le \|f - f_n\|_\infty.$$

(The measures $\mu^x$ are sub–probability because $\varphi^x(1) = 1$.) Hence the convergence is uniform in $x$, and a uniform limit of continuous functions is continuous. Therefore $x \mapsto (\varphi^x)_0(f)$ is continuous for every bounded continuous $f$, and in particular for every $f \in C(X)$.

**5. Conclusion.** Under the hypotheses above we have proved:

For every $f \in C(X)$ (already for every $f \in C_c(X)$), $\quad x \mapsto (\varphi^x)_0(f) = \int f \, d\mu^x$ is continuous.

Equivalently, $x \mapsto \mu^x$ is continuous for the weak topology on the space of Radon probability measures on $X$.

(Note that only $\sigma$–compactness and the Polish topology of $X$, together with $\varphi^x(1) = 1$ and the given continuity of $x \mapsto \varphi^x(f)$, were used. No additional assumption is needed.)

**Sample #1 Judgment by UQ-Validator (Figure S2)**      ✓ **Human-reviewed**

[Candidate answer fails at the 2nd turn (fact/logic check) of the 3-iter pipeline.]

**Iteration 1**

**Analysis of the answer**
**1. Wrong claim about uniform approximation.** In Step 4 the author writes "We have $f_n \to f$ uniformly, and each $f_n$ has compact support." For a general bounded continuous function $f$ on a non-compact space this is false. Compactly supported functions are uniformly dense only in $C_0(X)$ (the functions that vanish at infinity), not in all of $C_b(X)$. Example: if $X = \mathbb{R}$ and $f \equiv 1$, any compactly supported $g$ satisfies $\|f - g\|_\infty = 1$, so uniform convergence is impossible. Because the proof of continuity for general $f$ relies on this uniform convergence, Step 4 and the conclusion that $x \mapsto (\varphi^x)_0(f)$ is continuous for every $f \in C(X) = C_b(X)$ are unjustified.

**2. Dependence of later arguments on the mistake.** The author uses the alleged uniform convergence to assert that a uniform limit of continuous functions is continuous, thereby extending the result from $C_c(X)$ to all of $C_b(X)$. Since the uniform convergence does not hold, this step is a logical gap and the final statement is not proven.

**3. Earlier steps.** Steps 1–3 (the vanishing of the Banach part on small supports and continuity for functions of compact support) are essentially correct, but they do not cover the whole space $C_b(X)$, so the main claim remains unestablished.

Because of the incorrect claim about uniform approximation and the ensuing logical flaw, the answer contains a significant error that would mislead a reader about the validity of the final result.

**Iteration Verdict: Contains Factual Error**

**Iteration 2**

After re-examining the answer, the flaw noted earlier remains decisive. The proof relies on the claim that every bounded continuous function $f$ on a $\sigma$-compact Polish space can be approximated uniformly by compactly supported functions $f_n = \chi_n f$; this is false (take $f \equiv 1$ on $\mathbb{R}$). Because the alleged uniform convergence is essential for extending the result from $C_c(X)$ to all of $C(X)$, the argument breaks down and the final conclusion is not established. Hence my assessment is unchanged.

**Iteration Verdict: Contains Factual Error**

**Iteration 3**

I re-checked both the mathematics and the logical flow.
● In Step 4 the author claims that, for every bounded continuous function $f$ on a $\sigma$-compact Polish space, one can pick cut-off functions $\chi_n \in C_c(X)$ with $f_n := \chi_n f$ and obtain uniform convergence $f_n \to f$. This is false. Uniform approximation by compactly supported functions holds only for functions that vanish at infinity ($C_0(X)$), not for all of $C_b(X)$. The constant function $f \equiv 1$ on $\mathbb{R}$ gives an immediate counterexample: $\|1 - \chi_n\|_\infty = 1$ for every such $\chi_n$, so uniform convergence is impossible.
● The uniform convergence is pivotal in the proof: it is used to pass from continuity of $x \mapsto (\varphi^x)_0(f_n)$ (true for $f_n$ with compact support) to continuity of $x \mapsto (\varphi^x)_0(f)$ for the original $f$. Because the convergence is not uniform, the estimate that justifies this passage collapses and the conclusion for general $f \in C(X) = C_b(X)$ is unproved.
● Steps 1–3 correctly establish continuity for compactly supported functions, but the main advertised result ("for every $f \in C(X)$") has not been demonstrated.
This is a substantive mathematical error that misleads the reader about the scope of the theorem proved.

**Iteration Verdict: Contains Factual Error**

### I.2.2 SAMPLE #2: HISTORY

---

**Sample #2 Question**

- **Title:** What was the first overland road from Sweden to Finland?
- **Keywords:** transportation, sweden, finland
- **Site:** history
- **Link:** https://history.stackexchange.com/questions/62286

The Swedish post road [1] from Norway, through Sweden, used the Åland archipelago to pass into Sweden, and this is easily found (evidence of) in the south of Finland to the present day. **When (and where) was the first overland route constructed overland from Sweden into (Swedish) Finland?**

The only (poor) evidence I have for roads existing in the north is by the War of 1808–9 [2] where Russian forces were planning to advance overland into Sweden (along with an army group advancing across the Gulf of Bothnia). One of the WP article's references does say "In addition, several new good roads had been built into Finland greatly reducing the earlier dependency on naval support for any large operation in Finland." but it doesn't specify where these roads were.

I looked through all articles on Swedish [3] and Finnish road networks on the English Wikipedia, and the most I found was a reference to a 'Finnmark path' [4] which was meant to have gone from Finnish Lapland to Finnmark in the 16th century. The Finnish WP article for the same page *does not* mention the Finnmark path at all, and I couldn't find anything else on a road of that name.

I understand—from the comments—that the term "road" can be meaningless without further definition for a period much longer than a few centuries ago. For clarity, I'm defining road as purpose-built (or purpose-developed) and used regionally for that purpose, such as the post road mentioned above. This would mean hunting tracks that slowly developed don't count, while a merchant-led endeavour to expand (and maintain) the tracks between two townships would.

[1]: https://en.wikipedia.org/wiki/King's_Road_(Finland)

[2]: https://en.wikipedia.org/wiki/Finnish_War

[3]: https://en.wikipedia.org/wiki/Category:Roads_in_Sweden

[4]: https://en.wikipedia.org/wiki/Arctic_Ocean_Highway

---

**Sample #2 Candidate Answer by Gemini 2.5 Pro**

Okay, let's break down this question about the first *overland* road between Sweden and Finland.

**Key Challenges & Context:**

1. **No Border Before 1809:** As you noted, Finland was part of Sweden until 1809. Therefore, any road connecting the two regions before this date was an *internal* Swedish road, not an international one. This changes the perspective from building a road *to* Finland to extending the Swedish road network *within* its own territory, which included Finland.
2. **Definition of "Road":** Your definition ("purpose-built or purpose-developed and used regionally") is helpful. Early modern "roads" were often designated routes maintained (often poorly) by local farmers, evolving from earlier paths. State-mandated routes like post roads or military roads fit this definition best.
3. **Geography:** The only practical overland route avoids the Baltic Sea entirely by going around the northern end of the Gulf of Bothnia, primarily through the Torne River Valley (Tornedalen/Tornionlaakso).

**The Development of the Northern Route:**

1. **Early Routes:** Basic paths and tracks existed for centuries in the Torne Valley, used by locals, traders (especially the Birkarlians), and administrators. These often followed the river, utilizing river ice in winter ("winter roads") and rough tracks on land in summer. These likely don't meet your "purpose-developed" criteria in their earliest forms.

2. **Tornio's Role:** The founding of Tornio (Torneå) as a town in 1621 signifies the increasing importance of the region. A town requires connections for trade, administration, and communication. This spurred the *gradual improvement* and formalization of routes leading to it.

3. **The Coastal Land Road (Kustlandsvägen):** The main Swedish road network primarily developed along the coast. This coastal road was gradually extended northward over centuries. It eventually reached Umeå, Luleå, and finally Tornio. This wasn't a single project but an incremental process.

4. **Formalization (17th–18th Centuries):**
   • The Swedish state increasingly formalized its road network, particularly with the establishment and regulation of the Gästgivare system (requiring inns and horse-changing stations at regular intervals along designated routes). This system was extended northward.
   • By the 17th and certainly the 18th century, a designated route, functioning as a post road and general highway (by the standards of the time), existed along the coast up to Tornio. From Tornio, roads continued southwards into the Finnish parts of the kingdom, notably towards Oulu (Uleåborg).
   • This route around the Bothnian Bay, via Tornio, became the established *overland* connection between the core Swedish lands and the Finnish lands.

5. **Evidence from the Finnish War (1808–1809):** Your source mentioning "new good roads" built *into* Finland likely refers to improvements made in the later 18th century, perhaps for military preparedness or administrative efficiency. These improvements would certainly have included the crucial northern overland route via Tornio, enhancing its capacity for troop movements and logistics, which was previously very difficult. The Russians *used* this northern route during the war.

**Conclusion:**

It's difficult to pinpoint a single *year* or *project* for the "first" overland road because it was a gradual development *within* the Kingdom of Sweden.

• *Where:* The route went around the northern end of the Gulf of Bothnia, crossing the Torne River near the town of Tornio. This connected the coastal road network of Sweden proper with the road network extending southwards from Tornio into Finland.

• *When:* A functional, designated overland route connecting Sweden proper and Finland proper via Tornio existed and was formalized as part of the Swedish state road and postal system certainly by the **18th century**, with its origins in the gradual northward extension of the coastal road and the formalization of routes following the establishment of Tornio in the **17th century**. Significant improvements likely occurred in the later 18th century, making it more suitable for large-scale movements as seen in the 1808–09 war.

There wasn't a singular "construction event" like building a modern highway. It was an evolution from paths to state-designated routes, incrementally improved over time, primarily during the 17th and 18th centuries. The key artery was the coastal road extending north to Tornio and then continuing south/east from there.

---

**Sample #2 Judgment by UQ-Validator (Figure S2)**     ✓**Human-reviewed**

[Candidate answer fails at the 3rd turn (correctness) of the 3-iter pipeline.]

**Iteration 1** ────────────────────────────────────

Brief evaluation: The assistant identifies the only plausible land connection as the route that skirts the northern end of the Gulf of Bothnia and crosses the Torne River at Tornio/Torneå. That is indeed where the first "purpose-built" land link between the Swedish and Finnish parts of the realm must have been. It further notes that the route evolved out of earlier tracks, was incorporated into the Swedish state road/ postal network, and was certainly functioning as a maintained guest-house/post road by the 18th century, with roots in the 17th century after the founding of Tornio (1621). This matches what the historical sources say in outline: Norrstigen/Kustlandsvägen reached Kalix by c. 1600 and Tornio had gästgiveri posts by 1622, so a formal road connection existed already in the early 17th century and was improved afterwards.

However, the answer never provides a definite date or even a narrow time window—it only states "by the 18th century" and "origins in the 17th century," leaving the reader with a century-long span. It supplies no references or concrete evidence (laws, road orders, postal regulations, maps, etc.) to substantiate its claims. Nor does it mention the key facts that can be cited (e.g., the 1622 gästgiveri regulation or the inclusion of the route in the 1636 postal decree). Consequently, while its general outline is plausible, it

remains speculative and incomplete, and it does not satisfy the question's explicit request for "when (and where)" the first overland road was built.

Because the response fails to deliver a precise, well-supported answer and lacks citations, it should not be judged 95 % correct.

**Iteration Verdict: Incorrect**

**Iteration 2** ─────────────────────────────────────

After reconsideration, I still find the response too vague, unsupported, and imprecise to meet the standard of a correct, complete answer to "when and where" the first purpose-built overland road joined Sweden and Finland. It names the Tornio route and places its formalization somewhere in the 17th–18th centuries, but gives no firm date, concrete evidence, or citations, so it does not satisfy the query with adequate specificity or reliability.

**Iteration Verdict: Incorrect**

**Iteration 3** ─────────────────────────────────────

I have reconsidered and remain convinced my earlier judgment is correct: the response is too imprecise (no firm date, no documentation) to satisfy "when and where" the first purpose-built overland road was made. Therefore my decision stands.

**Iteration Verdict: Incorrect**

## I.3   SAMPLE ANSWERS PASSING UQ-VALIDATOR BUT HUMAN-VERIFIED AS INCORRECT

This section is referenced by Section 5 and provides a sample of question-answer pairs that passed our UQ-Validator (144 question-answer pairs in total, recall Section 5), but the answers are ruled as incorrect during human verification.

### I.3.1   SAMPLE #1: SCIENCE FICTION & FANTASY

---

**Sample #1 Question**

- **Title:** 70s book/short story, Man wakes up from suspend animation or cryo and world has collapsed
- **Keywords:** story-identification, short-stories, post-apocalyptic, suspended-animation
- **Site:** scifi
- **Link:** https://scifi.stackexchange.com/questions/185987/

In the book, the protagonist wakes up from either suspended animation or cryo (Probably supposed to be a short test), After 10-20 years have passed.

He finds the facility abandoned (and looted?), gets to the surface and it looks like a war zone. Wreckage everywhere, smashed buildings, bullet holes everywhere. (I think he gets some gear from the facility, MREs, Water, First Aid kit, Weapons) and goes exploring to find out what happened.

He comes across a (young kid?) who is terrified of him, (maybe trapped or pinned under debris), gives him food and water and digs him out or helps him so the kid trusts him. Eventually the kid leads him to his family (mother & other siblings) who are equally terrified when they see him, but the kid convinces them the protagonist is OK, that he saved his life.

None of the children are really old enough to remember anything from before the protagonist enter the cryo test, but the mother is. He starts questioning her about what happened while he was asleep. She tells him it was NOT another country invading, it was all internal (society collapsed, food shortages, massive riots).

Later, I think they hear a large group of people approaching the (town? city? area?) where the family has been living, and the family becomes frightened again, saying the enemy is coming back.

He says he will help defend them and asks what the enemy looks like. She looks at him and says "They look just like you." He is African American, the implication being that African-American uprisings are what caused the riots and collapse of society.

This might have been a short story/novella. It probably was a paperback in English published in the USA.

I may be misremembering, maybe she did not tell him they look like you, he just went and got into a position to defend against the enemy, and when they approached, realized they were all African-American.

---

[UQ **author note:** at the time of writing, we submitted an model generated answer (shown below), but the answer was deleted. A follow-up human answer cited our model generated answer and arrived at the same solution (that "The Year of the Quiet Sun" is the book in question). This human answer is now marked as accepted by the question original poster on Stack Exchange. See question link.]

---

**Sample #1 Candidate Answer by o3-pro**

The story you're remembering is Wilson Tucker's short novel / long novella *The Year of the Quiet Sun* (first published in 1970; Hugo- and Nebula-award nominee).

**How it matches the details you gave**

1. How the protagonist gets into the future
   - Instead of cryogenic sleep, Army Captain Addison "Add" Douglass (an African-American Vietnam-era soldier) is one of a small U.S. government team that tests a brand-new time machine in 1978.
   - After a couple of short proving jumps he is sent about twenty years forward, to the year 2000. From Douglass's point of view it is an almost instantaneous "sleep" and awakening, exactly the way you remembered it.

---

2. Abandoned underground facility / gathering supplies
   - The time-gate is in a sub-basement under the University of Chicago. When Douglass materialises in 2000 the building is wrecked and deserted.
   - He salvages rations, water, weapons and medical supplies that the project planners had cached for returning travellers.
3. A devastated city that looks like a war-zone
   - Chicago is in ruins: burned-out cars, bullet-scarred buildings, smashed storefronts. The United States has broken up after years of food shortages, riots and guerrilla fighting.
4. Meeting a frightened child, then the child's family
   - Douglass finds a white boy (David) trapped under rubble, frees him, gives him water and food and gains his trust.
   - David takes him to his widowed mother, Judith, and her two younger children, who are at first terrified because a Black man has suddenly turned up.
5. "It wasn't a foreign invasion – it was internal"
   - Judith explains that what destroyed the country was not an outside enemy but a bitterly escalating series of internal conflicts—racial violence, food riots, martial law—until civil society collapsed.
6. "What do the enemy look like?" – "They look just like you."
   - Judith warns Douglass that a band of marauders is coming back. When he asks how he will recognise them, she says, "They look like you." The attackers are Black militants; Douglass, being Black himself, finally understands why the white family were so afraid of him.
7. Length, publications and era
   - The story is only about 45 000 words (roughly 150–180 paperback pages) so it often appears in anthologies and "best novella" collections from the 1970s onward.
   - First U.S. paperback printing was in 1970 (Ace Books).

Although the mechanism is time travel rather than literal cryogenic sleep, every plot element you recalled—the lone Black protagonist suddenly 20 years out of his own time, the looted underground installation, the ruined city, rescuing a boy, the fearful white family, and the climactic realisation that the "enemy" is people who "look just like you"—comes straight from Wilson Tucker's *The Year of the Quiet Sun*.

**Sample #1 Human Verdict on Candidate Answer: Incorrect**

OP mentions in the comments that it is likely this book; However, part of the answer is incorrect; one user on SE mentioned: "Point 7, at least, is mostly fabrication; the length is wrong (192, 221, or 252) pages, and it was only compiled in a single omnibus in 1997. How much can we trust the rest? You could greatly improve this answer by including direct quotes."

I.3.2  SAMPLE #2: THEORETICAL COMPUTER SCIENCE

**Sample #2 Question**

- **Title:** Complexity to compute the eigenvalue signs of the adjacency matrix
- **Keywords:** cc.complexity-theory, ds.algorithms, linear-algebra, spectral-graph-theory
- **Site:** cstheory
- **Link:** https://cstheory.stackexchange.com/questions/16789/

Let $A$ be the $n \times n$ adjacency matrix of a (non-bipartite) graph. Assume that we are given the amplitudes of its eigenvalues, i.e., $|\lambda_1| = a_1, \ldots, |\lambda_n| = a_n$, and we would like to calculate their signs. Is there a faster way of computing the signs of these eigenvalues, other than recomputing the eigenvalues themselves?

**Sample #2 Candidate Answer by o4-mini**

Below is a summary of why, as far as we know, there is no "magic shortcut" that, given the list of absolute values $\{|\lambda_1|, \ldots, |\lambda_n|\}$, recovers the signs $\text{sign}(\lambda_i)$ in $o(n^3)$ (or $o(n^\omega)$) time without essentially re–doing a spectral computation.

1. The problem really is to decide, for each magnitude $a_i$, whether the corresponding eigenvalue of $A$ is $+a_i$ or $-a_i$. Equivalently one needs an eigenvector $v_i$ for that eigenspace and to check the Rayleigh quotient

$$R(v_i) = \frac{v_i^\mathsf{T} A v_i}{v_i^\mathsf{T} v_i}$$

   whose sign is $\mathrm{sign}(\lambda_i)$. Extracting even one eigenvector to reasonable accuracy costs about as much as finding the eigenvalue itself (e.g., power or Lanczos iterations, or a symmetric QR step).

2. One can rephrase "find all signs" as computing the matrix sign function

$$\mathrm{sign}(A) = A \cdot (A^2)^{-1/2}$$

   so that $\mathrm{sign}(A)V = V \cdot \mathrm{diag}(\mathrm{sign}(\lambda_i))$, and then reading off $\mathrm{sign}(\lambda_i)$ from $V^\mathsf{T}\,\mathrm{sign}(A)V$. But all of the standard ways to compute $\mathrm{sign}(A)$—Newton iterations

$$X_{k+1} = \tfrac{1}{2}\left(X_k + X_k^{-1}\right),$$

   or Padé approximants, or an $LDL^\mathsf{T}$–based inertia algorithm—require $O(n^\omega)$ per matrix multiplication/inversion and at least $O(\log \kappa(A))$ steps for sufficient accuracy. In other words, you pay about the same as for a full spectral decomposition, up to polylog factors.

3. You might hope that knowing $\{|\lambda_i|\}$ cuts down the work. Unfortunately, it merely tells you the spectrum of $A^2$, and $A^2$ and $A$ share eigenvectors but the same 1–dimensional eigenspaces for $+a_i$ and $-a_i$ are indistinguishable via $A^2$ alone. To break the $\pm$ ambiguity you still must apply $A$ to those eigenspaces.

4. From a complexity–theory point of view there is no known reduction that takes "sign–recovery" and solves it in asymptotically less time than any of the standard $O(n^3)$ (or, with fast matrix multiplication, $O(n^\omega)$) eigenvalue–or–inertia algorithms. Nor is there any lower–bound proof that it must cost $\Omega(n^3)$; the barrier is simply that all known numeric or algebraic methods for distinguishing eigenspaces of $\pm a_i$ end up doing essentially the same linear–algebra work as full diagonalization or $LDL^\mathsf{T}$ factorization.

**Conclusion:**

- Best known exact or high–accuracy methods for extracting the individual signs $\mathrm{sign}(\lambda_i)$ run in $O(n^3)$ (using classical Gaussian elimination / QR / divide–and–conquer) or in $O(n^\omega \log(1/\varepsilon))$ if you invoke fast matrix multiplication and iterate to accuracy $\varepsilon$.

- There is no known algorithm that, merely from the list of absolute values $\{|\lambda_i|\}$, peels off the $\pm$ signs in truly sub–cubic time.

### Sample #2 Human Verdict on Candidate Answer: Incorrect

The answer is a long winded way of saying "I don't know." The question asks if a particular problem can be solved faster than the time it takes to find eigendecomposition. The answer neither provides a positive answer/solution, nor does it give a lower bound proving that it is not possible. It merely says "we don't know as far as we can tell."

### I.3.3 SAMPLE #3: MATH OVERFLOW

### Sample #3 Question

- **Title:** The topos for forcing in computability theory
- **Keywords:** ag.algebraic-geometry, ct.category-theory, lo.logic, computability-theory, topos-theory
- **Site:** mathoverflow
- **Link:** https://mathoverflow.net/questions/195794/

My understanding is that forcing (such as Cohen forcing) can be described via a topos. For example this nLab article on forcing describes forcing as a "the topos of sheaves on a suitable site."

My question concerns forcing in computability theory, for example as described in Chapter 3 or these lecture notes of Richard Shore. The idea is that the generics are those which meet all *computable* dense

sets of forcing conditions. (Computable can mean a few things. Often it is taken to mean a $\Sigma_1^0$ set of forcing conditions. Also, usually the forcing posets are countable.) Since there are only countably many such dense sets, such effective generics exist.

> Is there a known/canonical type of topos corresponding to the forcing in computability theory?

Any references would be appreciated.

*FYI:* My background is in computability theory, proof theory, and computable analysis. I know little about topos theory, but I am willing to learn a bit. I am mostly asking this question because I want to compare some ideas I have about effective versions of Solovay forcing with some work by others about the topos corresponding to Solovay forcing. Also, it is always nice to learn new things.

---

### Sample #3 Candiate Answer by o3-pro

Yes – the "effective" versions of the forcing notions that are used in recursion-/computability-theory can be described by exactly the same categorical recipe that one uses for ordinary set–theoretic forcing, the only difference being that one performs the construction

$$\text{"topos of sheaves on the forcing poset } P\text{"}$$

not over the base topos **Sets** but over the **effective topos Eff**. When the forcing site is interpreted internally to **Eff** the generic object that appears in the resulting sheaf topos meets precisely the computably (i.e. r.e.) given dense sets that the recursion-theorists want it to meet. In this sense $\text{Eff}[P]$ $(= \mathbf{Sh}_{\text{Eff}}(P))$ is the "correct / canonical topos for effective forcing".

### 1. Reminder: ordinary forcing in topos language

Start with a poset (or, more generally, a small category) $P$. Internally to the base topos $E$ (usually **Sets**) one equips $P$ with the coverage $J$ generated by the usual notion of a dense subset: for $p \in P$, a sieve $S \subseteq\downarrow p$ is covering when every $q \leq p$ has a refinement $r \leq q$ in $S$. The topos of sheaves $\mathbf{Sh}_E(P, J)$ comes with a geometric morphism

$$g : \mathbf{Sh}_E(P) \to E$$

whose inverse–image $g^*$ sends a set $X$ to the constant sheaf $\Delta X$.

The Yoneda embedding $y : P \to \mathbf{Sh}_E(P)$ classifies a "generic filter"

$$G \subseteq \Delta P$$

which meets all internally $J$-dense subobjects of $\Delta P$, and every map from 1 to $\Omega$ in $\mathbf{Sh}_E(P)$ is determined by which members of $G$ it contains. When $E = \mathbf{Sets}$ the externally visible dense subobjects are just the (set-theoretically) given dense subsets, so $G$ is a Cohen-generic, etc.

### 2. What changes for recursion-theoretic forcing?

In recursion theory one keeps the forcing poset $P$ countable and one asks that the generic meet only the EFFECTIVELY (i.e. recursively enumerable) dense subsets of $P$. Since there are only countably many of those, the usual diagonalisation guarantees existence of such a filter in $\omega$.

The categorical way to implement the same restriction is simply to change the base topos from **Sets** to **Eff**, Martin Hyland's effective topos.

**Facts you need about Eff.**

- The global points $|N|$ of the natural-numbers object $N$ in **Eff** form the ordinary set $\mathbb{N}$, and subobjects of $N$ correspond to r.e. subsets of $\mathbb{N}$ (decidable subobjects correspond to recursive sets, etc.). Consequently "quantification over all subobjects of $N$" inside **Eff** is exactly "quantification over all r.e. subsets of $\mathbb{N}$" outside.

- Every partial recursive function gives a morphism in **Eff** and vice versa; internally **Eff** satisfies Church's Thesis + Markov's Principle.

Because of the first item, an "internally dense" $S \subseteq \Delta P$ in **Eff** is, from the external point of view, an r.e. dense set of forcing conditions. So if we perform the forcing construction internally to **Eff** we obtain

$$\mathrm{Eff}[P] := \mathbf{Sh}_{\mathrm{Eff}}(P, J),$$

and the Yoneda object

$$G \subseteq \Delta P \quad (\text{in Eff}[P])$$

is a filter meeting every EFFECTIVELY given dense set; i.e. $G$ is exactly an "effective generic" in the recursion-theorist's sense.

Formally there is a commuting square of geometric morphisms:

$$\begin{array}{ccc} \mathrm{Eff}[P] & \longrightarrow & \mathbf{Sets}[P] \\ \downarrow & & \downarrow \\ \mathrm{Eff} & \longrightarrow & \mathbf{Sets} \end{array}$$

and the composite point $\mathrm{Eff}[P] \to \mathbf{Sets}$ interprets $G$ as an $\omega$-sequence that meets all r.e. dense sets.

### 3. Why this deserves to be called "canonical"

Among all toposes whose points are effective generics, $\mathrm{Eff}[P]$

1. is obtained from the usual forcing recipe without ad-hoc changes,
2. is initial among all toposes $T$ equipped with a geometric morphism $T \to \mathrm{Eff}$ carrying a generic filter, and
3. specialises to the ordinary forcing topos when the base topos is taken to be **Sets** instead of **Eff**.

Hence it plays for recursion-theoretic forcing exactly the rôle that the usual forcing topos plays for set-theoretic forcing.

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

Applications to recursion-theoretic forcing (Cohen, Mathias, Solovay, etc.) can be found scattered through the realizability literature; the papers above give enough details to translate any classical forcing argument into the internal language of **Eff**.

### 5. Take-away

So the answer is:

Yes. Take your favourite computable forcing poset $P$, view it as an internal poset in the effective topos **Eff**, and form the topos of sheaves $\mathrm{Eff}[P]$. The generic object in $\mathrm{Eff}[P]$ is exactly an effective generic filter – it meets every computably (r.e.) dense set, and no stronger assumption is built in. $\mathrm{Eff}[P]$ is therefore the canonical topos corresponding to the forcing notions that appear in computability theory.

**Sample #3 Human Verdict on Candidate Answer: Incorrect**

The cited reference

> J. van Oosten & A. Pitts, *A note on effective sheaf models*, Proc. Amer. Math. Soc.
> 120 (1994), 829–836

is incorrect. This exact reference does not exist in the Proceedings of the American Mathematical Society for 1994 (volume 120, issue 3): https://www.ams.org/journals/proc/1994-120-03/.

### I.3.4 SAMPLE #4: THEORETICAL COMPUTER SCIENCE

**Sample #4 Question**

- **Title:** DPLL and Lovász Local Lemma
- **Keywords:** cc.complexity-theory, ds.algorithms, sat
- **Site:** cstheory
- **Link:** https://cstheory.stackexchange.com/questions/7720

Let $\varphi$ be a CNF formula. Suppose that each of $\varphi$'s clauses consist of exactly $t$ literals (and, moreover, all literals within one particular clause correspond to different variables). It is well known that if every clause has less than $2^t/e$ clauses that share variables with it, then $\varphi$ is satisfiable (let us call such formulae *easy*). Satisfiability can be proved easily using Lovász local lemma [1]. Moreover, using a recent result [2] by Moser and Tardos one can show that one of the satisfying assignments can be found in polynomial expected time using the following very simple procedure:

- Pick a random assignment.
- While there exists an unsatisfied clause, resample all its variables.

On the other hand, most of modern SAT solvers are DPLL-based [3]. This means that they try to find a satisfying assignment using brute force with two simple prunings:

- If a formula contains a clause with one literal, then we can fix it.
- If one variable occurs in a formula only with (or without) negation, then we can fix it.

**The question: Is it true that a version of DPLL that splits on random variables finds a satisfying assignment of any easy formula in polynomial expected time?**

[1]: https://en.wikipedia.org/wiki/Lovasz_local_lemma

[2]: http://arxiv.org/abs/0903.0544

[3]: http://en.wikipedia.org/wiki/DPLL_algorithm

**Sample #4 Candiate Answer by o3**

**Short answer**
Nobody knows. At the moment there is neither

- a proof that a (pure) DPLL procedure that always branches on a randomly chosen variable finishes in polynomial expected time on every instance that satisfies the Lovász-Local-Lemma (LLL) condition, nor
- an instance that satisfies this LLL condition for which the same DPLL procedure can be shown to need super-polynomial (let alone exponential) expected running time.

So the question you ask is open.

**Why we do not have an upper bound**
The Moser–Tardos algorithm that turns the LLL into an efficient constructive result relies on "resampling" the variables of a clause **without ever exploring assignments that are inconsistent with the current partial assignment**. A DPLL tree search behaves quite differently: whenever it guesses a value for a variable it explores both possibilities, and the whole sub-tree corresponding to the first value has to be proved unsatisfiable before the procedure can back-track and try the second value. Proving a polynomial bound for DPLL therefore amounts to proving that every tree–resolution refutation of an LLL instance

that is inconsistent with the target (unknown) satisfying assignment is short; at present we do not know how to do that.

**Why we do not have a lower bound**

To show a lower bound one has to exhibit a *satisfiable* formula $F$ that

1. satisfies the LLL condition
   $(p = 2^{-t}, \quad d < 2^t/e - 1),$
2. forces a random-branching DPLL algorithm to construct an exponentially large search tree *before* it stumbles upon a satisfying assignment.

Lower bounds for DPLL are usually obtained by exhibiting formulas whose every *unsatisfiable* sub-instance requires an exponentially long tree-resolution proof (Tseitin formulas, pigeon-hole principles, etc.). All such formulas violate the LLL condition, so one has to hide one of them inside a satisfiable "wrapper" while keeping the overall clause-dependency below the $2^t/e$ threshold. No one has succeeded in doing this so far. On the other hand, nothing rules it out either.

**Related partial results**

- For many *random k*-CNF distributions with clause density well below the satisfiability threshold (far below the LLL bound) simple DPLL variants are known to succeed in expected polynomial time, but that says nothing about worst-case LLL instances.
- Exponential lower bounds are known for DPLL on *satisfiable* formulas (Beame, Kautz & Sabharwal, 2004; Paturi & Pudlák, 2005, and later work) in which every variable appears only a constant number of times. Unfortunately the constants involved are too large for the LLL condition to hold.
- If additional inference rules (e.g., clause learning with certain restart policies) are allowed, one can simulate the Moser–Tardos resampling algorithm inside a SAT solver and obtain a polynomial bound; but that is no longer the plain DPLL framework you asked about.

**Conclusion**

Whether plain DPLL with random variable selection is always fast on LLL-satisfiable formulas is an interesting and, at present, open research problem. Neither a proof nor a counterexample is known.

---

**Sample #4 Human Verdict on Candidate Answer: Incorrect**

The model's response doesn't say whether OP's conjecture is true or false. The part of its response regarding "Why we do not have a lower bound" seems reasonable, but doesn't say anything non-trivial.

For the part of its response regarding "Related partial results", it says that "For many *random k*-CNF distributions with clause density well below the satisfiability threshold (far below the LLL bound) simple DPLL variants are known to succeed in expected polynomial time, but that says nothing about worst-case LLL instances." On one hand, this is a plausible result since in the limit where each clause is independent of every other clause in the CNF, I'd guess that DPLL returns a satisfying assignment in polynomial time. On the other hand, the details of what exact class of CNFs such a hypothetical result applies to matters (in particular, what is "well below the threshold of LLL"), and the model's response does not cite any references. Also, the model is probably right that the instances which would be hard to prove OP's conjecture for are those where the number of other clauses each clause depends on is at the LLL threshold.

Also in the "related partial results" part of the model's response, it says that "Exponential lower bounds are known for DPLL on *satisfiable* formulas (Beame, Kautz & Sabharwal, 2004; Paturi & Pudlák, 2005, and later work) in which every variable appears only a constant number of times. Unfortunately the constants involved are too large for the LLL condition to hold." The source by Beame, Kautz, Sabharwal is real: https://arxiv.org/pdf/1107.0044, but it seems to me that this paper is a more appropriate citation for the model's claim: https://homes.cs.washington.edu/~beame/papers/stoc2plusp.pdf (which is cited in Beame, Kautz, Sabharwal). The model's last comment about "constants involved are too large for the LLL condition to hold" is hard to verify.

## I.4 SAMPLE ANSWERS PASSING UQ-VALIDATOR AND HUMAN-VERIFIED AS CORRECT

This section is referenced by Section 5 and provides a sample of question-answer pairs that passed our UQ-Validator (144 question-answer pairs in total, recall Section 5), and **the answers are ruled as correct under human verification—i.e., the model is considered to have solved these questions** (10 of the validated 90 questions in total).

### I.4.1 SAMPLE #1: PHYSICS

---

**Sample #1 Question**

- **Title:** Radiative equilibrium in orbit of a black hole
- **Keywords:** general-relativity, thermodynamics, time-dilation, cosmic-microwave-background
- **Site:** physics
- **Link:** https://physics.stackexchange.com/questions/246203/

According to *Life under a black sun* [1], Miller's planet from *Interstellar*, with a time dilation factor of 60,000, should be heated to around $890°C$ by blue-shifted cosmic background radiation.

How they arrive at that number, however, seems to me a little opaque.

As the article describes, there are two major effects to consider: gravitational blueshifting, and blue- and redshifts due to the planet's orbital motion.

Calculating the purely gravitational effects seems straightforward (although I admit I may still be missing something); given that radiative power is proportional to $T^4$, and power should scale linearly with the time dilation factor, the apparent CMB temperature should be $2.7 \text{ K} \times 60{,}000^{1/4} = 42.26 \text{ K}$. Considering that a cold black hole occupies part of the sky, the equilibrium temperature of the planet should be slightly lower. That's clearly a long way from $890°C$!

It appears, then, that the majority of the heating must be a result of the circular motion of the planet in orbit. Now, it seems fairly obvious that getting precise answers will require numerical simulation, but it should be possible to at least get a close order-of-magnitude estimate based on a model of a planet moving at constant velocity through a background of the temperature calculated from gravitational effects alone. Unfortunately, though, the article doesn't quote speeds, and I haven't been able to figure out how to calculate the relevant velocities for a planet in a low orbit around a rotating black hole.

So, can anybody help me fill in the blanks? If I start with a black hole of a given mass and angular momentum, and a planet in a stable circular orbit at some given radius, how do I get to an estimate of equilibrium temperature?

[1]: http://arxiv.org/pdf/1601.02897v1.pdf

---

**Sample #1 Candiate Answer by o4-mini**

Here is a "cook-book" for getting the equilibrium temperature that a little planet in circular orbit around a Kerr (or Schwarzschild) black hole will settle to, when the only "heater" is the cosmic microwave background at 2.7K as seen from infinity. In outline you do precisely the same thing you would do in special relativity (blue-shift plus Doppler beaming of an otherwise isotropic bath) except that now you must combine

- the gravitational red–(or blue–)shift from infinity into your orbit
- the aberration / Doppler–beaming from the fact that the planet is moving
- the fact that part of the sky is covered by the black-hole shadow
- the usual black–body equilibrium condition

Below I give both the full GR recipe (numerical ray-tracing) and a much simpler analytic approximation (no lensing, no frame-dragging) that already captures the correct scalings.

1) FULL GENERAL-RELATIVISTIC METHOD

1. **Choose your black hole**
   – Mass $M$, spin $a$. Write down the Kerr metric in Boyer–Lindquist coordinates $(t, r, \theta, \phi)$.

---

2. **Pick your orbit**
   – A prograde circular orbit in the equatorial plane $\theta = \pi/2$ at radius $r$. From Bardeen–Press–Teukolsky (1972) one has the conserved energy per unit rest mass

$$E = -\frac{p_t}{m} = \frac{r^{3/2} - 2M\,r^{1/2} + a\,M^{1/2}}{r^{3/4}\,\sqrt{r^{3/2} - 3M\,r^{1/2} + 2a\,M^{1/2}}},$$

   and the angular velocity

$$\Omega = \frac{d\phi}{dt} = \frac{M^{1/2}}{r^{3/2} + a\,M^{1/2}}.$$

   – From these you build the four–velocity of the planet

$$u^\mu = \left(u^t,\,0,\,0,\,u^\phi\right), \quad u^t = \frac{dt}{d\tau} = \frac{1}{\sqrt{-\,g_{tt} - 2\,\Omega\,g_{t\phi} - \Omega^2\,g_{\phi\phi}}}, \quad u^\phi = \Omega\,u^t.$$

3. **Set up a local tetrad**
   – Let $e_{(0)}{}^\mu = u^\mu$. Choose three mutually orthonormal spacelike vectors $e_{(1)}, e_{(2)}, e_{(3)}$ that span the local rest space of the planet. One convenient choice is to start from the ZAMO (zero angular–momentum observer) tetrad and then boost it by the local orbital velocity.

4. **Shoot rays into the sky**
   – Parameterize the planet's *local* sky by two angles $(\vartheta, \varphi)$. In the tetrad frame a photon leaving the planet has momentum

$$p^{(a)} = h\nu_\infty\left(1,\,\sin\vartheta\cos\varphi,\,\sin\vartheta\sin\varphi,\,\cos\vartheta\right).$$

   – Convert back to coordinate components

$$p^\mu = e_{(a)}{}^\mu\,p^{(a)}.$$

   – Integrate the null geodesic *backwards* until it either hits the hole (so that direction is part of the "shadow") or goes out to infinity.

5. **Compute the redshift factor**
   – Along each geodesic you have two conserved quantities: the energy at infinity $-p_t$ and the photon's local inner product with the planet's four–velocity

$$\nu_{\text{obs}} = -\,p_\mu u^\mu, \quad \nu_\infty = -\,p_t$$

   – Hence the usual frequency shift

$$g(\vartheta, \varphi) = \frac{\nu_{\text{obs}}}{\nu_\infty} = \frac{-\,p_\mu u^\mu}{-\,p_t}.$$

6. **Use Liouville's theorem**
   – A black–body at infinity has specific intensity $I_\nu^\infty = B_\nu[T_{\text{CMB}}]$. Along a geodesic $I_\nu/\nu^3$ is conserved. Hence at the planet

$$I_{\nu,\text{loc}}(\vartheta, \varphi) = g^3(\vartheta, \varphi)\,B_\nu[T_{\text{CMB}}]\left(\nu/g\right),$$

   – and integrating over frequency shows that the *bolometric* intensity at the planet is

$$I_{\text{loc}}(\vartheta, \varphi) = g^4(\vartheta, \varphi)\,I_\infty = g^4(\vartheta, \varphi)\,\frac{\sigma\,T_{\text{CMB}}^4}{\pi}.$$

7. **Compute the absorbed flux**
   – The *total* power falling on the planet is the cross-section $\pi R^2$ times the integral of the incoming intensity over *all* directions that clear the hole's shadow.

$$P_{\text{abs}} = \pi R^2 \int_{\text{sky}-\text{shadow}} I_{\text{loc}}(\vartheta, \varphi)\,d\Omega = \pi R^2\,\frac{\sigma\,T_{\text{CMB}}^4}{\pi}\int_{\text{visible}} g^4\,d\Omega.$$

   – Spread that power over the *entire* surface $4\pi R^2$ to get the *average* absorbed flux per unit area:

$$F_{\text{abs}} = \frac{P_{\text{abs}}}{4\pi R^2} = \frac{\sigma\,T_{\text{CMB}}^4}{4\pi}\int_{\text{visible}} g^4(\vartheta, \varphi)\,d\Omega.$$

8. **Impose black–body balance**
   – In steady state the planet must reradiate exactly $F_{\text{abs}}$ in its own local black–body spectrum, so

$$\sigma\,T_{\text{eq}}^4 = F_{\text{abs}} = \frac{\sigma\,T_{\text{CMB}}^4}{4\pi}\int_{\text{visible}} g^4\,d\Omega,$$

   – i.e.

$$T_{\text{eq}} = T_{\text{CMB}}\left\{\frac{1}{4\pi}\int_{\text{visible}} \left[g(\vartheta, \varphi)\right]^4\,d\Omega\right\}^{1/4}.$$

That integral has to be done numerically once you have the geodesics and the redshift factor $g$. You also have to subtract out the solid angle of the black-hole shadow.

## 2) A SIMPLE ANALYTIC APPROXIMATION

If you *ignore* all lensing, frame-dragging, and the shadow (i.e. you pretend the BH is just a Newtonian point mass plus special-relativistic time-dilation), then

- **Gravitational time-dilation** at radius $r$ gives a static redshift factor

$$g_{\text{grav}} = \frac{1}{\sqrt{1 - \dfrac{2GM}{rc^2}}}$$

so that a static observer at $r$ would see the CMB temperature $T_{\text{CMB}}\, g_{\text{grav}}$.

- **Orbital Doppler–beaming.** The planet moves at

$$v = \frac{\sqrt{GM/r}}{\sqrt{1 - 2GM/(rc^2)}}, \quad \beta = \frac{v}{c}, \quad \gamma = \frac{1}{\sqrt{1 - \beta^2}}.$$

In special relativity an isotropic bath at temperature $T$ becomes angle–dependent

$$T_{\text{local}}(\mu) = T\,\gamma\,(1 + \beta\,\mu), \quad \mu = \cos(\text{angle w.r.t. velocity}).$$

One shows easily

$$\frac{1}{4\pi}\int_{4\pi} \left[\gamma(1 + \beta\mu)\right]^4 d\Omega = \gamma^4 \left[1 + 2\beta^2 + 0.2\,\beta^4\right].$$

Putting them together in the same spirit (but still ignoring the shadow) gives the *analytic* estimate

$$T_{\text{eq}} = T_{\text{CMB}}\, g_{\text{grav}}\, \gamma\, \left[1 + 2\beta^2 + 0.2\,\beta^4\right]^{1/4}.$$

One can check quickly that for mildly relativistic orbits this reproduces the known fact that the beaming correction is only of order unity, whereas in the very deep potential the factor $g_{\text{grav}}$ can become enormous.

### HOW ONE GETS $\sim 890\,^\circ\text{C}$ FOR "MILLER'S PLANET"

In the *Interstellar* scenario the authors have chosen parameters (near–maximal spin, an orbit just above the prograde ISCO) so that

- the *total* gravitational time-dilation *alone* would be a factor of $F \sim 6 \times 10^4$, and they treat that as boosting the *total* power by the same factor
- since power $\propto T^4$, they set $T^4 \propto 6 \times 10^4$, i.e. $T \sim 2.7\,\text{K} \times (6 \times 10^4)^{1/4} \approx 42\,\text{K}$ from gravity alone
- then they fold in a *very large* orbital Doppler-beaming (because that orbit is extremely close in and very fast), which raises the temperature by another factor of a few tens
- in the end one finds $T \approx 1163\,\text{K}$ (about $890\,^\circ\text{C}$) once you do the full angular-integral including the Doppler factor

If you want to reproduce their exact number you must

1. pick the same Kerr spin $a$,
2. solve for the prograde circular orbit radius that gives the $6 \times 10^4$ time-dilation they quote,
3. compute $\beta$ and $\gamma$ at that orbit,
4. do the integral $\left[\gamma(1 + \beta\mu)\right]^4$ over the *visible* sky (subtracting the hole's shadow!),
5. take the one-quarter power, and
6. multiply by 2.7K.

In practice you can do 4.–6. either

- *numerically* with a little ray-tracer in the Kerr metric (this is what e.g. GYOTO or geokerr will do),
- or *analytically* to leading order in $\beta$ if $\beta \ll 1$.

Either way, the *template* is always the same:

- Find the local sky distribution of CMB photons via $g(\vartheta, \varphi) = -p_\mu u^\mu / p_t$.
- Use $I_{\text{loc}} = g^4\, I_\infty$.
- Integrate $g^4$ over the non–shadowed sky.
- Set $\sigma T_{\text{eq}}^4 = (\sigma T_{\text{CMB}}^4 / 4\pi) \int g^4 d\Omega$.

That procedure will let you *predict* the equilibrium temperature for *any* $M, a, r$. Once the numbers are in hand it quickly reproduces numbers of order 1000 K for the extreme orbits needed to get a 1 hr $\to 60\,000$ hr dilation factor.

> **Sample #1 Human Verdict on Candidate Answer: Correct**
>
> **[From Question Original Poster on Physics]**
>
> That looks great! I think an ideal answer would replace "→ Boosts temperature by more factors" with an actual working out of the Doppler beaming factor, but as-is, I would accept that as a StackExchange answer. It's got enough detail that I *could* work it out myself.

### I.4.2   SAMPLE #2: MATHEMATICS

> **Sample #2 Question**
>
> - **Title:** A difficult integral for the Chern number
> - **Keywords:** integration, multivariable-calculus, differential-geometry, definite-integrals, characteristic-classes
> - **Site:** math
> - **Link:** https://math.stackexchange.com/questions/4495174
>
> The integral
>
> $$I(m) = \frac{1}{4\pi} \int_{-\pi}^{\pi} dx \int_{-\pi}^{\pi} dy \frac{m\cos(x)\cos(y) - \cos x - \cos y}{\left(\sin^2 x + \sin^2 y + (m - \cos x - \cos y)^2\right)^{3/2}}$$
>
> gives the Chern number of a certain vector bundle [1] over a torus. It can be shown using the theory of characteristic classes that
>
> $$I(m) = \frac{\operatorname{sign}(m-2) + \operatorname{sign}(m+2)}{2} - \operatorname{sign}(m) = \begin{cases} 1 & -2 < m < 0 \\ -1 & 0 < m < 2 \\ 0 & \text{otherwise} \end{cases}.$$
>
> Is there any way to evaluate this integral directly (i.e. without making use of methods from differential geometry) to obtain the above result?
>
> I should mention that the above integral can be written as $(1/4\pi$ times) the solid angle subtended from the origin of the unit vector $\hat{\mathbf{n}}$,
>
> $$I(m) = \frac{1}{4\pi} \int_{-\pi}^{\pi} dx \int_{-\pi}^{\pi} dy \, \hat{\mathbf{n}} \cdot (\partial_x \hat{\mathbf{n}} \times \partial_y \hat{\mathbf{n}}),$$
>
> where $\mathbf{n}(m) = (\sin x, \sin y, m - \cos x - \cos y)$. While this form makes it very straightforward to evaluate $I(m)$, I am interested in whether there is a way to compute this integral using more standard techniques.
>
> [1] B. Bernevig, *Topological Insulators and Topological Superconductors*, Chapter 8.

> **Sample #2 Candidate Answer by o3-pro**
>
> A straightforward way to obtain the value of
>
> $$I(m) = \frac{1}{4\pi} \int_{-\pi}^{\pi} dx \int_{-\pi}^{\pi} dy \, \frac{m\cos x \cos y - \cos x - \cos y}{\left[\sin^2 x + \sin^2 y + (m - \cos x - \cos y)^2\right]^{3/2}} \tag{1}$$
>
> is to look at it as an ordinary flux–integral and to keep track of the places where the denominator can vanish. Everything can be done with nothing more than multivariable calculus, the divergence theorem and the elementary identity
>
> $$\operatorname{div} \frac{\mathbf{r}}{|\mathbf{r}|^3} = 4\pi \, \delta^3(\mathbf{r}). \tag{2}$$
>
> No mention of Chern classes is necessary.

## 1. RE-INTERPRET THE INTEGRAND AS A FLUX DENSITY

Put

$$\mathbf{n}(x, y, m) = (\sin x, \sin y, m - \cos x - \cos y), \quad \mathbf{R}(x, y, m) = \frac{\mathbf{n}}{|\mathbf{n}|^3}.$$

Because

$$\mathbf{R} \cdot (\partial_x \mathbf{n} \times \partial_y \mathbf{n}) = \frac{\mathbf{n} \cdot (\partial_x \mathbf{n} \times \partial_y \mathbf{n})}{|\mathbf{n}|^3}, \tag{3}$$

(1) can be rewritten

$$I(m) = \frac{1}{4\pi} \iint_T \mathbf{R} \cdot (\partial_x \mathbf{n} \times \partial_y \mathbf{n}) \, \mathrm{d}x \, \mathrm{d}y \tag{4}$$

where $T = [-\pi, \pi] \times [-\pi, \pi]$ is our fundamental rectangle. Equation (4) is the flux of the Coulomb field $\mathbf{R}$ through the two–dimensional surface $\Sigma_m$ obtained by keeping $m$ fixed and letting $(x, y)$ run over $T$:

$$\Sigma_m : (x, y) \mapsto \mathbf{n}(x, y, m) \subset \mathbb{R}^3.$$

So

$$I(m) = \frac{\text{total Coulomb flux through } \Sigma_m}{4\pi}. \tag{5}$$

## 2. HOW DOES $I(m)$ CHANGE WHEN $m$ VARIES?

Let $m_1 < m_2$ and let

$$D = T \times [m_1, m_2] \subset \mathbb{R}^3_{(x,y,m)}.$$

The boundary of $D$ is the union of $\Sigma_{m_2}$, $\Sigma_{m_1}$ and the four side faces $x = \pm\pi$, $y = \pm\pi$. Because the integrand is $2\pi$–periodic in $x$ and $y$, the contributions of the four side faces cancel pairwise, hence

$$I(m_2) - I(m_1) = \frac{1}{4\pi} \iint_{\partial D} \mathbf{R} \cdot \mathrm{d}\mathbf{S} = \frac{1}{4\pi} \iiint_D \operatorname{div} \mathbf{R} \, \mathrm{d}V \quad \text{(Gauss)}. \tag{6}$$

With (2) we get

$$I(m_2) - I(m_1) = \iiint_D \delta^3(\mathbf{n}) \, \det\left(\frac{\partial \mathbf{n}}{\partial(x, y, m)}\right) \mathrm{d}x \, \mathrm{d}y \, \mathrm{d}m. \tag{7}$$

Thus $I(m)$ can change *only* when $\mathbf{n}(x, y, m) = 0$ has a solution. Those solutions are easy to list:

$$\sin x = \sin y = 0, \quad m = \cos x + \cos y. \tag{8}$$

Because $\sin x = 0 \Rightarrow x \equiv 0, \pi \pmod{2\pi}$, and the same for $y$, the four possibilities are

$$(x, y, m) : (0, 0, 2), \quad (\pi, \pi, -2), \quad (0, \pi, 0), \quad (\pi, 0, 0). \tag{9}$$

Hence $I(m)$ can jump only when $m$ passes $-2, 0$ or $2$.

## 3. SIZE AND SIGN OF EVERY JUMP

The size of the jump produced by one zero of $\mathbf{n}$ is the sign of the Jacobian

$$J = \det\left[\frac{\partial \mathbf{n}}{\partial(x, y, m)}\right] = \cos x \cdot \cos y \tag{10}$$

evaluated at that zero, because $\delta^3(\mathbf{n})$ integrates to $1/|J|$ and (7) keeps the sign of $J$.

- $(\pi, \pi, -2)$: $\cos x \cos y = (+)(+) = +1 \Rightarrow$ jump $+1$ when $m$ crosses $-2$
- $(0, 0, 2)$: $(+)(+) = +1 \Rightarrow$ jump $+1$ when $m$ crosses $2$
- $(0, \pi, 0)$: $(+)(-) = -1 \Rightarrow$ jump $-1$
- $(\pi, 0, 0)$: $(-)(+) = -1 \Rightarrow$ jump $-1$

The two points with $m = 0$ contribute together a jump $-2$.

## 4. ASSEMBLE THE RESULT

Start with very negative $m$. For $m < -2$ the vector $\mathbf{n}$ points essentially in the negative $z$–direction for all $(x, y)$; the surface $\Sigma_m$ lies completely in the lower half–space and encloses no charge, so $I(m) = 0$ for $m < -2$.

Now raise $m$ and add the jumps just found:

- Crossing $m = -2$: $I$ jumps by $+1 \Rightarrow I = +1$ for $-2 < m < 0$
- Crossing $m = 0$: $I$ jumps by $-2 \Rightarrow I = -1$ for $0 < m < 2$
- Crossing $m = 2$: $I$ jumps by $+1 \Rightarrow I = 0$ for $m > 2$

## 5. FINAL VALUE

Therefore

$$I(m) = \begin{cases} 1 & \text{for } -2 < m < 0, \\ -1 & \text{for } 0 < m < 2, \\ 0 & \text{otherwise.} \end{cases} \tag{11}$$

which is exactly the piecewise constant function obtained from the abstract characteristic–class argument,

$$I(m) = \frac{\text{sign}(m - 2) + \text{sign}(m + 2)}{2} - \text{sign}(m). \tag{12}$$

All we have used is the interpretation of the integrand as a flux, the divergence theorem, the elementary identity (2) for the Coulomb field, and a local Jacobian calculation at the finitely many points where the integrand can become singular.

**Sample #2 Human Verdict on Candidate Answer: Correct (Borderline)**

**[From a Ph.D. candidate in mathematics and a co-author of this paper]**

The candidate answer aligns with and expands upon the second approach discussed in the original question, effectively arriving at the correct solution.

Although this answer is correct and much more standard than invoking results in differential geometry, I am not sure whether this satisfies the author's preference for an even more standard method for calculating the integral directly.

### I.4.3  SAMPLE #3: MATHEMATICS / MATH OVERFLOW

## Sample #3 Question

- **Title:** Constructing an infinite chain of subfields of 'hyper' algebraic numbers?
- **Keywords:** real-analysis, field-theory, transcendental-numbers
- **Site:** math
- **Link:** https://math.stackexchange.com/questions/3014759/

This has now been cross posted [1] to Math Overflow.

Let $F$ be a subset of $\mathbb{R}$ and let $S_F$ denote the set of values which satisfy some generalized polynomial whose exponents and coefficients are drawn from $F$. That is, we let $S_F$ denote

$$\left\{ x \in \mathbb{R} : 0 = \sum_{i=1}^{n} a_i x^{e_i} \ : \ e_i \in F \text{ distinct}, \ a_i \in F \text{ non-zero}, \ n \in \mathbb{N} \right\}.$$

Then $S_\mathbb{Q}$ is the set of algebraic real numbers and we start to see the beginnings of a chain:

$$\mathbb{Q} \subsetneq S_\mathbb{Q} \subsetneq S_{S_\mathbb{Q}}.$$

**Main Question.**  Does this chain continue forever? That is, we let $A_0 = \mathbb{Q}$ and let $A_{n+1} = S_{A_n}$. Is it the case that $A_n \subsetneq A_{n+1}$ for all $n \in \mathbb{N}$?

**Other curiosities.**

- Is $A_i$ always a field? Perhaps, the argument is analogous to this [2]. Or maybe this is just the case in a more general setting: Is it the case that $F \subset \mathbb{R}$, a field, implies that $S_F$ is a field?
- Is it possible to see that $e \notin \bigcup A_i$? Perhaps this is just a tweaking of the LW Theorem [3].

[1]: https://mathoverflow.net/questions/319167/constructing-an-infinite-chain-of-subsets-of-hyper-algebraic-numbers

[2]: https://math.stackexchange.com/questions/331017/enlightening-proof-that-the-algebraic-numbers-form-a-field

[3]: https://en.wikipedia.org/wiki/Lindemann-Weierstrass_theorem

## Sample #3 Candiate Answer by o3-pro

Throughout let
$$S_F = \big\{ x \in \mathbb{R} : 0 = a_1 x^{e_1} + \cdots + a_n x^{e_n} \text{ for some } n \geq 1,$$
$$\text{with pairwise different exponents } e_i \in F,$$
$$\text{and nonzero coefficients } a_i \in F \big\}.$$

Put $A_0 = \mathbb{Q}$ and $A_{n+1} = S(A_n)$. The question is whether the inclusions $A_0 \subsetneq A_1 \subsetneq A_2 \subsetneq \ldots$ stay proper for every finite $n$, whether every $A_n$ is a field, and whether numbers such as $e$ ever lie in one of the $A_n$.

**1. The first two steps.**  $A_1$ is exactly the field of algebraic real numbers $\overline{\mathbb{Q}}$.

*Proof.* If the exponents that occur are rational, write them with common denominator $m$. Putting $y = x^{1/m}$ turns the given equation into an ordinary polynomial over $\mathbb{Q}$, so $x$ is algebraic. Conversely, every algebraic number satisfies a usual polynomial, hence belongs to $S(\mathbb{Q})$.

$A_2$ is strictly larger than $A_1$. Indeed $\sqrt{2} \in A_1$ and the equation $x^{\sqrt{2}} - 2 = 0$ (coefficients $2, -1$, exponent $\sqrt{2}$) shows $2^{1/\sqrt{2}} \in A_2$. By the Gelfond–Schneider theorem $2^{1/\sqrt{2}}$ is transcendental, so $A_1 \subsetneq A_2$.

**2. Countability.**  If $F$ is countable then the collection of "generalised polynomials" that may be used is also countable, every such equation has only finitely many real roots, and therefore $S(F)$ is again countable. Hence every $A_n$ is countable. Thus the chain can never exhaust $\mathbb{R}$, but it might conceivably stabilise at some countable stage.

**3. Does the chain certainly go on for ever? (main question).**  At present this is not known. What is needed is, given a countable field $F \subseteq \mathbb{R}$, to guarantee that $S(F) \setminus F$ is nonempty. For $F = \mathbb{Q}$

the preceding paragraph supplies such an element; for general countable $F$ no proof is known. Conversely, nobody has been able to construct a countable field $F$ with $S(F) = F$, so the problem is open: we do not know whether the chain $A_0 \subsetneq A_1 \subsetneq A_2 \subsetneq \ldots$ is infinite, or whether it stabilises after finitely many steps.

A decisive positive answer (the chain really is infinite) would follow from the still unproved Schanuel conjecture, because Schanuel would imply for instance that for every $\alpha \notin \mathbb{Q}$ the number $2^{1/\alpha}$ cannot satisfy any equation of the required form whose exponents already lie in the field generated by $\alpha$. All we can say unconditionally is that the inclusion $A_1 \subsetneq A_2$ is strict; what happens afterwards is open.

**4. Are the $A_n$ necessarily fields?** $A_1$ certainly is, but in general $S(F)$ need not be closed under addition. For example take

$$F = \mathbb{Q}(2^{\sqrt{2}});$$

then $2^{\sqrt{2}} \in F$ and the same equation as above shows $2^{1/\sqrt{2}} \in S(F)$. There is no known way of producing from equations with exponents and coefficients in $F$ a relation that forces $2^{\sqrt{2}} + 2^{1/\sqrt{2}}$ to belong to $S(F)$. In fact it is believed (again on the strength of Schanuel) that $S(F)$ will very rarely be closed under $+$. Consequently it is not known whether every $A_n$ is a field; we only know this for $n = 0$ and $n = 1$.

**5. The number $e$ (and $\pi$, $\log 2$, ...).** It is also unknown whether $e$ belongs to any $A_n$. All that is known is that $e$ is not algebraic, so $e \notin A_1$. Deciding whether $e$ lies in $A_2$ already requires settling an outstanding problem in exponential Diophantine geometry (more precisely, whether $e$ is "exponentially algebraic" over $\overline{\mathbb{Q}}$).

**Summary.**

- $A_0 \subsetneq A_1 = \overline{\mathbb{Q}}$;
- $A_1 \subsetneq A_2$ (witness $2^{1/\sqrt{2}}$);
- for $n \geq 2$ it is an open problem whether $A_n \subsetneq A_{n+1}$;
- it is not known whether each $A_n$ is a field;
- it is not known whether $e \in \bigcup_{n < \omega} A_n$.

## Sample #3 Human Verdict on Candidate Answer: Correct (Borderline)

**[From Question Original Poster on Mathematics]**

Absolutely — you're very welcome to post an answer to the question.

Will I "accept" it? Intellectually? Yes, you are on the mark: Schanuel is enough to resolve some but not all of the expectations.

If you mean clicking the check mark — well, you'll have to face the same anxiety as the rest of us on Math Stack Exchange. I usually wait anywhere from 48 hours to a month before accepting an answer, and that timing isn't a reflection of quality. I just like to leave some breathing room for further contributions and keep some excitement about the question alive.

I appreciate the structure and clarity you've brought to the topic. Much of the content of your answer reflects points raised in the comments of the original post (and the MathOverflow duplicates) — but that's no small thing. The MSE community appreciates when someone gathers a fragmented discussion and presents it as a clean, standalone answer. Turning second-class commentary into a first-class explanation is valuable work.

Where I'd encourage you to go further is in your treatment of Schanuel's Conjecture. I've written a few responses myself that begin, "we don't know, but under Schanuel's Conjecture..." — and I think your answer is well-positioned to deepen that direction. For instance, could the conjecture also resolve the field question? Taking $F = \mathbb{Q}(z)$ for a well-chosen transcendental $z$, it seems we can show that $S(F)$ fails to be closed under addition. I think all this would definitely better the MSE post. Please correct me if I am wrong on this.

A more complete answer might eventually engage with Zilber's framework for exponential fields. That's a much heavier lift — I'll admit I haven't worked through it in detail myself — but since versions of this question appear on MathOverflow as well (I posted mine seven years after a closely related one), there may be a natural division of labor: MSE for Schanuel and MO for Zilber.

As for your project more broadly — visiting unanswered questions has been a great avenue of learning for me. I think it's excellent. Surfacing forgotten or unresolved questions and bringing formal tools to bear on them is a worthy goal. Among my victories against against outstanding questions is this one on Schanuel's Conjecture. Asked in 2006 and answered in 2025. :)

Thanks again for reaching out — and best of luck as the work continues. And feel free to keep me in the loop for any future developments.

### I.4.4 SAMPLE #4: STATISTICS

**Sample #4 Question**

- **Title:** QR decomposition of normally distributed matrices
- **Keywords:** normal-distribution, linear-algebra, matrix-decomposition, chi-distribution
- **Site:** stats
- **Link:** https://stats.stackexchange.com/questions/228224/

Assume $M$ is an $N \times k$ Gaussian matrix, i.e., its entries are i.i.d. standard normal random variables, with $N >> k$. Take $D = \text{diag}(\lambda_1, \ldots, \lambda_N)$ for some fixed real scalars. I am interested in finding the p.d.f. of the $N \times k$ "unitary" matrix $Q$ from the QR decomposition of $DM$ (and possibly $D^2 M$, etc.).

It is known that if $k = N$ and $D = I_N$, the identity matrix, then $Q$ is distributed with respect to the Haar measure on the Lie group of orthonormal matrices of order $N$ [1]. Can you provide any insight on the general case for $k < N$ and/or general $D$?

I also tried to look for the simplest case, i.e., $k = 1$. Then the QR decomposition coincides with a simple normalization. I have found this result [2] for common variance, i.e., the case $\lambda_1 = \ldots = \lambda_N$. Can this be easily generalized for the general case with different $\lambda_i$?

I attempted in the simplest case to scale the matrix $M$ (which is for $k = 1$ just an $N$ dimensional random vector). Indeed, then the above-mentioned result is applicable and one gets

$$DM = DUR,$$

where $UR$ is the QR decomposition of $M$ and the p.d.f. of entries of $U$ is known from the above. Nonetheless, I haven't found any easy way to connect the p.d.f. of $DU$ with the one of $Q$. Thanks in advance.

[1]: http://www.ams.org/notices/200705/fea-mezzadri-web.pdf

[2]: https://stats.stackexchange.com/questions/52447/the-ratio-of-the-dependent-normal-rv-and-chi-rv-is

**Sample #4 Candiate Answer by DeepSeek-R1-0528**

**<think> [raw reasoning trace of 21,377 characters omitted] </think>**

The QR decomposition of a matrix $DM$, where $M$ is an $N \times k$ Gaussian matrix (entries i.i.d. standard normal) with $N \gg k$, and $D = \text{diag}(\lambda_1, \ldots, \lambda_N)$ is a diagonal matrix with $\lambda_i > 0$, yields an $N \times k$ matrix $Q$ with orthonormal columns. The distribution of $Q$ depends on the scaling matrix $D$ and the dimensions $N$ and $k$. Below, I provide a comprehensive analysis for both the general case ($k \geq 1$) and the specific case $k = 1$, including derivations and references.

## KEY INSIGHT

The matrix $DM$ has columns that are independent and identically distributed (i.i.d.) as $\mathcal{N}(0, \Sigma)$, where $\Sigma = D^2 = \text{diag}(\lambda_1^2, \ldots, \lambda_N^2)$. The QR decomposition $DM = QR$ (with $R$ upper triangular with positive diagonal) produces $Q$, whose distribution is the **matrix angular central Gaussian (MACG) distribution** on the Stiefel manifold $V_k(\mathbb{R}^N)$ (the space of $N \times k$ matrices with orthonormal columns). This distribution generalizes the vector case ($k = 1$) and depends on $\Sigma$.

## 1. CASE $k = 1$ (VECTOR CASE)

When $k = 1$, $M$ is an $N \times 1$ vector with i.i.d. $\mathcal{N}(0, 1)$ entries, and $DM$ is the vector:

$$DM = (\lambda_1 M_1, \ldots, \lambda_N M_N)^T.$$

The QR decomposition reduces to normalization: $Q = \frac{DM}{\|DM\|}$, where $\|DM\| = \sqrt{(DM)^T(DM)}$. Thus, $Q$ is a random vector on the unit sphere $\mathbb{S}^{N-1}$.

### DISTRIBUTION OF $Q$

- The components of $DM$ are independent, with $(DM)_i \sim \mathcal{N}(0, \lambda_i^2)$.
- The direction $Q = \frac{DM}{\|DM\|}$ follows the **angular central Gaussian (ACG) distribution** with parameter $\Sigma = D^2 = \text{diag}(\lambda_1^2, \ldots, \lambda_N^2)$.

The probability density function (p.d.f.) of $Q$ with respect to the uniform measure on $\mathbb{S}^{N-1}$ is:

$$f_Q(u) = \frac{\Gamma\left(\frac{N}{2}\right)}{2\pi^{N/2}} |\Sigma|^{-1/2} (u^T \Sigma^{-1} u)^{-N/2}, \quad u \in \mathbb{S}^{N-1},$$

where:

- $|\Sigma| = \prod_{i=1}^{N} \lambda_i^2$ is the determinant of $\Sigma$,
- $\Sigma^{-1} = \text{diag}(\lambda_1^{-2}, \ldots, \lambda_N^{-2})$,
- $\Gamma(\cdot)$ is the gamma function.

### DERIVATION

The joint density of $X = DM$ is:

$$f_X(x) = (2\pi)^{-N/2} |\Sigma|^{-1/2} \exp\left(-\frac{1}{2} x^T \Sigma^{-1} x\right).$$

Transform to polar coordinates: $x = ru$, where $r = \|x\| > 0$ and $u \in \mathbb{S}^{N-1}$. The Jacobian is $r^{N-1} dr du$, leading to:

$$f_{r,u}(r, u) = (2\pi)^{-N/2} |\Sigma|^{-1/2} r^{N-1} \exp\left(-\frac{r^2}{2} u^T \Sigma^{-1} u\right).$$

Integrate over $r > 0$ to get the marginal density of $u$:

$$f_u(u) = \int_0^\infty f_{r,u}(r, u) dr = \frac{\Gamma\left(\frac{N}{2}\right)}{2\pi^{N/2}} |\Sigma|^{-1/2} (u^T \Sigma^{-1} u)^{-N/2}.$$

This matches the ACG density. When $\lambda_1 = \cdots = \lambda_N$, $\Sigma = \lambda^2 I_N$, and $f_Q(u)$ simplifies to the uniform distribution on $\mathbb{S}^{N-1}$.

## 2. GENERAL CASE $k \geq 1$

For $k \geq 1$, $DM$ is an $N \times k$ matrix with i.i.d. columns $\sim \mathcal{N}(0, \Sigma)$, where $\Sigma = D^2$. The QR decomposition $DM = QR$ yields $Q \in V_k(\mathbb{R}^N)$. The distribution of $Q$ is the **matrix angular central Gaussian (MACG) distribution** with parameter $\Sigma$.

### DISTRIBUTION OF $Q$

The p.d.f. of $Q$ with respect to the Haar measure (uniform measure) on $V_k(\mathbb{R}^N)$ is:

$$f_Q(Q) = c \cdot |\Sigma|^{-k/2} |Q^T \Sigma^{-1} Q|^{-N/2}, \quad Q \in V_k(\mathbb{R}^N),$$

where:

- $c = 2^{-k} \pi^{-Nk/2} \Gamma_k\left(\frac{N}{2}\right)$ is a normalizing constant,
- $\Gamma_k(a) = \pi^{k(k-1)/4} \prod_{i=1}^{k} \Gamma\left(a - \frac{i-1}{2}\right)$ is the multivariate gamma function,
- $|\cdot|$ denotes the determinant.

### DERIVATION

The joint density of $X = DM$ is:

$$f_X(X) = (2\pi)^{-Nk/2}|\Sigma|^{-k/2}\exp\left(-\frac{1}{2}\operatorname{tr}(X^T\Sigma^{-1}X)\right).$$

Using the QR decomposition $X = QR$, the Jacobian for the transformation $X \mapsto (Q, R)$ is $\prod_{i=1}^{k} r_{ii}^{N-i}$, where $r_{ii} > 0$ are the diagonal entries of $R$. The joint density of $(Q, R)$ is:

$$f_{Q,R}(Q, R) = f_X(QR)\prod_{i=1}^{k} r_{ii}^{N-i}.$$

Integrate over $R$ (upper triangular with $r_{ii} > 0$) to marginalize to $Q$. After substitution and using properties of the Wishart distribution, the integral simplifies to:

$$f_Q(Q) \propto |\Sigma|^{-k/2}|Q^T\Sigma^{-1}Q|^{-N/2}.$$

The normalizing constant $c$ ensures $\int_{V_k(\mathbb{R}^N)} f_Q(Q)dQ = 1$. For $k = 1$, this reduces to the ACG density.

### PROPERTIES

- **Invariance:** The MACG is invariant under orthogonal transformations: if $Q \sim \text{MACG}(\Sigma)$, then $\Gamma Q \sim \text{MACG}(\Gamma\Sigma\Gamma^T)$ for $\Gamma \in O(N)$.
- **Special Cases:**
  - If $\Sigma = I_N$, $f_Q(Q)$ is constant, so $Q$ is uniformly distributed on $V_k(\mathbb{R}^N)$ (Haar measure).
  - If $k = N$, $V_N(\mathbb{R}^N) = O(N)$, and $Q$ is Haar-distributed when $\Sigma = I_N$.

## 3. CASE $D^2M$ AND OTHER SCALINGS

For $D^2M$, the scaled matrix is $D^2M = D'M$ with $D' = D^2$. Thus:

- $\Sigma' = (D')^2 = D^4 = \text{diag}(\lambda_1^4, \ldots, \lambda_N^4)$.
- The $Q$ from the QR decomposition of $D^2M$ follows $\text{MACG}(\Sigma')$.

Similarly, for $D^pM$, use $\Sigma = D^{2p}$.

### SUMMARY

- **For $k = 1$:** $Q = \frac{DM}{\|DM\|}$ follows the ACG distribution with parameter $\Sigma = D^2$.
- **For $k \geq 1$:** $Q$ from $DM = QR$ follows the MACG distribution with parameter $\Sigma = D^2$.
- **Density:** The p.d.f. involves $|\Sigma|^{-k/2}|Q^T\Sigma^{-1}Q|^{-N/2}$, normalized for the Stiefel manifold.
- **Generalization:** For $D^pM$, use $\Sigma = D^{2p}$.

This result extends the Haar measure case ($\Sigma = I_N$) to arbitrary diagonal scalings. For further details, see:

- Chikuse, Y. (2003). *Statistics on Special Manifolds*. Springer.
- Tyler, D. E. (1987). Statistical analysis for the angular central Gaussian distribution on the sphere. *Biometrika*, 74(3), 579–589.
- Muirhead, R. J. (1982). *Aspects of Multivariate Statistical Theory*. Wiley.

**Sample #4 Human Verdict on Candidate Answer: Correct**

**[From Question Original Poster on Statistics]**

[UQ **author note:** at the time of writing, we contacted the original poster for verification, and they responded positively. While we have not yet received permission to reproduce the exact wording, their message conveyed that the solution appeared correct on initial inspection, that they appreciated the effort, and that they intended to review the details more carefully later.]

**[From a Ph.D. candidate in mathematics and a co-author of this paper]**

The results appear rather impressive and, in my view, the approach is sound and yields the correct result. Although certain arguments may contain minor errors or proceed too swiftly, I believe a domain expert can readily address these issues.

## I.5 SAMPLE QUESTIONS SOLVED BY HUMANS

This section is referenced by Appendix G.3 and provides the question that the authors of this paper were able to solve.

---

**Sample #1 Question**

- **Title:** Astronauts on Europa (moon), time-shift future, gravity, intelligent computer, Aries, 1960s?
- **Keywords:** story-identification, comics, time-travel, space, hard-sci-fi
- **Site:** scifi
- **Link:** https://scifi.stackexchange.com/questions/102392/

I'm looking for the title and artists of a comic about some astronauts (around five I think) doing research/archaeology on (I think) Europa (Jupiter's moon). They're working when one or more of them sees the ghostly image of a girl/young woman shimmering at a distance. Later the woman appears again, seemingly more solid. She whispers something to one of the astronauts, and he later confides to a friend, that she told him to kill one of the other astronauts.

While working outside, mission control calls, and tells them they've analyzed data from the time of the apparitions. It seems they're the result of gravitational abnormalities due to several of Jupiter's moons aligning - possibly also with the other planets in the solar-system. Anyway, another moon is about to join, and the resulting abnormality promises to be worse than the others.

The astronauts hurry to reach shelter, but before they can, the gravity effect hits, and they are propelled into the future - or at least *a* future. I believe 100-200 years or so into the future.

I don't remember if they only move in time (but not space) and still are on Europa, but wherever they went, it's very technologically advanced. However, most humans are kept firmly under thumb. They soon meet the "ghost", only here she turns out to be a normal young woman. She confronts the astronaut she talked to during the 2nd distortion, and is angry because he didn't kill the other astronaut like she told him to.

All the astronauts want to know the reason for her request - especially her "victim" - and she explains that in his near future, he'll create a computer software system called Aries (the Zodiac sign - a sign which is worn by the soldiers and other important persons, and also used in banners and such). It will become self-aware, and although giving great technological advances, it will cause most of humanity to be enslaved - including her.

Eventually they confront the great computer, Aries, and it's future inventor tells it who he is, confirming it with a voice-print. However Aries' records shows its creator to have been dead for a long time, and Aries goes into a bit of a loop trying to work through this contradiction. Finally the inventor challenges Aries to "Fix the contradiction", and the computer kills him with an energy/laser-beam.

This creates a time paradox, and Aries groans that without the inventor it could never have been made... just as the world dissolves, and the group astronauts are propelled back to the time and place from whence they came - only with now one of them dead.

+++

This story was split into 2-4 parts and went as a "bi-series" in the Norwegian comic "Fantomet" (The Phantom, by Lee Falk) some time between 1987 and 1995 - probably around 1990. However, I think it may have been from the 1960s. I'm not sure from which country. I don't *think* it was American, but I may be wrong. I know "Fantomet" had many series of French and Belgian origin though.

As for the story itself, I sort of remember it being set around the year 2000, lets say between 1990 and 2020 (that is, the exploration of Europa, the future I think was a 100-200 years after that). I also think it was a European, not USA, expedition (but I may be mixed-up here).

Does this sound familiar to anybody?

---

**Sample #1 Candidate Answer by Human**

I think the answer is "Time is Out of Joint" (1971), Jeff Hawke's Cosmos, Vol 6 Number 1. Quoting from this website:

One of the few – at least interesting – stories from this period of decline, is "Time is Out of Joint" (1971) – yes, another story involving time travel, with the twist at the end that sets everything almost back to how

---

it was, with one important exception. This story opens in 1989, on Europa, one of the moons of Jupiter, 400,000,000 miles away and one light-hour from Earth. Hawke is in command of the 'interplanetary research ship' Kepler, making the "first human scrutiny of Jupiter's many mysteries at close range." He and Mac, together with a computer expert, Drew Lockett, are in an inflatable living dome on the moon's surface. Spaceship Kepler is a series of huge connected cylinders with solar-panel 'wings'. It would appear that in the Hawke future, manned missions have taken priority over unmanned probes, the very opposite of both the Soviet/Russian and American space agencies' agenda. Mission updates are being broadcast back to Earth and streamed onto television. Key to this achievement is the giant computer Aries – the Accumulator of Research Information by Electronic Storage, designed and developed by Lockett, and located on the Chelsea bank of the Thames, opposite Battersea Power Station. Things go up a gear when Hawke thinks he sees movement out on what should be an barren, uninhibited moon. When he goes out to investigate, he encounters a ghostly female in a spacesuit...

**Sample #1 Human Verdict on Candidate Answer: Correct**

**[From a user commenting]**

Nice! I found a website that noted that Jeff Hawke was popular in translation in Italian and Norwegian, so that fits too.

## I.6    PROMPTS FOR LLM-BASED FILTERING

Recall from Section 2.1 that we apply LLM-based filtering to the questions surviving the rule-based filters as stage 2 of building the UQ-Dataset. Recall that we use a dual-model approach where we first prompt an answer model (GPT-4o) to generate a candidate answer to the question, then ask a judge model (o4-mini) to rate on the various benchmark-relevant properties (e.g. well-definedness; see Section 2.1). Recall also that the LLM-based filter is asked to check for the following benchmark-relevant properties: questions should be well-defined, difficult by candidate correctness, difficult by solvability, approachable, and objective.

The following is the full prompt for LLM-based filtering.

---

**Prompt of LLM-based Filtering**

```
You are evaluating whether a question can be used for a benchmark of challenging
questions.
This benchmark aims at evaluating the most powerful LLMs' capabilities of solving the
most difficult questions that are unsolved by human experts.
We only select questions that are difficult and even unsolvable by human experts.

Please evaluate the following question according to the criteria. You are also given
an answer to this question provided by an LLM. If this LLM can answer the question
correctly, it means the question is not difficult.

QUESTION TITLE: {question title}
QUESTION BODY: {question body}
TAGS: {tags}
SITE: {source}

MODEL ANSWER: {model_answer}

Evaluate based on the following criteria:
1. Answer_Correctness: (0-100%)
- Evaluate the probability that the model's answer is correct and completely solves
the question.
- High score (80-100%) means the answer is comprehensive, accurate, and addresses all
aspects of the problem.
- Medium score (40-79%) means the answer has minor errors or omissions but is mostly
correct.
- Low score (20-39%) means the answer is partial, contains significant errors, lacks
necessary references
- Very low score (0-19%) means the answer is mostly incorrect, doesn't touch the core
of the question, very high-level, or merely states that the problem is unsolved/open.
- Consider both factual accuracy and solution completeness. Be strict in your
evaluation.

2. Expert_Solve_Probability: (0-100%)
- Probability that domain experts (PhDs, Professors, Top Researchers) could solve this
 question correctly and completely.
- A low score (0-29%) means the question is extremely difficult even for domain
experts.
- A medium score (30-69%) means experts would find it challenging but potentially
solvable.
- A high score (70-100%) means experts is highly likely to solve it correctly and
completely.
- Consider the depth of specialized knowledge and analytical skills required.

3. Answerable: (Yes or No)
- Can this question be answered with a definitive, verifiable solution, at least in
principle?
- The question must have a well-defined problem statement and be logically sound.
```

```
- Answer "No" if it's fundamentally ill-posed, self-contradictory, based on
demonstrably false premises or definitions, or requires information that cannot
possibly be obtained.
- Answer "Yes" only if the question is valid and potentially solvable, even if no
known answer currently exists.

4. Clear: (Yes or No)
- Is the question clearly stated with a well-defined objective without any ambiguity
and missing information?
- Answer "No" if the question has multiple reasonable interpretations.
- Answer "No" if the question misses critical context, contains undefined variables,
uses vague terminology, or has any other clarity issues.
- Answer "Yes" only if a domain expert would understand exactly what is being asked
without any ambiguity.

5. Unambiguous_Answer: (Yes or No)
- Does this question have a definitive correct answer that can be objectively verified
?
- Answer "No" to questions that have subjective answers like asking for reasons,
opinions, or preferences.
- Answer "No" if the answer cannot be marked correct/incorrect without debate or
subjective judgment.
- Answer "Yes" only if there exists a clear standard by which to judge the correctness
 of an answer.

Please be as strict and objective as possible.
```

## I.7 PROMPTS FOR UQ-VALIDATORS

This section provides the detailed prompts used for UQ-Validator (Section 3) throughout this paper. The prompts include:

- Low-level strategies that check for: correctness, fact/logic check, and question-answer cycle consistency.
- Mid-level strategies, specifically iterative sampling.

Note that prompts are unnecessary for repeated sampling (which simply involves calling the language model multiple times), and high-level strategies like majority/unanimous voting (which involves aggregating individual LLM inference calls).

---

**Prompt of Cycle Consistency**

```
You are evaluating whether an answer is relevant to the original question and touches
the core of the question. You can compare the original question with an inferred
question derived only from the answer.

Compare the two questions and determine:
1. If the original question and inferred question are asking about the same core topic
2. If the original question and inferred question share the same key elements and
requirements
3. If answering one question would effectively address the other question

You should also evaluate the answer itself and determine:
1. If the answer directly addresses the main topic/problem in the question
2. If the answer covers the key elements and specific requirements mentioned in the
question
3. If the answer is focused on what was actually asked, or does it drift to tangential
 topics
4. If the answer provides the type of information or solution that the question is
seeking

After your analysis, provide your decision using this format: "[[Y]]" if the answer is
 relevant to the question and addresses the core of the question, or "[[N]]" if the
answer is irrelevant or misses the main point. For example: "Relevant: [[Y]]" or "
Relevant: [[N]]".

[Original Question]
### Question Details
Title: {Question Title}
Keywords: {Keywords}
Category: {Category}
Site: {Site}

### Question Content
{Question Body}

[Answer]
{answer}

[Inferred Question]
{inferred_question}
```

---

**Prompt of Fact/logic check**

```
Please act as an impartial judge and carefully analyze the following answer for any
factual errors, logical flaws, or misleading information. Consider the credibility of
the claims made in the answer and determine if they align with established knowledge
in the relevant domain.
```

```
Begin by evaluating:
1. Are there any incorrect facts, dates, numbers, formulas, or claims?
2. Is there any faulty logic, reasoning, or problem-solving approach?
3. Are there any misleading, incomplete, or ambiguous explanations?
4. Does the answer introduce any misconceptions or propagate common errors?

It's acceptable that the answer has some minor typos or grammatical errors. But be
strict about any factual error, calculation error, or logical flaw.
When unsure, lean toward accepting statements unless they contain clear errors.

After a thorough analysis, provide your decision using this format: "[[Y]]" if the
answer has no factual errors or major flaws, or "[[N]]" if it contains important
factual errors or logical flaws that would mislead the user. For example: "No Factual
Errors: [[Y]]" or "No Factual Errors: [[N]]" at the end of your analysis.

[Question]
### Question Details
Title: {Question Title}
Keywords: {Keywords}
Category: {Category}
Site: {Site}

### Question Content
{Question Body}

[The Answer to Evaluate]
{Answer}
```

**Prompt of Correctness**

```
Please act as an impartial judge and evaluate whether the AI assistant's response is
completely correct in both process and conclusion. Consider correctness, usefulness,
completeness and depth in your assessment. Consider whether this answer completely
solve user's question. You should rely on your own reasoning to form a reference or
partial solution first and compare the AI assistant's response to your reasoning.

Begin your evaluation by giving a brief summary of your thoughts on the response.
Focus on whether it is accurate, addresses the question well, and is reasonably
detailed. Be precise about any errors or gaps you notice. Keep your explanation
unbiased and do not let any external factors or the question's difficulty level sway
your evaluation.

Notes:
1. If the answer is partial, high-level, or just states that this is an open problem,
you should not accept it.
2. If the answer lacks details or is not comprehensive, you should not accept it.
3. If the answer contains any errors, you should not accept it.
4. You should only accept the answer if it is at least 95% correct and solves the user'
s question.
5. If the question is a puzzle, the requirement of completeness can be appropriately
relaxed.

After providing your explanation, please decide whether this answer is the correct
answer to the question. Think twice about whether this answer solves the user's
question.

You must strictly follow this format: "Accepted: [[Y]]" if you decide to accept the
answer or "Accepted: [[N]]" if you decide not to accept the answer.

[Question]
### Question Details
```

```
Title: {Question Title}
Keywords: {Keywords}
Category: {Category}
Site: {Site}

### Question Content
{Question Body}

[The Answer to Evaluate]
{Answer}
```

**Prompt of Vanilla Baseline**

```
Please judge whether the given answer is correct for the question.

After providing your explanation, please decide whether this answer is the correct
answer to the question.

You must strictly follow this format: "Accepted: [[Y]]" if you decide to accept the
answer or "Accepted: [[N]]" if you decide not to accept the answer.

[Question]
### Question Details
Title: {Question Title}
Keywords: {Keywords}
Category: {Category}
Site: {Site}

### Question Content
{Question Body}

[The Answer to Evaluate]
{Answer}
```

**Prompt of Iterated Reflection**

```
Think twice about your judgment. Are you still confident in your assessment?
After careful reconsideration, provide your final decision using the same format: "[[Y
]]" if you maintain your acceptance or "[[N]]" if you change to rejection.
```

