# OpenReview forum: "UQ: Assessing Language Models on Unsolved Questions"
_ICLR.cc/2026/Conference — Submitted to ICLR 2026_

### Official Review · Reviewer_axDw · 2025-10-27

**Soundness:** 3
**Presentation:** 4
**Contribution:** 3
**Rating:** 4
**Confidence:** 2

**Summary:**

This paper presents UQ, a benchmark assessing LMs on unsolved questions. UQ consists of 500 challenging, diverse questions that do not have solutions yet. Together with these questions, this paper presents UQ-Validators (validation strategies that pre-screen candidate solutions) and UQ-Platform. On SOTA LMs, this paper observes unique insights.

**Strengths:**

- Currently there are a lot of LM evaluation datasets where the answers are already online, and perhaps even already in the LMs' memories. The UQ dataset overcomes that significant drawback and introduces a novel perspective in constructing datasets.
- Also, since these questions are unsolved yet, the solution to these problems can yield direct real-world value.
- The analyses on the LMs' problem-solving strategies produce interesting findings. Also, the discussion that connect the validation strategies to test-time scaling is interesting.
- The UQ-Platform is also a valuable empirical contribution that lays the foundation for future works.

**Weaknesses:**

- The design of the UQ-Validator (especially the mid-level and the high-level strategies) seems a bit off target to me. I think the validator should check for the problem-solving strategy, and then provide an estimation regarding whether the attempted solution is "on the right path". Following this design goal, the low-level strategies make sense to me, but I don't think the mid-level and the high-level strategies (which aim at coping with the randomness of LLM-as-judges) reach the correct goal.
- Considering the pass rates for SOTA LMs are very low, I am afraid that the UQ-Validator allows too small of a subset of candidate solutions. Perhaps the LLMs don't need to pass these strategies to solve these problems?
- While the study that transfers from the HLE to UQ improves the solidness of the UQ-Validator, the granularity of transfer is on the generator-validator answer accuracy gap feels a bit too coarse to me. I would find a more fine-grained (and more direct) evaluation more convincing: to validate the detection of problem-solving strategies.

**Questions:**

Could you respond to the items in the "weaknesses" section? I look forward to understanding more about the rationales behind the design.

---

> ### Author Response · Authors · 2025-11-21
> **Response to Reviewer axDw**
>
> We are grateful for the insightful feedback and valuable suggestions. Below, we address each proposed point one by one:
>
> ---
>
> > [W1] I think the validator should check for the problem-solving strategy, and then provide an estimation regarding whether the attempted solution is "on the right path".
>
> > [W3] I would find a more fine-grained (and more direct) evaluation more convincing: to validate the detection of problem-solving strategies.
>
> We thank the reviewer for this thoughtful suggestion. We’d like to clarify that:
>
> - **Evaluating strategy detection is desirable in principle but hard in practice.** For truly unsolved questions, even domain experts frequently do not know what the “right” high-level strategy should look like, and may reasonably disagree about promising approaches. This makes it extremely difficult to construct reliable labels for “good strategy vs. bad strategy”.
>
> - **Detecting problem-solving strategies is an open research problem in itself.** For example, what’s considered a “problem-solving strategy”? How do we evaluate if the strategy is good or bad? This problem is out-of-scope of our study.
>
> - **Current validators focus on what is reliably checkable.** Our validator is intentionally designed to focus on aspects that are currently feasible and reliably checkable: detecting factual errors, logical inconsistencies, misuse of known results, and clear internal contradictions, rather than only judging whether the overall strategy is on the right track.
>
> > [W1] But I don't think the mid-level and the high-level strategies (which aim at coping with the randomness of LLM-as-judges) reach the correct goal.
>
> The goal of our designed mid- and high-level strategies is to aggregate and stabilize low-level checks to reduce bias and variance. Single-prompt judgers tend to be biased towards themselves and siblings (as shown in Section 3.3), while our experiments show that compound validators reduce bias and improve precision/accuracy.
>
> > [W3] The granularity of transfer is on the generator-validator answer accuracy gap feels a bit too coarse to me.
>
> We’d like to clarify that **accuracy on HLE and HLE→UQ transfer still provides meaningful evidence of efficacy.** HLE is a very diverse dataset spanning 35+ subjects, so achieving good performance on it is non-trivial and indicates that the validator captures broadly useful correctness signals. Using validators calibrated on HLE and applying them to UQ without further tuning provides a coarse but informative sanity check that these signals generalize beyond a single dataset.
>
> > [W2] I am afraid that the UQ-Validator allows too small of a subset of candidate solutions.
>
> Thank you for the question. We’d like to clarify that **low pass rates are expected for long-unsolved, frontier-level questions.** UQ questions are high-vote, long-unsolved problems that have remained unanswered for years and are challenging even for top human experts. In such a regime, it is **natural and desirable** that current SOTA models pass only a small fraction of instances. Higher pass rates would likely indicate that the benchmark is too easy or already saturated.
>
> > [W2] Perhaps the LLMs don't need to pass these strategies to solve these problems?
>
> We totally agree with the reviewer. As mentioned in our paper, we always defer verdicts to the UQ-platform, precisely because validators may not be perfect.
>
> Instead of serving as oracles, **the role of validators is to identify a small subset of answers that look particularly promising (or suspicious) to prioritize for expert review**, not to reject all other answers as definitively wrong. Since we show all the validation results on UQ-Platform, the validator also provides reasoning traces that help experts verify answers more quickly.
>
>
> ---
>
> We hope this addresses the reviewer’s concern, and we are glad to provide additional details if needed.

---

### Official Review · Reviewer_yanc · 2025-10-30

**Soundness:** 2
**Presentation:** 3
**Contribution:** 2
**Rating:** 4
**Confidence:** 3

**Summary:**

The paper proposes a benchmark including unsolved problems from CS, math, science, and history. Also, provide a complete pipeline including data filtering, evaluation, and building a platform for experts to continuously update the dataset

**Strengths:**

1. The motivation to design both difficult and realistic benchmarks, which provide challenging practical questions, is novel and meaningful
2. The paper proposes a reference-free validation method to evaluate the correctness of the LLM response on the unlabeled question-answer datapoints
3. Focusing on currently unsolved problems stimulates the usage of LLM in answering new research questions or exploring new research directions

**Weaknesses:**

1. The reason behind the dataset creation criteria is unclear. How the rules set for the rule-based filter and LLM-based filter contribute to the difficulty and realism is not clearly illustrated.
2. Some processes use the LLM to provide the label. The reliability of the LLM on such tasks is unclear. For example, the “approachable: whether the question is logically sound and solvable in principle”, seems to be a challenging task for LLM to provide a reliable label.
3. The definition of unsolvable is general. Without further analysis, we can not figure out whether the reason for the LLM’s failure is insufficient knowledge or reasoning capability. Since multiple factors can make questions unsolvable, performance on this dataset does not clearly isolate which capability is being evaluated.
4. To achieve an oracle-free validator is challenging, but it is required with good performance to guarantee the quality of the benchmarks. The current performance level on the HLE dataset seems to be unsatisfying. Additionally, the validator is not well-connected with the human evaluation part. The paper mentions that the experts rate the responses that passed the validation. The low precision indicates that the expert labels are still significant to control the dataset quality.

**Questions:**

1. In the LLM-based filter stage, what is the accuracy of the LLM labels on each property checking task?
2. For the validator stage, instead of the binary label, would it be better to provide a numerical probability like a confidence score? A carefully designed threshold could then balance precision and recall, or identify questions requiring further human annotation.

---

> ### Author Response · Authors · 2025-11-21
> **Response to Reviewer yang (1/n)**
>
> We thank the reviewer for the insightful feedback and valuable suggestions.
>
> Below, we address each raised point individually:
>
> ---
>
> > [W1] The reason behind the dataset creation criteria is unclear. How the rules set for the rule-based filter and LLM-based filter contribute to the difficulty and realism is not clearly illustrated.
>
> We thank the reviewer for raising this question. We want to first clarify that **the filtering process is not meant to contribute to realism**. By “realistic”, we mean questions that arise when people genuinely want to know the answer but do not have it. The unsolved questions sourced from Stack exchange are inherently realistic. Then we’d like to clarify how the rules contribute to the difficulty.
>
> - **Rule-based filtering**.
>   * We require questions to be old enough (e.g., >2 years), to filter out questions that are unsolved simply because they are too new to have received attention.
>   * We remove questions with low engagement (few votes/views), to filter out those that are unsolved mainly because they were never really seen by the community rather than because they are hard.
>   * We remove questions with answers. Some questions have correct answers but are still marked as unsolved on Stack Exchange because the ask hasn’t accepted the answer. We want to filter out these questions to prevent contamination.
>
> - **LLM-based filtering**. We use GPT-4o to provide a candidate answer, and judge the correctness (0-100%) of the answer. During development, we observed that for genuinely hard unsolved questions, GPT-4o’s answers typically fail to touch the core of the question; thus, if GPT-4o can already give a reasonable or partially correct solution, we treat the question as too easy and exclude it. This step makes the dataset harder by discarding questions that are already within reach of strong generalist models.
>
> > [W2] Some processes use the LLM to provide the label. The reliability of the LLM on such tasks is unclear.
>
> > [W2] “approachable” seems to be challenging.
>
> We thank the reviewer for highlighting this concern. First we’d like to clarify that **the LLM-prefiltering stage is used to reduce the candidate pool** to a manageable size, since the original set of questions is far too large for exhaustive human review. We do not assume that an LLM can perfectly decide whether a question is logically sound and solvable in principle. Even if this filter were imperfect, we would still retain a large pool (7000+) that necessarily contains many good questions.
>
> We agree that “approachable” is non-trivial for LLMs. Therefore, we provide instructions such as “Answer ‘No’ if it's fundamentally self-contradictory, based on false premises or definitions, or requires information that cannot possibly be obtained.” to help LLMs make decisions. Besides, The LLM’s “approachable” label is used as a conservative heuristic filter to flag obviously ill-posed / speculative questions. Borderline or high-value candidates are kept for later human inspection.
>
> > Q1: what is the accuracy of the LLM labels on each property in LLM-based prefiltering?
>
> In the LLM prefiltering stage, we have continuous scores (e.g., Difficult by candidate correctness and Difficult by solvability) and binary properties. For binary criteria, we measure agreement with human annotators on randomly selected 30 questions.
>
> | Criteria              | Human-LLM agreement |
> | --------                | --------         |
> |  Well-defined     |    86.7%     |
> |  Approachable   |    80.0%     |
> |  Objective          |    96.7%     |
>
> We here provide two examples that human experts agree with the LLM prefilter.
>
>
> https://math.stackexchange.com/questions/3591355/probability-of-a-group-being-finite
> https://math.stackexchange.com/questions/358423/a-proof-of-dimrt-dimr1-without-prime-ideals
>
> > [W3] The definition of unsolvable is general. Performance on this dataset does not clearly isolate which capability is being evaluated.
>
> We thank the reviewer for the opportunity to clarify. First we would love to clarify that UQ uses “unsolved” but not “unsolvable”. Our focus is on unsolved questions for which the community has not yet produced a full solution. We do not claim these questions are “unsolvable” in principle.
>
> - **UQ-Dataset is intentionally a composite problem-solving benchmark**. We agree that multiple factors can cause failures. UQ-Dataset is therefore not designed to isolate a single micro-skill; it is meant to probe end-to-end problem solving in realistic settings. This is similar to many successful benchmarks like HLE which also do not clearly disentangle capabilities.
>
> - **Our next step can annotate capability types explicitly**. For example, annotating questions with the types of skills the questions seem to require and analyzing failure modes accordingly.

---

> > ### Author Response · Authors · 2025-11-21
> > **Response to Reviewer yang (2/n)**
> >
> > > W4: The current performance level on the HLE dataset seems to be unsatisfying. The low precision indicates that the expert labels are still significant to control the dataset quality.
> >
> > We agree with the reviewer that validators aren’t perfect. However, we’d like to clarify that:
> > - **Validators are designed to pre-screen, not verify solutions**. Ground truth verification is the role of the UQ-platform, and UQ isn’t complete with all three stages (dataset, validator, and platform).
> > - **Validators improve over time**. In Fig. 4 we show that validation capability grows faster than generation capability and that this generator-validator gap transfers. This trend suggests that with stronger models, the same evaluation paradigm of UQ gets better without modification.
> > - **Compound pipelines mitigate biases**. In Fig. 5, we demonstrate that known biases such as self-preference can be meaningfully addressed.
> > - **High-precision is inherently challenging on hard, unsolved questions**. In Finding #2 (Section 3.3), we provide relevant discussions on the inherent difficulty of high precision.
> > - **Even imperfect validators are still useful**. In Appendix E.3 we show that examining the reasoning traces of validators provide useful insights into why a solution is wrong.
> > - **Validators should be viewed as a research blueprint, not the final solution**. By demonstrating that with even imperfect validation: (1) structured multi-step validators can reduce bias relative to naïve judges, and (2) their reasoning traces are accurate enough to be trusted as explanatory signals, UQ-Validator defines a clear path for future work.
> >
> > > Q2: For the validator stage, instead of the binary label, would it be better to provide a numerical probability like a confidence score?
> >
> > We thank the reviewer for this thoughtful suggestion!
> >
> > - **Confidence scores are appealing but require good calibration**. We agree that numerical confidence scores could, in principle, enable more operations (e.g., tuning precision vs. recall, or flagging answers for additional human review). This would be particularly useful if the scores are well-calibrated.
> >
> > - **We use binary labels to keep the initial protocol simple**. Our pipeline already aggregates multiple judgment steps and reasoning traces; introducing a confidence score on top of these requires nontrivial additional design choices.
> >
> > - **Richer outputs are a natural direction for future work**. We view confidence scores and uncertainty estimates as a promising extension rather than something we ruled out. In future iterations of UQ-Validator, we plan to explore richer outputs (e.g., calibrated probabilities, multi-level verdicts) that can improve the overall paradigm.
> >
> > ---
> >
> > Thank you for your suggestions and we hope our responses address the reviewer’s concern

---

### Official Review · Reviewer_o1JY · 2025-11-01

**Soundness:** 2
**Presentation:** 3
**Contribution:** 2
**Rating:** 2
**Confidence:** 4

**Summary:**

The paper introduces UQ, a benchmark of unsolved questions sourced from Stack Exchange. The benchmark includes 500 questions curated through rule-based filtering, LLM filtering and human review. Model responses are judged by LLM validators that aim to rule out wrong answers via a hierarchical set of strategies such as correctness, consistency, logic structure etc. The dataset is hosted on a platform where domain experts can comment on questions, check model answers and even provide solutions to the problems. The paper argues that this paradigm increases realism while remaining difficult in a non-artificial way, compared to other exam-like benchmarks.

**Strengths:**

- The paper explores alternatives to artificial exam-like benchmarks and tries to tackle the problem of benchmarks with highly difficult but unrealistic problems.
- The collection pipeline is well curated, with several types of filters and explicit filtering criteria such as well-posedness, difficulty etc.
- The fact that the benchmark is hosted on a live platform helps checking question quality, model answers and even provide solutions to problems
- Validators are useful to rule out wrong answers, and there is a clear pipeline for that.

**Weaknesses:**

- The paper suggests that unsolved problems are “by construction” realistic. However, there are several unsolved problems that are really hard, but artificial. Conversely, not all difficult solved problems are inherently unrealistic and artificial. As an example, research problems are unsolved but still natural for the context. It’d be better to add some more evidence about the realism claim.
- Given the absence of ground truth answers and with validators being indicative but not conclusive (according to the authors), comparison between models is not grounded. UQ resembles more a testbed+platform rather than a traditional benchmark.
- The dataset is heavily weighted towards math-related problems (>60%), with several categories having only 1 problem. This might reflect the Stack Exchange supply, but makes the dataset less diverse.

**Questions:**

- It would be helpful to expand a bit and clarify the results in table 2, in particular the human pass rate.
- A more detailed discussion on realism and difficulty, with some examples, would help clarify this tradeoff.
- Given the heavy math presence in the dataset, are you planning on expanding other categories?

---

> ### Author Response · Authors · 2025-11-21
> **Response to Reviewer o1JY (1/n)**
>
> We thank the reviewer for the insightful feedback and valuable suggestions. Below, we carefully address your concerns, grouped by their topics.
>
> ---
>
> > [W1] However, there are several unsolved problems that are really hard, but artificial.
>
> > [W1] As an example, research problems are unsolved but still natural for the context.
>
> > [W1] It’d be better to add some more evidence about the realism claim.
>
> > [Q2] more detailed discussion on realism and difficulty, with some examples
>
> We thank the reviewer for raising these points. We first want to clarify that:
>
> - “Artificial” has a narrower meaning in our definition. It does not mean “any question written by a human.” Rather, **we use “artificially created questions” to denote questions that are constructed together with their solutions specifically for evaluation or examination.** In such benchmarks, difficulty and structure are tuned to test particular skills or expose model weaknesses.
>
> - **“Realistic” questions are those people genuinely want answers to but do not know.** The reviewer’s example of research problems fits exactly into this category and is the type of unsolved question UQ aims to capture.
>
> - For additional realism claim, **UQ-Dataset is grounded in real user activity and human votes**. All questions in UQ-Dataset are drawn from actual user posts on real platforms, with human engagement signals (e.g., votes, views) and then further filtered for clarity and objective verifiability. This means problems in UQ-Dataset are typically those real people found important enough to articulate and discuss.
>
> Compare how school teachers make exam problems vs how students run into problems on the first job; students who do well may not be those that excel at real-world problem solving, and this is the “distribution shift” UQ wants to avoid.
>
> > [W1] Conversely, not all difficult solved problems are inherently unrealistic and artificial.
>
> Based on our definitions above, we totally agree that many difficult solved problems are realistic. For example, researchers propose a research question and then it’s solved. However, we’d like to emphasize that difficult solved problems, even if “realistic”, can be vulnerable to cheating. Our concern is that when such problems are turned into static benchmarks with known answers, they become vulnerable to cheating, contamination, and overfitting to those known solutions. And that’s why we introduce UQ as a new evaluation paradigm.
>
> > [W2] UQ resembles more a testbed+platform rather than a traditional benchmark.
>
> We’d like to emphasize that **UQ represents a new evaluation paradigm, not a static benchmark**. Unlike common benchmarks that are create-once & run-once, UQ continuously evaluates and tracks model performance; we instead revisit model evaluation from the ground up, in exchange for making progress on genuinely unsolved, rather than contrived problems. This is a core part of our contribution rather than a weakness.
>
> Testbed-like behavior is a feature for unsolved questions. The UQ-Validators and UQ-Platform are designed to evolve with the frontier and to host community evaluations on genuinely unsolved questions. We believe this is what is needed in the unsolved-question regime.

---

> > ### Author Response · Authors · 2025-11-21
> > **Response to Reviewer o1JY (2/n)**
> >
> > > [W3] The dataset is heavily weighted towards math-related problems (>60%), with several categories having only 1 problem. This might reflect the Stack Exchange supply, but makes the dataset less diverse.
> >
> > > [Q3] Given the heavy math presence in the dataset, are you planning on expanding other categories?
> >
> > We thank the reviewer for pointing out this question. We’d like to clarify that:
> >
> > - **Even narrow-domain benchmarks can still be informative**. Many widely used benchmarks are in fact highly concentrated in one domain (e.g., AIME, Omni-MATH), yet they still provide signals about model capabilities on that class of problems.
> >
> > - **The STEM skew arises naturally from our scoping and filtering pipeline**. UQ-Dataset is explicitly scoped to questions that are clearly defined, objective, and verifiable. For dataset construction, we manually choose 80 sites for crawling. Sites whose questions are mainly subjective (e.g., asking for advice in Pets site) are discarded. We then apply multi-stage filtering to filter out questions with low engagement or are not clearly-defined, subjective, etc. It’s natural that the remaining questions are heavily weighted towards math. Figure 4 shows the change of distribution after each filtering stage. Sparse domains are often those with very few questions, typically those where user posts are rarer, more subjective, or lack a clear, checkable notion of correctness.
> >
> > - **UQ-Dataset is live and will be periodically updated**. As described in Section 2.3, UQ-Dataset is designed as a live resource: when at least 20% of questions are marked as solved, we will refresh the dataset with new unsolved questions. This naturally creates opportunities to increase diversity.
> >
> > - **Future releases aim to broaden categories via new sources and submissions**. In future platform releases, we plan to (i) incorporate additional sources beyond the current Stack Exchange sites, and (ii) introduce a question submission system that allows researchers and practitioners to contribute their own unsolved questions.
> >
> > > [Q1] expand a bit and clarify the results in table 2, in particular the human pass rate
> >
> > Thanks for raising the question. We first clarify the metrics:
> > - % answers passed UQ-Validator: fraction of evaluated model answers that pass the UQ-Validator.
> > - % answers passed human-reviewers or human pass rate: fraction of the answers that human experts judge to be correct.
> > - Human/UQ-Validator judgment agreement: fraction of answers where humans and the UQ-Validator give the same binary verdict (correct vs. incorrect).
> > - Human-rated accuracy of UQ-Validator reasoning trace: fraction of validator reasoning traces that humans judge as substantively correct, not just whether the final verdict is correct.
> >
> > Then we’d like to clarify the results:
> > - The validator pass rate vs. human pass rate comparison shows that the validator is more optimistic than human experts, but **the relative ranking of models by validator roughly matches the ranking by humans**.
> > - The high human / validator agreement rate indicates that, despite some optimism, **the validator’s binary decisions align with human experts on most answers**.
> > - The high human-rated accuracy of validator reasoning traces is a stricter metric where humans inspect the whole reasoning trace. The high scores here show that **the validator’s explanations are largely correct and informative**. Since these reasoning traces are shown on UQ-Platform, they provide useful guidance for human verification and help experts quickly understand why an answer is likely correct or incorrect.
> >
> > ---
> >
> > Hope our response addresses the above concerns and we are happy to add anything else that helps clarify.

---

> > > ### Comment · Reviewer_o1JY · 2025-11-25
> > >
> > > Thank you for the detailed clarifications and for the additional explanation of the validator metrics and human pass rate. The paradigm is interesting and potentially valuable, but I still find difficult to interpret scores and metrics for questions with unknown ground truth.

---

### Official Review · Reviewer_JCCv · 2025-11-01

**Soundness:** 2
**Presentation:** 2
**Contribution:** 3
**Rating:** 4
**Confidence:** 5

**Summary:**

The paper introduces UQ, a novel benchmark designed to evaluate frontier AI models on genuinely unsolved questions sourced from communities like Stack Exchange. The authors argue that current benchmarks are either too artificial or too easy. UQ addresses this by curating a dataset of 500 difficult and realistic problems, developing an LLM-based system (UQ-Validators) to pre-screen answers, and launching a public platform for ongoing community verification. This paradigm shifts AI evaluation from solving known problems to tackling the frontiers of human knowledge, making progress on the benchmark directly valuable.

**Strengths:**

+ A significant strength is the creation of the UQ-Platform. The authors acknowledge that automated validation is insufficient for these complex, open-ended problems. By building an ecosystem for human experts to review, rate, and verify AI-generated answers, they create a sustainable, long-term evaluation method that can adapt as models improve.

+ The benchmark's design directly measures a model's ability to solve problems that are currently unsolved by humans. This is a powerful concept because success isn't just about matching human performance on a solved task.

**Weaknesses:**

- The core premise, using unsolved questions, creates a fundamental verification bottleneck. Without ground truth, evaluation relies on the UQ-Validators (which are imperfect) and human experts. Sourcing and compensating a diverse pool of domain experts to verify a large volume of complex answers is costly, time-consuming, and not easily scalable.

- The UQ-Validator pipeline is intricate and depends on LLMs judging other LLMs, a process known to have biases. The paper's own results show the best validator has limited precision (~40%), meaning it incorrectly passes many wrong solutions (false positives) and may discard correct ones (false negatives). This questions the reliability of the automated screening process.

- Because the problems are unsolved, a model's failure doesn't necessarily indicate a flaw. The problem might be intractable. This makes it difficult to distinguish between a model's capability gap and the inherent difficulty of the question, unlike benchmarks with known solutions where failure is a clear signal.

- The entire UQ-Validator system is built on the hypothesis that validating an answer is significantly easier than generating it. While the paper provides evidence for this on the HLE dataset, this assumption may not hold true for all types of unsolved problems. For deeply complex proofs or creative tasks, constructing the verification can be as intellectually demanding as creating the solution itself.

**Questions:**

The paper contrasts "artificially difficult" questions with UQ's "naturally arising" ones. Could you elaborate on your definition of "artificial" and provide more detail on how this leads to the "distribution shift" you aim to avoid?

You define a "difficult" benchmark as one that challenges frontier models, yet you critique exam-style benchmarks for being quickly saturated. How do you reconcile these points? What properties of unsolved questions make their difficulty more durable and a better long-term measure of capability?


Considering that researchers often increase the complexity of problems after solving easier ones, what is the core disadvantage of creating progressively harder artificial questions? Why is preserving the "realism" of naturally-arising problems prioritized over creating a more scalable and continuously challenging artificial benchmark?


In the LLM-based filtering stage, you mention using three repeated calls to judge each question. Could you provide data or analysis on the variance of these judgments? How did you validate that these scores were stable and reliable enough to serve as a critical filter in your pipeline?

The final UQ-Dataset contains 500 questions, curated from an initial pool of over 3 million. How did you end up only on exactly 500 questions.

The paper states that benchmarks based on user queries often skew towards "easy to articulate" problems. How does sourcing questions from Stack Exchange, a user-query-based platform, avoid this same potential bias?

During the LLM-based filtering stage, a model (GPT-4o) is used to generate a candidate answer to help assess the question's quality. What was the rationale for choosing this specific model? Was there a concern that using a more powerful, state-of-the-art model at this stage might solve too many questions, thereby misclassifying them as "easy"?

UQ-Dataset is not a representative sample of "unsolved questions" but rather a sample of popular, well-defined, STEM-oriented questions that remain unanswered on a specific set of websites. Progress on this benchmark may not generalize to other forms of frontier problem-solving.

Because the questions are so difficult, models are expected to solve very few of them. As seen in the initial results, the top model solved only 4 out of 500 questions after human verification. This creates an extremely sparse signal. A model could make significant internal improvements in its reasoning capabilities but still not be powerful enough to solve another UQ question, resulting in a reported score of zero progress.

What question is not artificially created? Artificial is well-defined.

---

> ### Author Response · Authors · 2025-11-21
> **Response to Reviewer JCCv (1/n)**
>
> We appreciate your thoughtful feedback and believe they make our paper stronger. We hope to carefully address them below, grouped by topic.
>
> ---
>
> > [W1] The core premise, using unsolved questions, creates a fundamental verification bottleneck.
>
> We’d like to emphasize that UQ aims to create a new evaluation paradigm, not a static benchmark. Unlike common benchmarks that are create-once & run-once, UQ continuously evaluates and tracks model performance; and unlike exam-style questions, every increment of UQ performance adds to human knowledge. In this sense, the verification bottleneck (as the reviewer points out) is necessary and inevitable, because knowledge creation requires human-in-the-loop.
>
> > [W1] Sourcing and compensating a diverse pool of domain experts to verify a large volume of complex answers is costly, time-consuming, and not easily scalable.
>
> We agree that human verification is costly and time-consuming. In the same vein as above, we believe that this is inevitable for the new paradigm, which needs not to be scalable at first. We’d also like to emphasize that our design of UQ-Validators and UQ-Platform mitigate the cost of human experts.
>
> > [W1] evaluation relies on the UQ-Validators (which are imperfect)
> > [W2] UQ-Validator pipeline is intricate … known to have biases.
> > …the best validator has limited precision (~40%).
>
> We totally agree with the reviewer that validators aren’t perfect. However, we’d like to clarify that:
> 1. **Validators are designed to pre-screen, not verify solutions**. Ground truth verification is the role of the UQ-platform, and UQ isn’t complete with all three stages (dataset, validator, and platform).
> 2. **Validators improve over time**. In Fig. 4 we show that validation capability grows faster than generation capability and that this generator-validator gap transfers. This trend suggests that with stronger models, the same evaluation paradigm of UQ gets better without modification.
> 3. **Compound pipelines mitigate biases**. In Fig. 5, we demonstrate that known biases such as self-preference can be meaningfully addressed.
> 4. **High-precision is inherently challenging on hard, unsolved questions**. In Finding #2 (Section 3.3), we provide relevant discussions on the inherent difficulty of high precision.
> 5. **Even imperfect validators are still useful**. In Appendix E.3 we show that examining the reasoning traces of validators provide useful insights into why a solution is wrong.
> 6. **Validators should be viewed as a research blueprint, not the final solution**. By demonstrating that with even imperfect validation: (1) structured multi-step validators can reduce bias relative to naïve judges, and (2) their reasoning traces are accurate enough to be trusted as explanatory signals, UQ-Validator defines a clear path for future work.
>
>
> > [W4] … UQ-Validator … is built on the hypothesis that validating an answer is significantly easier than generating it … this assumption may not hold true for all types of unsolved problems. For deeply complex proofs or creative tasks, constructing the verification can be as intellectually demanding as creating the solution itself.
>
> We agree with the reviewer in principle. Indeed, in the limit, this is the essence of the P vs NP problem—is generation as easy as verification? We just don’t know.
>
> Pragmatically, however, if the generator-validator gap exists and widens with increasing model capability (Section 3.1) on problems as diverse as HLE (spanning math, chemistry, legal, humanities, …), it surely warrants further exploration, which is what UQ-validators set out to do, and as mentioned in the paper, we always defer verdicts to the UQ-platform, precisely because validators may not be perfect.
> Importantly, we’d also like to clarify that **UQ is explicitly scoped to a subclass of unsolved questions that are clearly-defined, objective and unambiguous**, which are generally easier to validate than open-ended, creative problems. Indeed, we empirically show that within this scope, verification can be easier than generation; for example, candidate answers typically expose locally checkable structure (e.g., intermediate claims, references, or sub-lemmas) that can be scrutinized **without having to discover the solution**.
>
> > [W3] Because the problems are unsolved, a model's failure doesn't necessarily indicate a flaw.
>
> We clarify that this is indeed the role of the UQ-platform. Human experts are the ultimate arbiters on model performance, and their reviews can be rated by other humans (Appendix F).

---

> > ### Author Response · Authors · 2025-11-21
> > **Response to Reviewer JCCv (2/n)**
> >
> > > [Q1] elaborate on your definition of "artificial" and provide more detail on how this leads to the "distribution shift" you aim to avoid?
> > > [Q10] What question is not artificially created? Artificial is well-defined.
> >
> > “Artificial” simply means that a question is constructed with the solution in mind, often specifically targeting known model weaknesses. “Realistic” means questions reflect real-world value and use cases.
> >
> > Compare how school teachers make exam problems vs how students run into problems on the first job; students who do well may not be those that excel at real-world problem solving, and this is what we referred to as the “distribution shift”.
> >
> > UQ aims to construct model evals that match real-world use cases. We don’t deny that there exists well-designed benchmarks with known solutions (e.g. terminal bench), though clearly those are more grounded in the real-world scenarios than purely artificial exam problems.
> >
> > > [Q2] You define a "difficult" benchmark as one that challenges frontier models, yet you critique exam-style benchmarks for being quickly saturated. How do you reconcile these points?
> > > [Q3] what is the core disadvantage of creating progressively harder artificial questions? Why is preserving the "realism" of naturally-arising problems prioritized?
> >
> > We first clarify that we don’t claim “exam-style benchmarks are always quickly saturated”. Exam-style benchmarks can indeed be made very difficult (L64); it’s just that some exams are quickly saturated. It is always possible to construct an exam that makes LLMs score zero (e.g. choosing niche domains or manually design convoluted puzzles); we mainly claim that they may not be realistic—i.e., reflecting real-world needs.
> >
> > > [Q2] What properties of unsolved questions make their difficulty more durable and a better long-term measure of capability?
> >
> > We believe there’s a misunderstanding. We don’t claim that their difficulty is generally “more durable” or they are a “better long-term measure”; we simply claim that unsolved questions are typically both difficult and realistic. Indeed, some difficult and realistic unsolved questions can be solved quickly (in fact, 4 has been solved since we started this project), and that their solutions yield real-world value.
> >
> > > [Q4] … Could you provide data or analysis on the variance of these judgments? How did you validate that these scores were stable and reliable enough to serve as a critical filter in your pipeline?
> >
> > First, we clarify that the LLM-prefiltering stage is only used to reduce the candidate pool to a manageable size, since the original set of questions is far too large for exhaustive human review. Even if this filter were imperfect, we would still retain a large pool (7000+) that necessarily contains many good questions.
> > To assess stability, we report the mean standard deviation across three repeated judgments on 50 randomly selected samples. Variance across the three runs is low:
> >
> > | Criteria                      | Mean std over 3 judgments  |
> > | ----------------------------- | ----  |
> > | Well-defined (0 or 1)         | 0  |
> > | Approachable (0 or 1)         | 0.01  |
> > | Objective (0 or 1)            | 0     |
> > | Candidate Correctness (0-100) | 5.85  |
> > | Expert Solvability (0-100)    | 10.63 |
> >
> > As an additional sanity check, we measured human–LLM agreement on 30 randomly selected questions for the three binary criteria:
> >
> > | Criteria              | Human-LLM agreement |
> > | --------                | --------         |
> > |  Well-defined     |    86.7%     |
> > |  Approachable   |    80.0%     |
> > |  Objective          |    96.7%     |
> >
> >
> > > [Q5] How you end up only on exactly 500 questions?
> >
> > It’s a design choice and nothing special about the number 500 (e.g. HLE picks 2500 problems, GPAQ diamond has 198) other than that this is a reasonable number balancing human review bandwidth and dataset size and diversity.
> >
> > > [Q6] benchmarks based on user queries often skew towards "easy to articulate" problems; what above Stack Exchange, a user-query-based platform?
> >
> > We’d like to clarify that by user queries, we meant prompts akin to those in Chatbot Arena / WildChat datasets. In contrast, Stack Exchange has active moderation, question quality standards, and community editing. Question in UQ (such as [1]) are generally more well-thought and well-formatted also due to our collection pipeline for ensuring quality.
> >
> > [1] https://math.stackexchange.com/questions/358423/a-proof-of-dimrt-dimr1-without-prime-ideals

---

> > > ### Author Response · Authors · 2025-11-21
> > > **Response to Reviewer JCCv (3/n)**
> > >
> > > > [Q7] why GPT-4o?
> > >
> > > GPT-4o was a stable and widely-used generalist model at the time of dataset construction. GPT-4o is also cost-efficient compared to reasoning models (we have limited budget). Another reason is that filtering based on GPT-4o answers give lower false negatives – if a (now weak) generalist model can get it correct, then the question is very likely too easy.
> > >
> > > > [Q8] Progress on this benchmark may not generalize to other forms of frontier problem-solving.
> > >
> > > As mentioned earlier, we clarify that UQ is not meant as a benchmark, but a new evaluation paradigm. UQ-Dataset is just its first instantiation and is explicitly scoped to well-defined, objectively checkable problems, which naturally emphasizes STEM-like tasks. We do not claim that UQ-Dataset covers all frontier problem types (Appendix C.1); extending UQ beyond STEM and beyond Stack Exchange to other domains and task types is important future work for this new paradigm.
> > >
> > > > [Q9] sparse signal. A model could make significant internal improvements .. but still not be powerful enough to solve another UQ question.
> > >
> > > Sparse success is an inherent property of frontier unsolved questions: if small internal model changes reliably moved many questions from unsolved to solved, the task would likely be too easy or contaminated. We do not solely rely only on “# fully solved”: UQ-Validator pass rates provide a denser, more sensitive signal that can reflect improved reasoning and intermediate steps even without full solutions. Solving even a handful of UQ questions is meaningful (which is the design goal of collecting “realistic” questions), since we prioritize high-vote, high-view questions where each genuinely new solution is a substantial contribution.

---

### Author Response · Authors · 2025-11-26
**General Response**

We thank all reviewers for their thoughtful feedback, and will incorporate them to improve our work. We are delighted that the reviewers:
- found that our proposed evaluation paradigm is forward-looking and meaningfully differs from existing benchmarks (JCCv, o1JY, yanc, axDw);
- believe that the creation of UQ-platform meaningfully contributes to real-world utility (yanc), adds sustainability of the new evaluation paradigm (JCCv), and lays a foundation for future work (axDw);
- found our experimental findings interesting and the paradigm mitigates test set contamination (axDw);
- found the UQ-validator pipeline useful (o1JY, yanc); and
- found the paper well-presented (o1JY, yanc, axDw).

---

We now address common feedback from the reviewers here, and provide detailed rebuttal individually.

## Clarification on “artificial” vs. “realistic” and disadvantages of “artificially difficult” benchmarks (JCCv, o1JY, yanc)

- By "artificial," we mean that questions are explicitly designed with both the solutions and often the model weaknesses in mind. This process can skew toward "tricking the model" rather than reflecting realistic use cases where models struggle. In contrast, “realistic” means that the questions reflect real-world value and use cases.

- UQ questions are posted by humans who are seeking answers to problems they find interesting (including research problems as noted by Reviewer o1JY). They reflect a real information need rather than contrived questions that the question authors may not care about (but are nonetheless compensated to create). Crucially, all UQ questions are grounded in engagement signals (e.g. upvotes by the community and view counts), indicating that people found them important enough to articulate and discuss.

- Artificially difficult questions, on the other hand, become vulnerable to contamination once turned into static benchmarks. Moreover, progress on such benchmarks may not reflect real-world utility because they may be contrived, ad-hoc, and otherwise not grounded in real-world interest (unlike the UQ-dataset).

We emphasize that UQ provides a complementary paradigm to such static model evaluations, where progress requires genuine problem-solving with reduced risk of pattern-matching to training data.

## Clarification on verification bottleneck and low precision of UQ-Validators (JCCv， yanc)

We appreciate this critical feedback. We want to emphasize several key points:

- **The verification bottleneck is our research question (not a weakness)**. UQ is explicitly designed to study evaluation in the absence of ground truth, a setting that traditional benchmarks avoid but that characterizes real frontier problems. In this paradigm, human verification is necessary and inevitable, because knowledge creation requires human-in-the-loop. Rather than claiming to have solved this challenge, we provide a systematic framework and initial algorithms as a foundation for future work.
- **Validators improve over time**. In Fig. 4 we show that validation capability grows faster than generation capability and that this generator-validator gap transfers. This trend suggests that with stronger models, the same evaluation paradigm of UQ gets better without modification.
- **Validators are pre-screening tools, not final oracles**. Answers passing the UQ-Validator are flagged for expert review. This design provides two key benefits (i) Validators let experts prioritize a smaller, high-yield subset rather than reviewing all generations randomly (ii) Table 2 shows validator reasoning has ~92% human-rated accuracy. Even when the binary verdict is imperfect, the explanations help experts quickly understand potential issues.
- **Achieving high precision is inherently difficult in an imbalanced setting**. For genuinely unsolved questions, current models have very low true success rates, leading to extreme class imbalance. The reported ~40% precision represents substantial enrichment over random inspection (base rate ~2-8%), which is already valuable for triage.

---

**Finally, we’d like to reiterate that UQ aims to create a new evaluation paradigm, not a static benchmark**. Unlike common benchmarks that are create-once & run-once, UQ continuously evaluates and tracks model performance; and unlike exam-style questions, every increment of UQ performance adds to human knowledge.

We hope this clarifies the overarching design choices; the subsequent sections address each reviewer’s comments in detail.

---

### Meta-Review · Area_Chair_RTgd · 2025-12-16

**Summary:**

The paper is well written and introduces a novel benchmark designed to evaluate frontier models on realistic yet challenging questions. The proposed framework does not require ground truth answers and is claimed to support continuous evaluation.
The reviewers collectively raised several valid and substantive concerns, which I largely share. My primary concerns are detailed below.

1. **Lack of Justification for "Real-World" Claims**.
Although the authors' rebuttal partially clarified the distinction between "artificial" and "realistic" questions, I agree with Reviewers [JCCv, o1JY, and yanc] that the "realistic" component requires more evidence and discussion. Specifically, since terms such as "real-world value," "real-world utility," and "real-world interest" are repeatedly emphasized, the authors must provide stronger justification to support the magnitude of these claimed benefits.

2. **Challenges in Evaluation and Validator Precision**.
As noted by Reviewers [JCCv, yanc], if the problems posed are excessively difficult, several factors could contribute to model failure (e.g., insufficient model capability, lack of necessary input knowledge, or ill-posed questions). These confounding factors not only complicate the comparative evaluation of different models but also pose significant challenges for validators seeking to produce meaningful feedback.
This issue closely links to the concerns that the validators exhibit low precision [JCCv, yanc, axDw] and that the signals returned by these validators are inherently difficult to interpret [o1JY]. The authors state that the oracle-free validator is intended to identify a small subset of answers that appear particularly promising (or suspicious) to prioritize for expert review. It would be highly beneficial to quantify the utility of this functionality, perhaps by measuring its efficiency or impact.

3. **Insufficient Evidence for Continuous Evaluation**.
A key feature claimed by the authors is "continuous evaluation," with the rebuttal asserting that "every increment of UQ performance adds to human knowledge." However, the evidence provided to support the claimed benefit of this continuous evaluation mechanism is limited to a few screenshots of user interactions shown in Appendix F. More substantial data or statistics are necessary to fully substantiate the advantages of this feature.

**Reviewer Concerns:**

There are many concerns or questions raised by all reviewers. I here list a few major ones.
In my opinion, the authors have addressed the following concerns via the rebuttal:

- Details about LLM-based filtering stage [Reviewer JCCv, yanc]
- Diversity/imbalance of problem categories [Reviewer JCCv, yanc]

The following concerns are still outstanding:
- Dataset creation criteria: artificial vs. realistic [Reviewer JCCv, o1JY, yanc]
- Confounding factors for hard questions: hard to isolate the individual factors like the model capability gap, insufficient input knowledge, and so on [Reviewer JCCv, yanc]
- Verification bottleneck: low-precision or reliable validator [Reviewer JCCv, yanc] and inconclusive feedback of validator [o1JY]

**Reviewer Scores:**

If they had been able to participate fully in the discussion, the following reviewers would potentially consider changing the scores.
- Reviewer JCCv and yanc could be satisfied with the clarification on the LLM-based filtering part.
- Reviewer axDw could be satisfied with the response on the mid-level and the high-level strategies (which aim at coping with the randomness of LLM-as-judges) do not reach the correct goal.

---

### Decision · Program_Chairs · 2026-01-26

Reject